# From Optimization Dynamics to Generalization Bounds via Łojasiewicz Gradient Inequality

**Fusheng Liu**                                                              *fusheng@u.nus.edu*
*Institute of Data Science*
*National University of Singapore*

**Haizhao Yang**                                                              *hzyang@umd.edu*
*Department of Mathematics*
*University of Maryland College Park*

**Soufiane Hayou**                                                           *hayou@nus.edu.sg*
*Department of Mathematics*
*National University of Singapore*

**Qianxiao Li**                                                              *qianxiao@nus.edu.sg*
*Department of Mathematics*
*National University of Singapore*

**Reviewed on OpenReview:** *https://openreview.net/forum?id=mW6nD3567x*

## Abstract

Optimization and generalization are two essential aspects of statistical machine learning. In this paper, we propose a framework to connect optimization with generalization by analyzing the generalization error based on the optimization trajectory under the gradient flow algorithm. The key ingredient of this framework is the Uniform-LGI, a property that is generally satisfied when training machine learning models. Leveraging the Uniform-LGI, we first derive convergence rates for gradient flow algorithm, then we give generalization bounds for a large class of machine learning models. We further apply our framework to three distinct machine learning models: linear regression, kernel regression, and two-layer neural networks. Through our approach, we obtain generalization estimates that match or extend previous results.

## 1 Introduction

From the perspective of statistical learning theory, the goal of machine learning is to find a predictive function that can give accurate predictions on new data. For supervised learning problems, empirical risk minimization (ERM) is a common practice to achieve this goal. The idea of ERM is to minimize a cost function on observed data using an optimization algorithm. Therefore, a fundamental question is: *given a training algorithm, does it produce a solution with good generalization*? This question has been the subject of a substantial body of literature, which answers this question in terms of the implicit bias of optimization methods such as stochastic gradient descent (SGD). For example, one line of works considered the case where gradient methods converge to minimal norm solutions on kernel regression (Bartlett et al., 2020; Tsigler & Bartlett, 2020; Liang & Rakhlin, 2020; Liang et al., 2020), and then analyzed the generalization properties of those minimal norm solutions by bias–variance tradeoff. Another line of works focused on the Neural Tangent Kernel (NTK) regime (Allen-Zhu et al., 2019; Arora et al., 2019; Cao & Gu, 2020; Ji & Telgarsky, 2020; Chen et al., 2021) where SGD iterates converges to a global minimum with a short distance from initialization. They suggested to use norm-based measures to theoretically derive generalization bounds. Specifically, these papers studied the generalization of (deep) neural networks on a norm-constrained parameter space

$\mathcal{W} = \{W : W \in \mathcal{B}(0, R)\}$, where $W$ is the collection of weight matrices for all layers, and $\mathcal{B}(0, R)$ is a ball with radius $R$. On the NTK regime, SGD with random initialization is proved to be able to find a global minimum in the parameter space $\mathcal{B}(0, R)$ with $R = \sqrt{y^{\top} \left(\Theta^{(L)}\right)^{-1} y}$, where $y$ is the label and $\Theta^{(L)}$ is the NTK matrix defined on the training input data (Arora et al., 2019; Cao & Gu, 2020). All of these works have made significant progress on the interplay of optimization and generalization. However, the settings that they focused on are specific in the sense that (1) the phenomenon of norm minimization has only been proven to occur with the quadratic loss with an appropriate initialization scheme; (2) $R$ has been theoretically obtained only in the NTK regime; (3) these works has considered the models after convergence and the generalization analysis of models during training are not completely understood. Therefore, the connection between optimization and generalization still remains incomplete understandings in general scenarios.

In this paper, we aim to tackle these issues by studying the connection between optimization and generalization of a wide class of machine learning models. Thus, a more precise analysis of how the parameter evolves when the training time varies is needed. For this purpose, inspired by the Łojasiewicz gradient inequality (LGI) condition in Bolte et al. (2007), which states that the gradient norm is lower bounded by some power of the value function, we consider a uniform version of LGI that extends this condition to a non-vanishingly small set. Specifically, we propose the *Uniform-LGI* (Definition 2.1), which is a modified version of the LGI. This assumption plays a critical role in connecting optimization and generalization through norm-based generalization bounds. The first section is dedicated to the numerical validation of the Uniform-LGI on different machine learning models. By introducing the *local* Uniform-LGI condition along the optimization path, we derive convergence rates and generalization bounds that yield bias-variance tradeoffs during training. Our framework can be applied to a broad class of machine learning models to obtain optimization results and generalization estimates.

**Contributions.** Our contributions are three-fold:

- First, we design a finite sample test algorithm to verify the Uniform-LGI condition along the training path. Our numerical results suggest the Uniform-LGI condition is generally satisfied when training machine learning models and is more general compared to the Polyak-Łojasiewicz (PL) condition (Polyak, 1963). Then, we propose a framework for connecting optimization dynamics and generalization performance based on the Uniform-LGI condition.

- Specifically, we first analyze the convergence rate for the loss functions that satisfy the Uniform-LGI condition (Theorem 2.2) under the gradient flow algorithm. Through Rademacher complexity theory, we derive a generalization bound (Theorem 2.3) for a wide class of hypothesis spaces that holds during the training process. The generalization bound exhibits bias–variance tradeoff pattern.

- We illustrate different use cases of our framework, showing how we obtain generalization estimates for a linear regression problem (Theorem 3.3), kernel regression (Theorem 3.5), and two-layer neural networks (Theorem 3.8 for shallow networks & Theorem E.8 for overparameterized case). These bounds are derived in a unified way, and either match existing results derived for individual cases or expand upon the scenarios where we can rigorously establish the phenomenon of benign overfitting.

## 2 Main Results

In this section, we present our main results. We first introduce the notations and the problem setting in Section 2.1. We then in investigate the key ingredient, the Uniform-LGI condition of our framework in Section 2.2. Lastly, we derive optimization results and generalization bounds for the gradient flow trajectory in Section 2.3 under the Uniform-LGI condition.

### 2.1 Setup and Notations

Consider a hypothesis space $\mathcal{F} = \{f(w, \cdot) : \mathbb{R}^d \to \mathbb{R} \mid w \in \mathcal{W}\}$, where $\mathcal{W}$ is a parameter set in Euclidean space. Given a loss function $\ell : \mathbb{R} \times \mathbb{R} \to \mathbb{R}$, and a training set $S = \{(x_i, y_i)\}_{i=1}^{n} \subseteq \mathbb{R}^d \times \mathbb{R}$ with $n$

independent and identically distributed (i.i.d.) samples from a joint distribution $\mathcal{D}$, the goal of ERM is to optimize the empirical loss function $\mathcal{L}_n(w)$ on $S$:

$$\min_w \mathcal{L}_n(w) := \frac{1}{n} \sum_{i=1}^{n} \ell\left(f(w, x_i), y_i\right). \tag{1}$$

**Notations.** We use $\|\cdot\|$ to denote the $\ell_2$ norm of a vector or the spectral norm of a matrix, and use $\|\cdot\|_F$ to denote the Frobenius norm of a matrix. For two vectors, we use $\langle,\rangle$ to denote their inner product. For a symmetric matrix $A$, we use $\lambda_{\min}(A)$, resp. $\lambda_{\max}(A)$ to denote the smallest, resp. the largest, eigenvalue of $A$. For any non-negative integer $n$, let $[0:n] = \{0, 1, \ldots, n\}$. We use $\mathcal{O}(\cdot)$ to denote the Big-O bound.

## 2.2 Investigation on Uniform-LGI

Now we introduce the key component of our framework: the Uniform-LGI, a condition that holds for a wide class of loss functions. We give several examples of loss functions that satisfy the Uniform-LGI. Our numerical results suggest that the Uniform-LGI is generally satisfied when training neural network models.

The classical LGI gives a lower bound on the gradient of a differentiable function based on its value above its minimum. Many functions, e.g., real analytic functions and subanalytic functions, satisfy this property, at least locally (Bolte et al., 2007). Here, we require a uniform version of this inequality as a condition to control the optimization trajectory. Let us define this notion below.

**Definition 2.1** (Uniform-LGI). A loss function $\mathcal{L}(w)$ satisfies Uniform-LGI on a set $\mathcal{S}$ with constants $\theta \in [1/2, 1)$ and $c > 0$, if

$$\|\nabla \mathcal{L}(w)\| \geq c \left(\mathcal{L}(w) - \min_{v \in \mathcal{S}} \mathcal{L}(v)\right)^{\theta}, \; \forall w \in \mathcal{S}. \tag{2}$$

The Uniform-LGI is a more general condition than the well-known PL-condition (Karimi et al., 2016) in the sense that 1) The $\mu$-PL condition is a special case for the Uniform-LGI when $c = \sqrt{2\mu}, \theta = 1/2$. The Uniform-LGI contains a wider class of functions than the PL-condition. For instance, consider $\mathcal{L}(w) = w^{2k}$, this function satisfies the PL-condition only when $k = 1$. When $k \geq 2$, it satisfies the general Uniform-LGI with $\theta = 1 - 1/2k$. Figures 1(a), 1(b), Figure 2(a), 2(b), 2(c) show that many nontrivial setups do not have $\theta = 1/2$ in practice. Besides, the well-known PL-condition in (Karimi et al., 2016) is a defined in a neighborhood of a *global* minimum, indicating that every stationary point is a global minimum, while the Uniform-LGI is more general that is defined in an arbitrary set, where there may exist no global minimum. This yields that the Uniform-LGI covers more scenarios than the PL-condition when analyzing optimization properties as it is difficult for an optimization algorithm to find a global minimum in practice.

Note that this is an inequality, so the pair of $(c, \theta)$ such that the Uniform-LGI holds is not unique. Usually the smallest $\theta$ is defined as the Uniform-LGI exponent. The classical LGI states that for any subanalytic function (Bolte et al., 2007) including a.e. differentiable models with squared loss, cross-entropy loss and hinge loss, the exponent $\theta$ is strictly less than 1. In the following, we give some examples of loss functions satisfying the Uniform-LGI globally over their entire domains or locally along the optimization path.

**Global Uniform-LGI.** Let the loss function $\mathcal{L}(w_L, \ldots, w_1) = (w_L \cdots w_1)^2$, which can be viewed as a one-dimensional $L$-layer linear neural network model with squared loss on the data $(1, 0)$. Then $\|\nabla \mathcal{L}\|^2 = 4(w_L \cdots w_1)^4 \sum_{i=1}^{L} 1/w_i^2 \geq 4L(w_L \cdots w_1)^{4-2/L} = 4L(\mathcal{L})^{2-1/L}$. Therefore, $\mathcal{L}$ satisfies Uniform-LGI condition globally on $R^L$ with $c = 2\sqrt{L}$ and $\theta = 1 - 1/2L$ for $L \geq 1$.

Apart from the global Uniform-LGI, there exists a large number of loss functions satisfying the Uniform-LGI locally within a given range. Since we want to study the optimization and generalization of a training algorithm, next we investigate the Uniform-LGI condition along the optimization path during training.

**(Local) Uniform-LGI along the optimization path.** Given an initialization $w^{(0)}$, a gradient-based optimization algorithm $\mathcal{A}$ (e.g. GD, SGD, etc.) produces a series of parameters $w^{(0)}, w^{(1)}, w^{(2)}, \ldots, w^{(k)}$ for the first $k$ steps. If $\mathcal{A}$ converges, $k$ can be infinite and the limiting parameter $w^{(\infty)}$ is a stationary point of $\mathcal{L}(w)$. To numerically verify the Uniform-LGI condition (find $\theta, c$) along the entire optimization path

$\{w^{(i)}\}_{i=0}^{\infty}$, i.e., find $c, \theta$ such that $\log \|\nabla \mathcal{L}(w)\| \geq \log c + \theta \log \left(\mathcal{L}(w) - \min_{v \in \mathcal{S}} \mathcal{L}(v)\right)$ holds on $S = \{w^{(i)}\}_{i=1}^{\infty}$, we design an algorithm for finite sample test. First, for the first $k$ collected data points, we consider to use linear regression to fit these $k$ data points and get the slope $\theta_k$, then we adjust $c$ to catch the worst case, i.e., set $c_k = \min_{i \in [0:k-1]} \|\nabla \mathcal{L}(w^{(i)})\| / \left(\mathcal{L}(w^{(i)}) - \mathcal{L}(w^*)\right)^{\theta_k}$. Then the obtained $c_k, \theta_k$ are the constants such that the Uniform-LGI holds for the first $k$ data points. While numerical experiments can never *prove* an inequality, it is useful in estimating the constants by looking at the limiting values as the number of points increases. This is called finite-size scaling analysis (Privman, 1990) and is frequently used in fields such as statistical physics to investigate numerically the behaviour of infinite-size systems (e.g. phase transitions).

---

**Algorithm 1** Finite sample test for Uniform-LGI

---

**Input:** loss function $\mathcal{L}(w)$; a collection of parameters $w^{(0)}, w^{(1)}, w^{(2)}, \ldots, w^{(K)}$; the optimal loss value $\mathcal{L}(w^*)$; start point $K_0$; step $s$

1: **for** $k = 0$ to $K$ **do**
2:     calculate the gradient norm $\|\nabla \mathcal{L}(w^{(k)})\|$
3: **end for**
4: **for** $k = K_0$ to $K$ step $s$ **do**
5:     collect the $k$ data $\left\{\left(\log \left(\mathcal{L}(w^{(i)}) - \mathcal{L}(w^*)\right), \log \|\nabla \mathcal{L}(w^{(i)})\|\right)\right\}_{i=0}^{k-1}$
6:     fit the data by linear regression, and return the slope $\theta$
7:     $\theta_k \leftarrow \theta$
8:     $c_k \leftarrow \min_{i \in [0:k-1]} \frac{\|\nabla \mathcal{L}(w^{(i)})\|}{\left(\mathcal{L}(w^{(i)}) - \mathcal{L}(w^*)\right)^{\theta_k}}$
9: **end for**
10: fit the data $\left\{\left(\log \left(\mathcal{L}(w^{(i)}) - \mathcal{L}(w^*)\right), \log \|\nabla \mathcal{L}(w^{(i)})\|\right)\right\}_{i=0}^{k}$ by linear regression, and return the slope $\theta$
11: $\theta_{K+1} \leftarrow \theta$
12: $c_{K+1} \leftarrow \min_{i \in [0:K]} \frac{\|\nabla \mathcal{L}(w^{(i)})\|}{\left(\mathcal{L}(w^{(i)}) - \mathcal{L}(w^*)\right)^{\theta_{K+1}}}$

**Output:** the estimated Uniform-LGI constants $(\theta_{K_0}, c_{K_0}), (\theta_{K_0+s}, c_{K_0+s}), \ldots, (\theta_{K+1}, c_{K+1})$

---

Note that the pair of $(c, \theta)$ such that the Uniform-LGI holds is not unique, and the output $(\theta_{K+1}, c_{K+1})$ is *one* possible pair of constants such that the Uniform-LGI holds on the first $K$ collected data points. Intuitively, if Algorithm 1 outputs a series of estimated Uniform-LGI constants that converge to some $(\theta^*, c^*)$, then we have good evidence to support the assumption that $\mathcal{L}(w)$ satisfies equation (2) along the optimization path $\{w^{(i)}\}_{i=0}^{\infty}$ with $\theta^*, c^*$. Moreover, if $\theta \in [1/2, 1)$ and $c > 0$, then $\mathcal{L}(w)$ satisfies the Uniform-LGI on $\{w^{(i)}\}_{i=0}^{\infty}$. In the following, we present numerical experiments that confirms the validity of the Uniform-LGI along GD/SGD paths on synthetic models and neural network models under Algorithm 1.

**Synthetic models.** We consider training three synthetic models by GD with optimal loss values $\mathcal{L}(w^*) = 0$, for which we can *provably* estimate the Uniform-LGI constants: (a) differentiable but non-analytic [1] loss function $\mathcal{L}(w) = e^{-\frac{1}{|w|}}$ for $w \neq 0$, 0 for $w = 0$. For the non-analytic model, by the classical LGI property we know that there exists no $\theta$ and $c$ such that the Uniform-LGI holds around the minimum $w = 0$, which is consistent with Figure 1(a); (b) differentiable loss function $\mathcal{L}(w) = \frac{1}{4} w^{\frac{4}{3}} + \frac{1}{2} w^2$. For this loss function, we have $\frac{|\nabla \mathcal{L}(w)|}{\mathcal{L}(w)^{\theta}} \sim \mathcal{O}(w^{\frac{1-4\theta}{3}})$. Therefore, the Uniform-LGI constant $\theta$ should be at least $1/4$, which is consistent with the result in Figure 1(b) that the estimated $\theta = 0.308$; (c) undetermined linear regression model $\mathcal{L}(w) = \frac{1}{2n} \sum_{i=1}^{n} (w^\top x_i - y_i)^2$ on a synthetic dataset. We know that the squared loss is always strongly convex, thus it satisfies the PL-condition ($\theta = 0.5$), which is consistent with Figure 1(c) that the estimated $\theta \approx 0.5$. More details about the experiments are provided in Appendix A.

We report the estimated Uniform-LGI constants $(\theta_{K_0}, c_{K_0}), (\theta_{K_0+s}, c_{K_0+s}), \ldots, (\theta_{K+1}, c_{K+1})$ by finite sample test (Algorithm 1), and denote by $(\theta^*, c^*)$ the final estimation $(\theta_{K+1}, c_{K+1})$. Figure 1 shows three different cases: (a) $(\theta, c)$ do not converge; (b) $(\theta, c)$ converge with $\theta^* < 1/2$ and $c^* > 0$; (c) $(\theta, c)$ converge with $\theta^* \approx 0.5$. This demonstrates that our method is robust in determining whether a model satisfies the Uniform-LGI.

---

[1] A real function is said to be analytic if it possesses derivatives of all orders and agrees with its Taylor series in a neighborhood of every point.

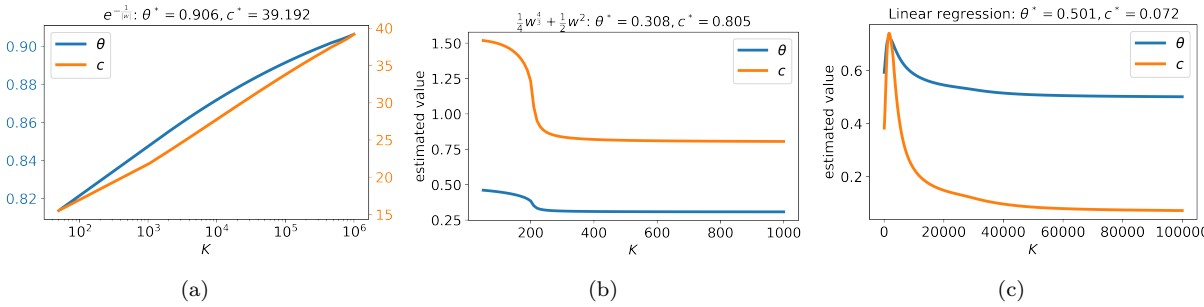

(a)          (b)          (c)

Figure 1: Investigation of the Uniform-LGI on synthetic models by finite sample test (Algorithm 1). For the non-analytic model (a), both $\theta$ and $c$ do not converge. For the synthetic model (b), $c$ converges to a positive number but $\theta$ converges to a number that is $< 1/2$. For the linear regression model (c), the Uniform-LGI holds along the training path with $\theta^* \approx 0.5$ as expected since linear regression satisfies the PL condition.

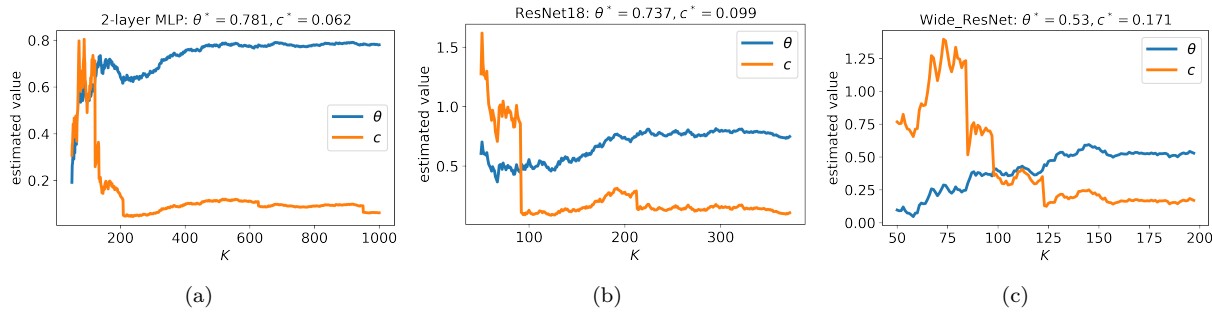

(a)          (b)          (c)

Figure 2: Investigation of the Uniform-LGI condition on neural network models by finite sample test (Algorithm 1). For all of these models, $\theta$ and $c$ converges to $\theta^* \in [1/2, 1)$ and $c^* > 0$ respectively. Hence, these two models satisfy the Uniform-LGI condition along the optimization path with constants $\theta^*$ and $c^*$.

**Neural network models.** We train three neural network models: two-layer multilayer perceptron (MLP) with width 100 (no bias) on the MNIST dataset (LeCun et al., 1998); ResNet18 (He et al., 2016), Wide-ReseNet-16-8 (Zagoruyko & Komodakis, 2016) on the CIFAR10 dataset (Krizhevsky et al., 2009). For each experiment, we train the network using SGD (no momentum) with random shuffling, batch size 64 and fixed learning rate 0.01. For the MLP model, we stop the training with 1000 epochs and set the optimal loss value to be 0. For the ResNet models, we stop the training once the cross-entropy loss is less than 0.001 and estimate the local optimal loss value as $\min_{k \in [0:K]} \mathcal{L}(w^{(k)})$, which is smaller than $10^{-3}$. To avoid division by zero, we delete the parameter $w^*$ in $w^{(0)}, w^{(1)}, w^{(2)}, \ldots, w^{(K)}$. Then we apply the finite sample test (Algorithm 1) with starting point $K_0 = 50$, step $s = 1$ to obtain the estimated Uniform-LGI constants.

We report $(\theta_{K_0}, c_{K_0}), (\theta_{K_0+s}, c_{K_0+s}), \ldots, (\theta_{K+1}, c_{K+1})$, and denote $(\theta^*, c^*)$ by the average of the last 5 iterates, i.e., $\theta^* = (\theta_{K-3} + \cdots + \theta_{K+1})/5, c^* = (c_{K-3} + \cdots + c_{K+1})/5$. As shown in Figure 2, the estimated $(\theta, c)$ converge for both models, and the Uniform-LGI holds along the optimization path.

Table 1 in Appendix A reports the confidence intervals of $\theta^*, c^*$ of 10 independent runs over random initialization and data reshuffling. One interesting observation is that by increasing the width and depth of ResNet18, the SGD path of Wide-ResNet-16-8 has a smaller Uniform-LGI exponent $\theta^*$ that is close to $1/2$. This is parallel to the phenomenon in Liu et al. (2020) that overparameterized neural networks satisfy the PL condition. More discussion for the Uniform-LGI on the overparameterized networks are in Appendix E.5.

### 2.3 Optimization and Generalization Results Under the Uniform-LGI Condition

In Section 2.2, empirical results suggest that the Uniform-LGI is generally satisfied on the training path for various machine learning models. In this section, we derive optimization and generalization results assuming that the Uniform-LGI holds. We consider to problem of optimizing the empirical loss (1) with gradient flow:

$$\frac{dw^{(t)}}{dt} = -\nabla \mathcal{L}_n(w^{(t)}), \ t \in [0, +\infty), \tag{3}$$

where $w^{(t)}$ is the parameter vector at time $t$, $w^{(0)}$ is the initialization. Assume $\forall (x, y) \sim \mathcal{D}, \|x\| \le 1, |y| \le 1$.

First, we give an optimization result under the Uniform-LGI condition. We show that when $\theta = 1/2$, the convergence rate is linear; when $\theta \in (1/2, 1)$, the convergence rate is sublinear. Furthermore, we give an explicit estimate for the distance between the initialization and the parameter during the training process.

**Theorem 2.2** (Optimization). *For a fixed initialization $w^{(0)}$, suppose that there exist $\theta_n \in [1/2, 1)$ and $c_n > 0$ such that the loss function $\mathcal{L}_n(w)$ satisfies the Uniform-LGI on $\{w^{(t)} : t \ge 0\}$ with $\theta_n, c_n$. Then $w^{(t)}$ converges to a stationary point $w^{(\infty)}$ with convergence rate given by*

$$\theta_n = 1/2: \quad \mathcal{L}_n(w^{(t)}) - \mathcal{L}_n(w^{(\infty)}) \le e^{-c_n^2 t} \left( \mathcal{L}_n(w^{(0)}) - \mathcal{L}_n(w^{(\infty)}) \right);$$

$$\theta_n \in (1/2, 1): \quad \mathcal{L}_n(w^{(t)}) - \mathcal{L}_n(w^{(\infty)}) \le (1 + Mt)^{-1/(2\theta_n - 1)} \left( \mathcal{L}_n(w^{(0)}) - \mathcal{L}_n(w^{(\infty)}) \right),$$

*where $M = c_n^2 (2\theta_n - 1) \left( \mathcal{L}_n(w^{(0)}) - \mathcal{L}_n(w^{(\infty)}) \right)^{2\theta_n - 1}$. The distance between the initialization $w^{(0)}$ and the parameter $w^{(t)}$ at time $t$ is bounded by*

$$\|w^{(0)} - w^{(t)}\| \le \frac{1}{c_n(1 - \theta_n)} \left[ \left( \mathcal{L}_n(w^{(0)}) - \mathcal{L}_n(w^{(\infty)}) \right)^{1-\theta_n} - \left( \mathcal{L}_n(w^{(t)}) - \mathcal{L}_n(w^{(\infty)}) \right)^{1-\theta_n} \right].$$

The proof of Theorem 2.2 is given in Appendix B. This theorem shows that if the loss function satisfies the Uniform-LGI along the gradient flow path, then it converges to a stationary point with an explicit convergence rate and distance estimate. The estimation of $c_n, \theta_n$ for different $n$ should be analyzed case by case based on the loss function, as shown in Section 3. Our optimization result can also be extended to the standard gradient descent algorithm. See Appendix C for details.

Once we have a distance estimate for the parameter during the training process, we can derive generalization bounds for the norm-constrained parameter space during the training process based on the Rademacher complexity theory. For the generalization error analysis, we assume that there exists an almost everywhere differentiable function $\Psi : \mathbb{R}^{p+q} \to \mathbb{R}$ such that the model $f(w, \cdot)$ can be represented in the following form,

$$\forall x \in \mathbb{R}^d, \ f(w, x) = \Psi \left( \alpha_1^\top x, \dots, \alpha_p^\top x, \beta_1, \dots, \beta_q \right), \tag{4}$$

where $\alpha_1, \dots, \alpha_p \in \mathbb{R}^d$, $\beta_1, \dots, \beta_q \in \mathbb{R}$, and $w = \text{vec} \left( \{\alpha_1, \dots, \alpha_p, \beta_1, \dots, \beta_q\} \right) \in \mathbb{R}^{pd+q}$. vec is the vectorization that concatenates all elements into a column vector. A wide class of functions can be represented in the form (4), including linear functions, fully connected neural networks and convolutional neural networks.

**Additional notations.** For the loss function $\ell$, we use $L_\ell(\mathcal{S})$ to denote its Lipschitz constant (the maximal gradient norm) on $\mathcal{S}$ with respect to its first argument. For $\Psi$ in (4), we define $L_\Psi(\mathcal{S}) := \left( L_\Psi^{(1)}(\mathcal{S}), \cdots, L_\Psi^{(p)}(\mathcal{S}), L_\Psi^{(p+1)}(\mathcal{S}), \cdots, L_\Psi^{(p+q)}(\mathcal{S}) \right)^\top$, where $L_\Psi^{(i)}(\mathcal{S})$ is the Lipschitz constant of $\Psi$ on $\mathcal{S}$ with respect to the $i$-th variable. Let $w^{(0)} := \text{vec} \left( \{\alpha_1^{(0)}, \dots, \alpha_p^{(0)}, \beta_1^{(0)}, \dots, \beta_q^{(0)}\} \right)$ and use $\mathcal{L}_\mathcal{D}(w)$ to denote the expected loss $\mathbb{E}_{(x,y) \sim \mathcal{D}} [\ell(f(w, x), y)]$. For $a = (a_1, \dots, a_p)^\top$ and $b = (b_1, \dots, b_q)^\top$, we define $\mathcal{S}_{a,b} = \{w : \forall i \in [p], j \in [q], \|\alpha_i\| \le a_i, |\beta_j| \le b_j\}$, and $M_{a,b} = \sup_{w \in \mathcal{S}_{a,b}, \|x\| \le 1, |y| \le 1} \ell(f(w, x), y)$. Note that for any loss function $\ell : \mathbb{R} \times \mathbb{R} \to [0, 1]$ that is 1-Lipschitz in the first argument, we have that $L_\ell(\mathcal{S}_{a,b}) = M_{a,b} = 1$ for any $a, b$. An example for such a loss function is the ramp loss (Huang et al., 2014) that is commonly used for classification.

In the next theorem, to simplify the notation, we consider studying the generalization of $w_\varepsilon$ when the loss $\mathcal{L}_n(w_\varepsilon)$ first reaches $\varepsilon \cdot \mathcal{L}_n(w^{(0)})$ for some given $\varepsilon \in [0, 1]$. Then by Theorem 2.2, the distance $\|w^{(0)} - w_\varepsilon\|$ is bounded above by $\frac{\left(\mathcal{L}_n(w^{(0)}) - \mathcal{L}_n(w^{(\infty)})\right)^{1-\theta_n} - \left(\varepsilon \mathcal{L}_n(w^{(0)}) - \mathcal{L}_n(w^{(\infty)})\right)^{1-\theta_n}}{c_n(1-\theta_n)}$. Finally by estimating the initial loss value and the final loss value, we can use the distance estimate to get a generalization bound. To showcase a bias-variance tradeoff, we bound the test error $\mathcal{L}_\mathcal{D}(w_\varepsilon)$ rather than the generalization gap $\mathcal{L}_\mathcal{D}(w_\varepsilon) - \mathcal{L}_n(w_\varepsilon)$.

**Theorem 2.3** (Generalization). *For a fixed initialization $w^{(0)}$, suppose that there exist $\theta_n \in [1/2, 1)$ and $c_n > 0$ such that the loss function $\mathcal{L}_n(w)$ satisfies the Uniform-LGI on $\{w^{(t)} : t \geq 0\}$ with $\theta_n, c_n$, and assume that there exist $M_\delta, \bar{M}_\delta$ such that with probability at least $1 - \delta/2$ over the training samples $S$, $\mathcal{L}_n(w^{(0)}) \leq M_\delta$, $\mathcal{L}_n(w^{(\infty)}) \leq \bar{M}_\delta$. Then for any given $\varepsilon, \delta \in [0, 1]$, with probability at least $1 - \delta$ over $S$, the generalization bound of any parameter $w_\varepsilon$ with $\mathcal{L}_n(w_\varepsilon) = \varepsilon \mathcal{L}_n(w^{(0)})$ is given by*

$$\mathcal{L}_\mathcal{D}(w_\varepsilon) \leq \varepsilon \mathcal{L}_n(w^{(0)}) + \sup_{\|a\|^2 + \|b\|^2 \leq 2r_{n,\delta,\varepsilon}^2} \frac{2\sqrt{2} r_{n,\delta,\varepsilon} L_\ell(\mathcal{S}_{a,b}) \|L_\Psi(\mathcal{S}_{a,b})\|}{\sqrt{n}} + 3M_{a,b}\sqrt{\frac{3(p+q) + \log(4/\delta)}{2n}}, \quad (5)$$

*where $r_{n,\delta,\varepsilon} = \frac{\left(M_\delta - \bar{M}_\delta\right)^{1-\theta_n} - \left(\varepsilon M_\delta - \bar{M}_\delta\right)^{1-\theta_n}}{c_n(1-\theta_n)}$.*

The proof of Theorem 2.3 is in Appendix D. We use the distance estimate in Theorem 2.2 to get a norm-based parameter space $\{w : \|w - w^{(0)}\| \leq r\}$. Then this allows us to use Rademacher complexity theory to obtain the generalization result. There are several key terms in (5): (1) $M_\delta$: This is a high probability upper bound for loss value at initialization. In practice, for commonly used initialization schemes, such as Xavier initialization (Glorot & Bengio, 2010) and Kaiming initialization (He et al., 2015), $\mathcal{L}_n(w^{(0)})$ is uniformly bounded with high probability; (2) $\theta_n, c_n$: These two quantities are the Uniform-LGI constants along the optimization path. The asymptotic analysis of $\theta_n, c_n$ is model-dependent, and we provide several examples in Section 3; (3) $L_\ell(\mathcal{S}_{a,b}), M_{a,b}$: These quantities are directly related to the loss function $\ell$. For instance, given a loss function $\ell : \mathbb{R} \times \mathbb{R} \to [0, 1]$ that is 1-Lipschitz in the first argument, we have that $L_\ell(\mathcal{S}_{a,b}) = M_{a,b} = 1$; (4) $\|L_\Psi(\mathcal{S}_{a,b})\|$: This term is related to the properties of $f$. For example, when $f$ is linear, $\Psi(x, y) = x + y$, then $\|L_\Psi(\mathcal{S}_{a,b})\| = \sqrt{2}$. When $f$ is a two-layer neural network, i.e., $f(w, x) = \sum_{i=1}^m v_i \phi(u_i^\top x)$, we will show in Section 3.3 that $\sup_{\|a\|^2 + \|b\|^2 \leq 2r^2} \|L_\Psi(\mathcal{S}_{a,b})\|$ is bounded above by $\sqrt{2}r$ (equation (29)). For two-layer neural networks, $p$ and $q$ are both equal to the number of the hidden units, which is usually much smaller than the sample size $n$. In this case, the bound is dominated by the first two terms, which represent a bias-variance tradeoff pattern during training. To see this, if $\varepsilon = 1$ (no training, high bias, low variance), then $r_{n,\delta,\varepsilon} = 0$, and the bound is dominated by the initial loss; if $\varepsilon = \bar{M}_\delta/M_\delta$ (convergence model, low bias, high variance), then the generalization bound is dominated by the model complexity $r_{n,\delta,\varepsilon}$.

*Remark* 2.4. The generalization bound also holds for the converged model by simply setting $\varepsilon$ to be $\bar{M}_\delta/M_\delta$. In particular, when the final empirical loss value is 0, $\varepsilon$ can be 0. For the gradient descent algorithm, one can also directly use the optimization result (the distance bound) in Appendix C to derive the generalization result. The only quantity that needs to be changed is that $r_{n,\delta,\varepsilon} = \frac{\left(M_\delta - \bar{M}_\delta\right)^{1-\theta_n} - \left(\varepsilon M_\delta - \bar{M}_\delta\right)^{1-\theta_n}}{c_n(1-\theta_n)(1-\eta\beta_\mathcal{L}/2)}$, where $\eta$ is the learning rate, $\beta_\mathcal{L}$ is the smoothness constant. See Appendix C for details.

## 3 Applications

In this section, we apply our framework (Theorem 2.2 and Theorem 2.3) to three machine learning models. To obtain clean expressions of the generalization bound in terms of the sample size $n$, we consider a range of $n$ related to the dimension $d$. In particular, we consider underdetermined systems where the ratio $n/d$ remains finite unless stated otherwise:

$$\exists\, \gamma_0, \gamma_1 \in (0, \infty), \text{ s.t. } \forall d, \ \gamma_0 d \leq n = n(d) \leq \gamma_1 d.$$

This setting is non-asymptotic since we do not require $d$ to be sufficiently large. For each application, we consider regression problems with squared loss $\ell(y, \hat{y}) = (y - \hat{y})^2/2$, which is not *globally* Lipschitz nor globally bounded. Thus, we cannot apply directly Theorem 2.3 to the squared loss, since it requires to bound $L_\ell(S_{a,b})$ and $M_{a,b}$ which depend on $a, b$ and the model architecture. To mitigate this issue, we consider to

evaluate the generalization with a new loss $\tilde{\ell}$ that is globally Lipschitz and globally bounded. Specifically, let $\tilde{\ell} : \mathbb{R} \times \mathbb{R} \to [0, l_0]$ that is $\sqrt{l_0}$-Lipschitz (on the first argument) for some $l_0 > 0$ and $\tilde{\ell}(y, y) = 0$. For the squared loss, this can be achieved by truncating $\ell$ at $|y - \hat{y}| = \sqrt{l_0}$, and then using a constant extension past the truncated point to make it continuous. In this case, $L_\ell(S_{a,b}) = \sqrt{l_0}$, $M_{a,b} = l_0$ for any $a, b$. Finally, to use Theorem 2.3, it remains to estimate the new empirical loss value $\frac{1}{n} \sum_{i=1}^n \tilde{\ell}(f(w_\varepsilon, x_i), y_i)$ in terms of $\mathcal{L}_n(w_\varepsilon)$. By the Lipschitzness property and Cauchy–Schwarz inequality, one can show that $\frac{1}{n} \sum_{i=1}^n \tilde{\ell}(f(w_\varepsilon, x_i), y_i) \le \sqrt{2l_0 \mathcal{L}_n(w_\varepsilon)}$ (see the inequality (22) in Appendix E.1). Combining the above derivations, applications of the framework are straightforward.

*Remark* 3.1. One can also get generalization bounds under the squared loss $\ell(y, \hat{y}) = (y - \hat{y})^2/2$. Note that $M_{a,b} \le \sup_{w \in S_{a,b}, \|x\| \le 1} f(w, x)^2 + 1$ and $L_\ell(S_{a,b}) \le \sup_{w \in S_{a,b}, \|x\| \le 1} |f(w, x)| + 1$. Then we can get an upper bound for $\mathcal{L}_\mathcal{D}(w_\varepsilon)$ by Theorem 2.3.

### 3.1 Underdetermined $\ell_2$ Linear Regression

We begin with an underdetermined linear regression model $f(w, x) = w^\top x$ with squared loss:

$$\arg\min_{w \in \mathbb{R}^d} \mathcal{L}_n(w) := \frac{1}{2n} \sum_{i=1}^n \left( w^\top x_i - y_i \right)^2, \tag{6}$$

where the input data matrix $\mathcal{X} = (x_1, \ldots, x_n)^\top \in \mathbb{R}^{n \times d}$ has full row rank $(d > n)$. Then the above regression model has at least one global minimum with zero loss.

**Target function.** Suppose the training data is generated from an underlying function $g : \mathbb{R}^d \to \mathbb{R}$ with $y_i = g(x_i)$, $\forall i \in [n]$. Let $\mathcal{Y} = (y_1, \cdots, y_n)^\top$, and assume that there exits $c^* > 0$ such that

$$\|\mathcal{Y}\| \le c^* \sqrt{\lambda_{\max}(\mathcal{X}\mathcal{X}^\top)}. \tag{7}$$

(7) actually indicates that $g$ is Lipschitz with a dimension independent Lipschitz constant. For example, $g(x) = \phi(x^\top w^*)$, where $w^* \in \mathbb{R}^d$ with $\|w^*\|_2 \le c^*$ for some constant $c^*$, and $\phi(\cdot)$ is Lipschitz with $\phi(0) = 0$.

**Assumption 3.2.** There exists symmetric positive-definite matrices $\Sigma_d \in \mathbb{R}^{d \times d}$ with $0 < \lambda_0 \le \lambda_{\min}(\Sigma_d^2) \le \lambda_{\max}(\Sigma_d^2) \le \lambda_1$ for any $d$, such that the entries of $\mathcal{X}\Sigma_d$ are i.i.d. subgaussian random variables with zero mean and subgaussian moments[2] bounded by 1.

This assumption is reasonable in practice in the sense that it only requires the data (rows of the data matrix $\mathcal{X}$) to be i.i.d. and entries of each row of $\mathcal{X}$ (feature vector components) are not necessarily independent.

By applying our framework, we get the following optimization and generalization results for the models during training when the empirical loss function $\mathcal{L}_n(w_\varepsilon) = \varepsilon \mathcal{L}_n(w^{(0)})$.

**Theorem 3.3.** *Consider the undertermined $\ell_2$ linear regression model (6). Suppose that there exists a constant $c_0 \ge 1$ that is independent of $d$ such that $\left\| w^{(0)} \right\|_2 \le c_0$. Then the followings hold:*

1. *$\mathcal{L}_n(w)$ satisfies Uniform-LGI along the gradient flow curve $\left\{ w^{(t)} : t \ge 0 \right\}$ with $c_n = \sqrt{\frac{2\lambda_{\min}(\mathcal{X}\mathcal{X}^\top)}{n}}$, $\theta_n = 1/2$.*

2. *$\mathcal{L}_n(w^{(t)})$ converges to zero linearly, i.e., $\mathcal{L}_n(w^{(t)}) \le \exp\left( -2\lambda_{\min}(\mathcal{X}\mathcal{X}^\top) t/n \right) \mathcal{L}_n(w^{(0)})$.*

3. *If $\gamma_1 \in (0, 1)$, then under Assumption 3.2, for any $\varepsilon \ge 0$ and any target function that satisfies (7), with probability at least $1 - \delta - \tau^{d-n+1} - \tau^d$ over the training samples $S$, the generalization bound of the parameter $w_\varepsilon$ with $\mathcal{L}_n(w_\varepsilon) = \varepsilon \mathcal{L}_n(w^{(0)})$ for some $\varepsilon \in [0, 1]$ is given by*

$$\mathbb{E}_{(x,y) \sim \mathcal{D}} \left[ \tilde{\ell}(f(w_\varepsilon, x), y) \right] \le \sqrt{2l_0 \varepsilon \mathcal{L}_n(w^{(0)})} + \frac{4(c_0 + c^*)\sqrt{\frac{2(1-\varepsilon)\lambda_1}{\lambda_0}} \left( C + \frac{C}{\sqrt{\gamma_0}} + \sqrt{\frac{\log(4/\delta)}{cn}} \right)}{\frac{\tau}{C_1}\left( \frac{1}{\sqrt{\gamma_1}} - 1 \right)\sqrt{n}} + 3l_0 \sqrt{\frac{3 + \log(4/\delta)}{2n}},$$

*where $c, C, C_1 > 0$ and $\tau \in (0, 1)$ depend only on the subgaussian moment of the entries.*

---

[2]The subgaussian moment of $X$ is defined as $\inf \left\{ \mathcal{M} \ge 0 \mid \mathbb{E}e^{tX} \le e^{\mathcal{M}^2 t^2/2}, \ \forall t \in \mathbb{R} \right\}$.

The proof of Theorem 3.3 is given in Appendix E.1. The generalization bound reveals a bias-variance tradeoff for the linear regression model in the high dimension setting. For $\varepsilon = 1$, the generalization bound is dominated by the initial loss. For $\varepsilon = 0$ (convergence model), we get a generalization bound $\mathcal{O}\left(\sqrt{\log(1/\delta)/n}\right)$.

**Comparison.** This result is related to Bartlett et al. (2020) that studied the phenomenon of benign overfitting in high-dimensional $\ell_2$ linear regression. First, we summarize the different settings and assumptions in our paper and Bartlett et al. (2020) through an example: the training data $\{(x_i, y_i)\}_{i=1}^n \subset \mathbb{R}^d \times \mathbb{R}$ are i.i.d. drawn from $\mathcal{D}$. In our case, for any $(x, y) \in \mathcal{D}$, $y = x^\top w^*$ for some $w^* \in \mathbb{R}^d$ with uniformly bounded norm in $d$, and entries of $x$ are i.i.d. (thus $\lambda_0 = \lambda_1 = 1$) with mean 0, variance $\Sigma = \mathbb{E}[xx^\top]$ (diagonal). Thus $\Sigma$ has a flat spectrum with condition number 1. We further assume that $\|x\| \leq 1$ for all $d$. Let $\mathcal{X} = (x_1, \ldots, x_n)^\top \in \mathbb{R}^{n \times d}, \mathcal{Y} = (y_1, \ldots, y_n) \in \mathbb{R}^n$, then the minimum norm solution $\hat{w} = \mathcal{X}^\dagger \mathcal{Y}$ ($\dagger$ is the pseudo inverse). Let the excess risk $E = \mathbb{E}_x[(x^\top \hat{w} - x^\top w^*)^2]$, then our result states that $E = \mathcal{O}(1/\sqrt{n})$ given that $d$ and $n$ diverge but their ratio remains finite. Now we consider a new data set $\{(\tilde{x}_i, \tilde{y}_i)\}_{i=1}^n \subset \mathbb{R}^d \times \mathbb{R}$ that are i.i.d. drawn from $\tilde{\mathcal{D}}$. In Bartlett et al. (2020), the setting is that for any $(\tilde{x}, \tilde{y}) \in \tilde{\mathcal{D}}, \tilde{y} = \tilde{x}^\top \tilde{w}^*$ for some $\tilde{w}^* \in \mathbb{R}^d$ with uniformly bounded norm in $d$, and entries of $\tilde{x}$ are independent with mean 0, variance $\tilde{\Sigma} = \mathbb{E}[\tilde{x}\tilde{x}^\top]$. If the entries of $\tilde{x}$ are independent, then $\tilde{\Sigma}$ is diagonal. Let the minimum norm solution w.r.t. the new data set $\tilde{w} = \tilde{\mathcal{X}}^\dagger \tilde{\mathcal{Y}}$, then it is shown in Bartlett et al. (2020) that when $\tilde{\Sigma}$ has a suitable decaying eigenspectrum, the excess risk $\tilde{E} = \mathbb{E}_{\tilde{x}}[(\tilde{x}^\top \tilde{w} - \tilde{x}^\top \tilde{w}^*)^2] \to 0$ when $n \to \infty$. Therefore, we can see that there exist two cases for benign overfitting: the data covariance matrix has a decaying eigenspectrum (Bartlett et al., 2020); the data covariance matrix has a flat eigenspectrum but each data vector has a bounded norm (Theorem 3.3). Due to the rescaling of the data vector, even though $\Sigma$ has a flat eigenspectrum, the benign overfitting provably happens when $\hat{w}$ has a bounded norm. To have a better understanding of the benign overfitting phenomenon when the data covariance matrix has a flat eigenspectrum, we consider an example in Bartlett et al. (2020) that we can deduce from our result. Under the same notation as above, we consider the following data transformation for any $(x, y) \in \mathcal{D}$: $\tilde{x} = \tilde{\Sigma}^{1/2} x \sqrt{d}$, $\tilde{w}^* = \frac{\tilde{\Sigma}^{-1/2} w^*/\sqrt{d}}{\|\tilde{\Sigma}^{-1/2} w^*/\sqrt{d}\|}, \tilde{y} = \frac{y}{\|\tilde{\Sigma}^{-1/2} w^*/\sqrt{d}\|}$. Then the induced new data distribution $\tilde{\mathcal{D}}$ satisfies the assumptions and settings in Bartlett et al. (2020). Note that $\tilde{\mathcal{X}} = \mathcal{X} \tilde{\Sigma}^{1/2} \sqrt{d}, \tilde{\mathcal{Y}} = \frac{\mathcal{Y}}{\|\tilde{\Sigma}^{-1/2} w^*/\sqrt{d}\|}$, thus the minimum norm solution $\tilde{w} = \tilde{\mathcal{X}}^\dagger \tilde{\mathcal{Y}} = \frac{\tilde{\Sigma}^{-1/2} \hat{w}}{\|\tilde{\Sigma}^{-1/2} w^*\|}$. Therefore, the excess risk $\tilde{E} = \mathbb{E}_{\tilde{x}}[(\tilde{x}^\top \tilde{w} - \tilde{x}^\top \tilde{w}^*)^2] = \frac{E}{\|\tilde{\Sigma}^{-1/2} w^*/\sqrt{d}\|^2}$. We can see that there exists an explicit relation between $\tilde{E}$ and $E$. In the high dimension setting ($d$ and $n$ diverge but have a finite ratio), our result shows that $E = \mathcal{O}(1/\sqrt{d})$. We consider a benign overfitting example in (Bartlett et al., 2020, Part 1 of Theorem 6): the eigenvalues of $\tilde{\Sigma}$ decay with a rate given by $\sigma_j = j^{-1} \log^{-\beta}(j+1)$ with $\beta > 1$. Now we show that $\tilde{E} \to 0$ when entries of $w^*$ are order $1/\sqrt{d}$. Notice that $\frac{E}{\|\tilde{\Sigma}^{-1/2} w^*/\sqrt{d}\|^2} \sim \frac{d\sqrt{d}}{\sigma_1^{-1} + \ldots + \sigma_d^{-1}} = \frac{d\sqrt{d}}{\sum_{j=1}^d j \log^\beta(j+1)} < \frac{d\sqrt{d}}{\sum_{j=2}^d j} \to 0$. In summary, we can transform our assumptions and settings to analyze the case of sharp eigenspectrum for benign overfitting, and get a result that is consistent with Bartlett et al. (2020).

## 3.2 Kernel Regression

Consider a positive definite kernel $k : \mathcal{X} \times \mathcal{X} \to \mathbb{R}$ with a corresponding feature map $\varphi : \mathbb{R}^d \to \mathcal{F}$ satisfying $\langle \varphi(x), \varphi(y) \rangle_\mathcal{F} = k(x, y)$. We assume that $|k(x, x)| \leq 1, \forall x \in \mathcal{X}$. Let $\mathcal{H}$ be the reproducing kernel Hilbert space (RKHS) with respect to $k$. If $\mathcal{F} = \mathbb{R}^s$, then the kernel regression model with $\ell_2$ loss is to solve the following problem

$$\arg\min_{w \in \mathbb{R}^s} \mathcal{L}_n(w) := \frac{1}{2n} \sum_{i=1}^n \left( w^\top \varphi(x_i) - y_i \right)^2. \tag{8}$$

Similar to the $\ell_2$ linear regression case, we consider the following target function:

**Target function.** Suppose the training data is generated by an underlying function $g : \mathbb{R}^d \to \mathbb{R}$ with $y_i = g(x_i), \forall i \in [n]$. We further assume that there exists $c^* > 0$ such that

$$\|\mathcal{Y}\| \leq c^* \cdot \sqrt{\lambda_{\max}(k(\mathcal{X}, \mathcal{X}))}, \tag{9}$$

where $k(\mathcal{X}, \mathcal{X})$ is the $n \times n$ kernel matrix with $k(\mathcal{X}, \mathcal{X})_{ij} = k(x_i, x_j)$.

For instance, the following functions satisfy (9): $g(x) = \phi(\varphi(x)^\top w^*)$ where $w^* \in \mathbb{R}^s$ with $(\forall s) \|w^*\|_2 \leq c^*$ for some constant $c^*$, and $\phi(\cdot)$ is Lipschitz with $\phi(0) = 0$. To get the generalization results of kernel regression, we will discuss two types of kernels separately: radial basis function (RBF) (Broomhead & Lowe, 1988) kernel and inner product kernel.

**RBF kernel.** We study the RBF kernel of the form $k(x, y) = \varrho(\|y - x\|)$ for a certain RBF $\varrho$. For the input data, we define the *separation distance* of $\mathcal{X}$ as $\mathsf{SD} := \frac{1}{2} \min_{i \neq j} \|x_i - x_j\|, \forall i, j \in [n]$.

**Inner product kernel.** We consider $k(x, y) = \varrho\left(\frac{x^\top \Sigma_d^2 y}{d}\right)$, where $\Sigma_d^2$ is defined in Assumption 3.2.

Following El Karoui et al. (2010), we make the following assumption on the function $\varrho$:

**Assumption 3.4.** $\varrho$ is $C^3$ in a neighborhood of 0 with $\varrho(0) = 0$, $\varrho(1) > \varrho'(0) \geq 0$, $\varrho''(0) \geq 0$.

We now apply our framework to get optimization and generalization results for the final convergence model when $\varepsilon = 0$. For the RBF kernel, the generalization bound depends on the separation distance of the samples. For the inner product kernel, we study the spectrum property of the high-dimensional random kernel matrix.

**Theorem 3.5.** *Consider the kernel regression model (8). Suppose that there exists a constant $c_0 \geq 1$ that is independent of $d$ such that $\left\|w^{(0)}\right\|_2 \leq c_0$. Then the followings hold:*

1. *$\mathcal{L}_n(w)$ satisfies Uniform-LGI along the gradient flow curve $\left\{w^{(t)} : t \geq 0\right\}$ with $c_n = \sqrt{\frac{2\lambda_{\min}(k(\mathcal{X}, \mathcal{X}))}{n}}$, $\theta_n = 1/2$, where $c_n$ is controlled by the kernel and input samples.*

2. *$\mathcal{L}_n(w^{(t)})$ converges to zero linearly, i.e., $\mathcal{L}_n(w^{(t)}) \leq \exp\left(-2\lambda_{\min}(k(\mathcal{X}, \mathcal{X}))t/n\right)\mathcal{L}_n(w^{(0)})$.*

3. *For any target function that satisfies (9) we have:*

   - *For the RBF kernel[3], suppose that $\varrho : \mathbb{R}_{\geq 0} \to \mathbb{R}_{\geq 0}$ is a decreasing function and $\varrho(\|x\|) \in L^1(\mathbb{R}^d)$. If there exists two positive constants $q_{\min}$ and $q_{\max}$ such that $\mathsf{SD} \in [q_{\min}, q_{\max}]$ for all $n$, then with probability at least $1 - \delta$ over the training samples, the generalization bound of the parameter $w_\varepsilon$ with $\mathcal{L}_n(w_\varepsilon) = \varepsilon \mathcal{L}_n(w^{(0)})$ for some $\varepsilon \in [0, 1]$ is given by*

     $$\mathbb{E}_{(x,y)\sim\mathcal{D}}\left[\tilde{\ell}\left(f(w_\varepsilon, x), y\right)\right] \leq \sqrt{2l_0 \varepsilon \mathcal{L}_n(w^{(0)})} + \frac{C(c_0, c^*, \varrho, d, q_{\min}, q_{\max})\sqrt{1-\varepsilon}}{\sqrt{n}} + 3l_0\sqrt{\frac{3 + \log(4/\delta)}{2n}},$$

     *where $C(c_0, c^*, \varrho, d, q_{\min}, q_{\max})$ is a constant that depends only on $c_0, c^*, \varrho, d, q_{\min}, q_{\max}$.*
   - *For the inner product kernel, under Assumption 3.2 & 3.4, if $d$ is large enough and $\delta > 0$ is small enough such that $d^{-1/2}\left(\sqrt{3}\delta^{-1/2} + \log^{0.51} d\right) \leq 0.5(\varrho(1) - \varrho'(0))$, then with probability at least $1 - \delta - d^{-2}$ over the samples, the generalization bound of the parameter $w_\varepsilon$ with $\mathcal{L}_n(w_\varepsilon) = \varepsilon \mathcal{L}_n(w^{(0)})$ for some $\varepsilon \in [0, 1]$ is given by*

     $$\mathbb{E}_{(x,y)\sim\mathcal{D}}\left[\tilde{\ell}\left(f(w_\varepsilon, x), y\right)\right] \leq \sqrt{2l_0 \varepsilon \mathcal{L}_n(w^{(0)})} + \frac{C\left(1 + \sqrt{\log(1/\delta)/n}\right)\sqrt{1-\varepsilon}}{\sqrt{n}} + 3l_0\sqrt{\frac{3 + \log(4/\delta)}{2n}},$$

*where $C$ depends on $c_0, c^*, \gamma_1, \varrho''(0), \varrho'(0), \varrho(1)$ and the subgaussian moment of the entries.*

The proof of Theorem 3.5 is given in Appendix E.2.

*Example* 1. Kernels satisfying the conditions and assumptions in Theorem 3.5 include (1) RBF Gaussian: $\varrho(r) = e^{-\rho r^2}, \rho > 0$; (2) RBF Multiquadrics: $\varrho(r) = (\rho + r^2)^{\beta/2}, \rho > 0, \beta \in \mathbb{R}\backslash 2\mathbb{N}, \beta < -d$; (3) Inner product Polynomial kernel: $\varrho(r) = r^\beta, \beta \in \mathbb{Z}^+, \beta \geq 2$; (4) NTK corresponding to Two-layer ReLU neural networks on $\mathbb{S}^{d-1}(\sqrt{d})$: $\varrho(r) = \frac{r(\pi - \arccos(r))}{2\pi}$.

**Comparison.** The result of the inner product kernel is related to Liang & Rakhlin (2020) who derived generalization bounds for the minimum RKHS norm estimator. They showed that when the data covariance

---

[3]Here $d$ is fixed and $n$ is varied.

matrix and the kernel matrix enjoy certain decay of the eigenvalues, the generalization bound vanishes as $n$ goes to infinity. For example, for exponential kernel and the covariance matrix $\Sigma := \mathbb{E}[xx^\top]$ with the $j$-th eigenvalue $\lambda_j(\Sigma) = j^{-\alpha}$, the $\ell_2$ generalization bound becomes $\mathcal{O}(n^{-\frac{\alpha}{2\alpha+1}})$ when $\alpha \in (0,1)$ and $n > d$. In comparison, we do not assume the eigenvalue decay property for the covariance matrix and still are able to obtain an optimal generalization bound $\mathcal{O}\left(n^{-1/2}\right)$ in the high dimension setting. Further, we extend the works in Liang & Rakhlin (2020) by proving a new result of the RBF kernel. Note that the result of the RBF kernel is not under the high-dimensional setting; thus it is not a direct adaptation of Liang & Rakhlin (2020), and the proof itself is of independent interest.

### 3.3 Two-layer Neural Networks

In this section, we show that our framework can be applied to neural network models. We obtain generalization bounds for shallow neural networks under the Uniform-LGI assumption. First, define a two-layer ReLU neural network with width $m$:

$$f(w,x) := v^\top \phi(Ux) = \sum_{i=1}^{m} v_i \phi(u_i^\top x), \tag{10}$$

where $\phi(x) = \max\{0,x\}$, $x \in \mathbb{R}^d$ is the input, $v = (v_1,\ldots,v_m)^\top \in \mathbb{R}^m$, $U = (u_1,\ldots,u_m)^\top \in \mathbb{R}^{m\times d}$ are the parameters, $w = \text{vec}(\{v,U\})$. Trivially, there exists an almost everywhere differentiable function $\Psi : \mathbb{R}^{2m} \to \mathbb{R}$ with $\Psi(s_1,\ldots,s_m,t_1,\ldots,t_m) = \sum_{i=1}^{m} s_i \phi(t_i)$ such that (4) holds. We consider minimizing the quadratic loss $\mathcal{L}_n(w) := \frac{1}{2n}\sum_{i=1}^{n}(f(w,x_i)-y_i)^2$ by gradient flow (3) under a fixed initialization $w^{(0)}$.

The numerical observations in Figure 2 show that the Uniform-LGI condition is generally satisfied on *each* optimization path when training the neural network models (including the two-layer neural networks). In the next proposition, we theoretically prove that there exist $c_n > 0, \theta_n \in [1/2,1)$ such that the Uniform-LGI holds for the gradient flow path over *each* choice of the training samples.

**Proposition 3.6.** *For a fixed initialization $w^{(0)}$ and a fixed choice of the training sample $S_n$ with size $n$, there exist two constants $c_n(S_n) > 0, \theta_n(S_n) \in [1/2,1)$ such that the loss function $\mathcal{L}_n(w)$ satisfies the Uniform-LGI along the gradient flow curve $\{w^{(t)} : t \geq 0\}$ with $c_n(S_n), \theta_n(S_n)$. Specifically, for the population loss $\mathcal{L}_\mathcal{D}(w)$ and its induced gradient flow curve $\{w_\mathcal{D}^{(t)} : t \geq 0\}$, if $\mathcal{L}_\mathcal{D}(w)$ is subanalytic (Bolte et al., 2007), then there exist $c_\mathcal{D} > 0, \theta_\mathcal{D} \in [1/2,1)$ such that $\mathcal{L}_\mathcal{D}(w)$ satisfies the Uniform-LGI along $\{w_\mathcal{D}^{(t)} : t \geq 0\}$ with $c_\mathcal{D}, \theta_\mathcal{D}$.*

The proof can be found in Appendix E.3. Note that the Uniform-LGI constants $c_n(S_n), \theta_n(S_n)$ are random variables that depend on the choice of the training sample $S_n$. Based on the limiting values $(c_\mathcal{D}, \theta_\mathcal{D})$, we make the following assumption that $c_n(S_n) > 0, \theta_n(S_n) \in [1/2,1)$ hold uniformly with high probability over the choice of $S_n$.

**Assumption 3.7.** *For the Uniform-LGI constants in Proposition 3.6 and any $\delta \in (0,1)$, there exists $c_{n,\delta} > 0$, $\theta_{n,\delta} \in [1/2,1)$ such that with probability at least $1-\delta/3$ over the sample $S_n$, $\theta_n(S_n) \leq \theta_{n,\delta}, c_n(S_n) \geq c_{n,\delta}$.*

This assumption is also supported by the statistical results in Table 1. Indeed, the numerical results suggest that the Uniform-LGI constants satisfy this assumption with high confidence over the training sample. We are now ready to state the main result for the two-layer neural network model.

**Theorem 3.8.** *Consider the two-layer neural network model (10) with initialization $w^{(0)}$. Under Assumption 3.7, for any $\delta \in (0,1)$ we have the following:*

- *With probability at least $1-\delta/3$ over the training sample, the convergence rate is given by*

$$\theta_{n,\delta} = 1/2: \quad \mathcal{L}_n(w^{(t)}) - \mathcal{L}_n(w^{(\infty)}) \leq e^{-c_{n,\delta}^2 t}(\mathcal{L}_n(w^{(0)}) - \mathcal{L}_n(w^{(\infty)}));$$
$$\theta_{n,\delta} \in (1/2,1): \quad \mathcal{L}_n(w^{(t)}) - \mathcal{L}_n(w^{(\infty)}) \leq (1+Mt)^{-1/(2\theta_n-1)}(\mathcal{L}_n(w^{(0)}) - \mathcal{L}_n(w^{(\infty)})),$$

*where $M = c_{n,\delta}^2(2\theta_{n,\delta}-1)\left(\mathcal{L}_n(w^{(0)})\right)^{2\theta_{n,\delta}-1}$.*

- *Under Assumption 3.2, assume that there exist $\bar{M}_\delta$ such that with probability at least $1-\delta/2$ over the training samples $S$, $\mathcal{L}_n(w^{(\infty)}) \le \bar{M}_\delta$. for any target function that satisfies (7), then with probability at least $1-\delta$ over the training samples, the generalization bound of the parameter $w_\varepsilon$ with $\mathcal{L}_n(w_\varepsilon) = \varepsilon \mathcal{L}_n(w^{(0)})$ for some $\varepsilon \in [0,1]$ is given by*

$$\mathbb{E}_{(x,y)\sim\mathcal{D}}\left[\tilde{\ell}\left(f(w_\varepsilon, x), y\right)\right] \le \sqrt{2l_0 \varepsilon \mathcal{L}_n(w^{(0)})} + \frac{4}{\sqrt{n}}\left(\frac{\left(\tilde{C}\left(1 + \log(1/\delta)/n\right)\right)^{1-\theta_{n,\delta}} - (\varepsilon M_\delta - \bar{M}_\delta)^{1-\theta_{n,\delta}}}{c_{n,\delta}(1-\theta_{n,\delta})}\right)^2 +$$

$$3l_0\sqrt{\frac{6m + \log(12/\delta)}{2n}},$$

*where $\tilde{C}$ depends only on $c^*, \gamma_0, \lambda_0, \|w^{(0)}\|$ and the subgaussian moment of the entries.*

The proof of Theorem 3.8 is provided in Appendix E.4. We can see that for shallow neural networks ($m$ much smaller than $n$), the generalization bound is dominated by the first two terms. When the gradient flow converges to a global minimum with zero training loss ($\bar{M}_\delta = 0$), then the generalization bound for the convergence model ($\varepsilon = 0$) reduces to $\widetilde{\mathcal{O}}\left(c_{n,\delta}^{-2}(1 - \theta_{n,\delta})^{-2}n^{-1/2}\right)$, where $\widetilde{\mathcal{O}}$ hide the logarithmic factor.

**Empirical evaluation.** To demonstrate that our generalization bound is non-vacuous and can effectively capture the test error, we perform experiments on the CIFAR10 dataset (first two classes). First, we get the Uniform-LGI constants $c_{n,\delta}, \theta_{n,\delta}$ by finite sample test (Algorithm 1) with varied sample size $n$. Then, we calculate the generalization bound (up to some constant) $c_{n,\delta}^{-2}(1 - \theta_{n,\delta})^{-2}n^{-1/2} + \sqrt{m/n}$. Finally, we randomly flip the label and record the Uniform-LGI constants $c_{n,\delta}, \theta_{n,\delta}$ with different ratio of label flip. See Appendix A for implementation details. From Figure 3 (a), we observe that $c_{n,\delta}, \theta_{n,\delta}$ have good dependency on the sample size $n$. Figure 3 (b) shows that the generalization bound decreases as $n$ increases, indicating that our bound is non-vacuous. As shown in Figure 3 (c), when the ratio of label flip increases, $c_{n,\delta}$ decreases and $\theta_{n,\delta}$ increases. Our results (Theorem 3.8) show that if $c_{n,\delta}$ decreases and $\theta_{n,\delta}$ increases, then we should expect worse optimization and generalization error, which also correlates with the known findings that random labels negatively affect both optimization and generalization (Zhang et al., 2016).

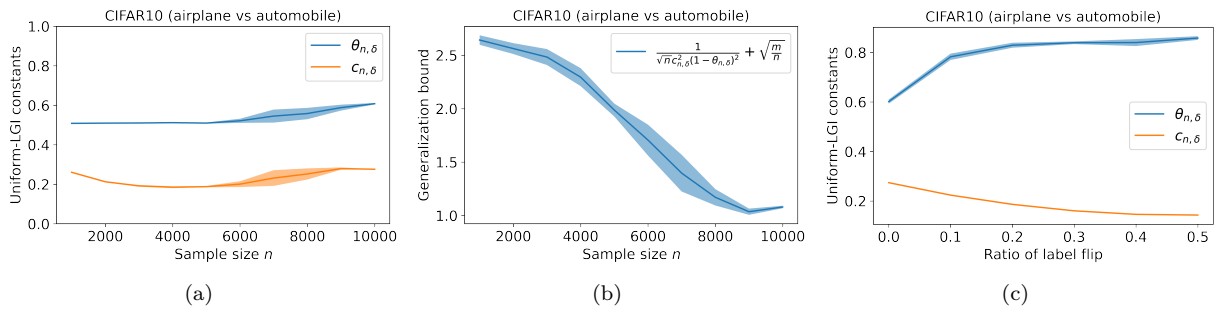

Figure 3: (a) The Uniform-LGI constants $c_{n,\delta}, \theta_{n,\delta}$ obtained by finite sample test (Algorithm 1) with respect to sample size $n$. (b) Dependence of our generalization bound $c_{n,\delta}^{-2}(1 - \theta_{n,\delta})^{-2}n^{-1/2} + \sqrt{m/n}$ on the sample size $n$. (c) The Uniform-LGI constants $c_{n,\delta}, \theta_{n,\delta}$ with respect to the ratio of label flip. All Results are reported with the mean and standard deviation over 5 independent runs.

**Overparameterized neural netowrks.** Our generalization bound becomes vacuous when $m$ is large enough. But for overparameterized neural networks, the PL condition ($\theta_{n,\delta} = 1/2$) is proved to hold in the NTK regime (Liu et al., 2020). To make our framework complete, we derive optimization and generalization results for the overparameterized two-layer neural network. The generalization result is based on the Rademacher complexity theory in Arora et al. (2019) in the NTK regime. In particular, we theoretically show that $\theta_{n,\delta} = 1/2$ and $c_{n,\delta} = \mathcal{O}(1)$ over the training sample and standard random initialization when $m$ is large enough, and we give a generalization bound of order $\mathcal{O}\left(\sqrt{\log(n/\delta)/n}\right)$, which confirms the phenomenon of benign over-fitting in the high dimension setting. More details are provided in Appendix E.5.

## 4 Related Works

**Optimization.** Theoretically analyzing the training process of most machine learning models is a challenging problem as the loss landscapes are generally non-convex. One approach to studying non-convex optimization problems is to use the Polyak-Łojasiewicz (PL) condition (Polyak, 1963), which characterizes the local geometry of loss landscapes and ensures the existence of global minima. It is shown in Karimi et al. (2016) that GD admits linear convergence for a class of optimization objective functions under the PL condition. In this paper, we modify the original LGI to obtain convergence rates (not necessarily converges to a global minimum) and generalization estimates *during* the training process.

**Generalization.** Traditional VC dimension-based generalization bounds depend on the number of parameters and are vacuous for large models such as overparameterized neural networks. To overcome this limitation, several non-vacuous generalization bounds are proposed, e.g. the norm/margin-based generalization bounds (Neyshabur et al., 2015; Bartlett et al., 2017; Golowich et al., 2018; Neyshabur et al., 2019), and the PAC-Bayes-based bounds (Dziugaite & Roy, 2017; Neyshabur et al., 2018; Zhou et al., 2019; Rivasplata et al., 2020). However, these bounds tend to ignore or focus less on optimization, e.g., norm-based generalization bounds may not discuss how small-norm solutions are obtained through practical training. In this paper, we connect the optimization and generalization by deriving generalization bounds based on the optimization path length during training model training.

**Interplay of optimization and generalization.** Implicit bias builds the bridge between optimization and generalization, which has been widely studied to explain the generalization ability of machine learning models. Recent works (Soudry et al., 2018a;b; Nacson et al., 2019a;b; Lyu & Li, 2020) showed that linear classifiers or deep neural networks trained by GD/SGD maximizes the margin of the separating hyperplanes and therefore generalizes well. Other works (Arora et al., 2019; Zou & Gu, 2019; Cao & Gu, 2020; Ji & Telgarsky, 2020; Chen et al., 2021) considered overparameterized neural networks in the lazy training regime where the minimizer has good generalization due to the low "complexity" of the parameter space. In this work, we focus on specific conditions on loss functions under which we can connect optimization and generalization on the situations which do not satisfy those above.

**Comparison with existing works on the LGI condition.** Recent work by Frei & Gu (2021) studies optimization and generalization via general PL-conditions. Their key assumptions, problem settings, and type of results differ from ours on many aspects. In Frei & Gu (2021), the authors analyze a loss function $f$ that satisfies the $(g, \xi, \alpha, \mu)$-proxy PL inequality, which is a more general condition than the LGI in this paper. For example, by setting $g = f, \xi = \min_w f(w)$, the proxy PL inequality corresponds to the LGI. However, a key assumption in Frei & Gu (2021) is that $f(w)$ satisfies a $(g, \xi, \alpha, \mu)$-proxy PL inequality *globally* for all $w \in \mathbb{R}^p$, while our assumption only requires that $f(w)$ satisfies the LGI *locally*, e.g., along the optimization path. For the problem settings, we cover different situations. Frei & Gu (2021) considers the online learning setting where samples are observed one-by-one. In contrast, we focus on the classic gradient flow/descent setting, where the training samples are given before training. Thus, these two works provide different types of results. Through the global proxy-PL inequality, Frei & Gu (2021) gives an upper bound for the *best case* of the test error, i.e., $\min_{t<T} \mathcal{L}_{\mathcal{D}}(w^{(t)})$. In this work, we obtain bounds for the test error $\mathcal{L}_{\mathcal{D}}(w^{(t)})$ for any time $t$, which can be used to showcase bias-variance trade-off patterns. For instance, a simple application of our theory yields a new case of benign overfitting on linear/kernel regression in the high dimensional setting (Theorem 3.3 & 3.5). Moreover, in this work, we propose a finite sample test algorithm that can be applied to verify the Uniform-LGI and estimate the Uniform-LGI constants for standard deep learning settings. Another work by Foster et al. (2018) studies similar LGI conditions. Under the key assumption that the *population risk* $\mathcal{L}_{\mathcal{D}}(w)$ satisfies the LGI on some given parameter space $\mathcal{W}$, the authors show that the excess risk $\mathcal{L}_{\mathcal{D}}(w) - \min_{w \in \mathcal{W}} \mathcal{L}_{\mathcal{D}}(w)$ is bounded above by $\sup_{w \in \mathcal{W}} \|\nabla \mathcal{L}_n(w) - \nabla \mathcal{L}_{\mathcal{D}}(w)\|$. In practice, it is not easy to validate whether $\mathcal{L}_{\mathcal{D}}$ satisfies the LGI because it depends on the data distribution and requires a very large number of samples. In comparison, we only assume that the *empirical risk* $\mathcal{L}_n(w)$ satisfies the LGI along the training path, and the Uniform-LGI condition can be numerically verified by the finite sample test algorithm. Under these two different assumptions, both our results and those of Foster et al. (2018) yield consistent generalization bounds $\mathcal{O}(1/\sqrt{n})$ for the linear regression models in the high-dimensional setting. A local version of the PL condition has been studied in Chatterjee (2022). This local PL condition is a

particular case of our Uniform-LGI with $\theta = 1/2$. In the case of feed-forward neural networks, Chatterjee (2022) show that gradient descent with proper initialization converges to a global minimum given that (1) the activation functions are smooth and strictly increasing; (2) the minimum value of the parameters is large enough; (3) the learning rate is small enough. In this work, we give both convergence guarantee (to a local minimum) and generalization analysis based on the Uniform-LGI condition.

## 5 Conclusion

In this work, we address the questions of when and why an optimization algorithm finds a minimum with good generalization properties. For this purpose, we propose a framework to bridge the gap between optimization and generalization based on the optimization path. The pivotal component is the Uniform-LGI condition: a condition on the loss function that is generally satisfied during the training of standard machine learning models. Using this assumption, we show that gradient flow converges to a stationary point with an explicit convergence rate and we derive generalization bound that hold during training. Finally, we apply the framework to three widely used machine learning models. By estimating the optimization path length, we get non-vacuous generalization bounds, even in the high dimensional case.

## 6 Acknowledgement

We thank the anonymous TMLR reviewers for several helpful comments. H. Yang is partially supported by the US National Science Foundation under award DMS- 2244988, DMS-2206333, and the Office of Naval Research Young Investigator Award. Q. Li is supported by the National Research Foundation, Singapore, under the NRF fellowship (project No. NRF-NRFF13-2021-0005).

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

## A  Experiments

In this section, we provide statistical results for the Uniform-LGI constants obtained by Algorithm 1 and details of the numerical evaluation of Figure 1 and 3.

**Experiments details for Figure 1.**  All the three models are trained with gradient descent with fixed learning rate $\eta$. Specifically, the following hyper-parameters are used for training: for model (a), initialization $w^{(0)} = 0.1$, $\eta = 0.1$, epoch $K = 10^6$, start point $K_0 = 50$, step $s = 10^3$; for model (b), $w^{(0)} = 1$, $\eta = 0.01$, epoch $K = 10^3$, start point $K_0 = 50$, step $s = 10$; for the linear regression model (c), the dataset contains 100 points $\{(x_i, y_i)\}_{i=1}^{100} \subseteq \mathbb{R}^{200} \times \mathbb{R}$, where $x_i(\forall i \in [100])$ are uniformly drawn from the 200-dimensional unit sphere, and $y_i(\forall i \in [100])$ are generated by a linear target function $y_i = \beta^\top x_i$ for some $\beta \in \mathbb{R}^{200}$ with $\|\beta\| = 1$. We train with random initialization, $\eta = 0.1$, epoch $K = 10^5$, start point $K_0 = 50$, step $s = 100$.

Next, we list experiments details for Figure 3.

**Dataset.**  We use the CIFAR10 dataset in our numerical evaluation. In particular, we only select the first two classes (airplane versus automobile) with totally 10000 training images. For the experiments in Figure 3 (a) and (b), we randomly choose the sample from the whole 10000 training images with size $n = 1000, 2000, \ldots, 10000$, where each class has the same sample size. For the experiment in Figure 3 (c), we use the whole 10000 training images.

**Model.**  We use two-layer fully connected neural network (no bias) with width $m = 512$, ReLU activation as the training model.

**Optimizer.**  We optimize the cross entropy loss by full batch gradient descent with random initialization and leaning rate 0.01. We stop training once the training loss is less than 0.001 or epoch reaches 20000. We do not use weight decay, dropout or batch normalization.

**Label flip.**  We only flip the training labels with different ratio.

**Uniform-LGI constants.**  We calculate the Uniform-LGI constants by the finite sample test (Algorithm 1) after the training. For the obtained training parameters $w^{(k)}$ at epoch $k$ for $k \in [0 : K]$, we estimate the optimum $w^*$ of $\mathcal{L}(w)$ as $\arg\min_{k \in [0:K]}$. To avoid division by zero, we delete the parameter $w^*$ in $w^{(0)}, \ldots, w^{(K)}$. Then we apply the finite sample test (Algorithm 1) with start point $K_0 = 1000$, step $s = 100$ to obtain the estimated Uniform-LGI constants. We set $(c_{n,\delta}, \theta_{n,\delta})$ to be the final estimation $(c_{K+1}, \theta_{K+1})$. We report the results with the mean and standard deviation over 5 independent runs.

The following table shows the statistical results of different models' Uniform-LGI constants obtained by Algorithm 1.

| Model | $\theta^*$ | $c^*$ |
|---|---|---|
| Linear Regression | $0.496 \pm 0.001$ | $0.063 \pm 0.0015$ |
| 2-layer MLP | $0.785 \pm 0.005$ | $0.066 \pm 0.013$ |
| ResNet18 | $0.751 \pm 0.019$ | $0.068 \pm 0.016$ |
| Wide ResNet | $0.514 \pm 0.011$ | $0.169 \pm 0.037$ |

Table 1: Mean and standard error for the Uniform-LGI constants $\theta^*, c^*$ obtained by finite sample test (Algorithm 1) over 10 independent runs over random initialization and data reshuffling.

## B  Proof of Theorem 2.2

In this section we will prove Theorem 2.2. The proof contains two parts, one is the convergence of the gradient flow, and another is the convergence rate derivation.

*Proof.* First, by the gradient flow (3), we have

$$\frac{d\mathcal{L}_n(w^{(t)})}{dt} = \left\langle \nabla\mathcal{L}_n(w^{(t)}), \frac{dw^{(t)}}{dt} \right\rangle = - \left\| \nabla\mathcal{L}_n(w^{(t)}) \right\|^2 \leq 0,$$

which indicates that the loss value $\mathcal{L}_n(w^{(t)})$ is non-increasing for $t$. Since $\mathcal{L}_n(w^{(t)})$ is always non-negative, then $\mathcal{L}_n(w^{(t)})$ converges to some limit $\mathcal{L}_\infty$, which is also the optimal loss value along the gradient flow path.

For the convergence rate, by the Uniform-LGI condition,

$$
\begin{aligned}
\frac{d\left(\mathcal{L}_n(w^{(t)}) - \mathcal{L}_\infty\right)}{dt} &= \left\langle \nabla\mathcal{L}_n(w^{(t)}), \frac{dw^{(t)}}{dt} \right\rangle \\
&= - \left\| \nabla\mathcal{L}_n(w^{(t)}) \right\|^2 \\
&\leq -c_n^2 \left( \mathcal{L}_n(w^{(t)}) - \mathcal{L}_\infty \right)^{2\theta_n}.
\end{aligned}
\tag{11}
$$

Therefore

$$\left( \mathcal{L}_n(w^{(t)}) - \mathcal{L}_\infty \right)^{-2\theta_n} d\left( \mathcal{L}_n(w^{(t)}) - \mathcal{L}_\infty \right) \leq -c_n^2 dt.$$

Integrating on both sides of the equation, we can get $\forall t \in [0, \infty)$,

when $\theta_n = \frac{1}{2}$,

$$\mathcal{L}_n(w^{(t)}) - \mathcal{L}_\infty \leq e^{-c_n^2 t} \left( \mathcal{L}_n(w^{(0)}) - \mathcal{L}_\infty \right); \tag{12}$$

when $\frac{1}{2} < \theta_n < 1$,

$$\mathcal{L}_n(w^{(t)}) - \mathcal{L}_\infty \leq (1 + Mt)^{-1/(2\theta_n - 1)} \left( \mathcal{L}_n(w^{(0)}) - \mathcal{L}_\infty \right), \tag{13}$$

where $M = c_n^2 (2\theta_n - 1) \left( \mathcal{L}_n(w^{(0)}) - \mathcal{L}_\infty \right)^{2\theta_n - 1}$.

For the distance bound, we consider to bound the gradient flow path length $\int_0^T \left\| \frac{dw^{(t)}}{dt} \right\| dt$. Notice that

$$
\begin{aligned}
\frac{d\mathcal{L}_n(w^{(t)})}{dt} &= \left\langle \nabla\mathcal{L}_n(w^{(t)}), \frac{dw^{(t)}}{dt} \right\rangle \\
&= - \left\| \nabla\mathcal{L}_n(w^{(t)}) \right\| \left\| \frac{dw^{(t)}}{dt} \right\| \\
&\leq -c_n \left( \mathcal{L}_n(w^{(t)}) - \mathcal{L}_\infty \right)^{\theta_n} \left\| \frac{dw^{(t)}}{dt} \right\|.
\end{aligned}
$$

Hence $\forall t \in [0, \infty)$,

$$
\begin{aligned}
\int_0^t \left\| \frac{dw^{(s)}}{ds} \right\| ds &\leq \int_0^t -\frac{1}{c_n} \left( \mathcal{L}_n(w^{(s)}) - \mathcal{L}_\infty \right)^{-\theta_n} d\mathcal{L}_n(w^{(s)}) \\
&= \frac{1}{c_n(1 - \theta_n)} \left[ \left( \mathcal{L}_n(w^{(0)}) - \mathcal{L}_\infty \right)^{1-\theta_n} - \left( \mathcal{L}_n(w^{(t)}) - \mathcal{L}_\infty \right)^{1-\theta_n} \right].
\end{aligned}
$$

Now we begin to prove that $w^{(t)}$ converges. Notice that for any $s > t \geq 0$, we have

$$\int_t^s \left\| \frac{dw^{(\tau)}}{d\tau} \right\| d\tau \leq \frac{\left( \mathcal{L}_n(w^{(t)}) - \mathcal{L}_\infty \right)^{1-\theta_n} - \left( \mathcal{L}_n(w^{(s)}) - \mathcal{L}_\infty \right)^{1-\theta_n}}{c_n(1 - \theta_n)}.$$

Then for any discrete time sequence $t_1, t_2, \ldots, t_n, \ldots$ that increases to infinity, we have $\forall p > q > 0$,

$$\int_{t_q}^{t_p} \left\| \frac{dw^{(s)}}{ds} \right\| ds \leq \frac{\left( \mathcal{L}_n(w^{(t_q)}) - \mathcal{L}_\infty \right)^{1-\theta_n} - \left( \mathcal{L}_n(w^{(t_p)}) - \mathcal{L}_\infty \right)^{1-\theta_n}}{c_n(1-\theta_n)}.$$

Now we prove that the sequence $w^{(t_1)}, \ldots, w^{(t_n)}, \ldots$ is a Cauchy sequence in the Euclidean space with $\ell_2$ metric. Notice that $\frac{\left( \mathcal{L}_n(w^{(t_n)}) - \mathcal{L}_\infty \right)^{1-\theta_n}}{c_n(1-\theta_n)}$ converges to zero, thus it is a Cauchy sequence. Then for any $\varepsilon > 0$, there exists $M > 0$ such that $\forall p > q > M$,

$$\|w^{(t_p)} - w^{(t_q)}\| \leq \int_{t_q}^{t_p} \left\| \frac{dw^{(s)}}{ds} \right\| ds$$

$$\leq \frac{\left( \mathcal{L}_n(w^{(t_q)}) - \mathcal{L}_\infty \right)^{1-\theta_n} - \left( \mathcal{L}_n(w^{(t_p)}) - \mathcal{L}_\infty \right)^{1-\theta_n}}{c_n(1-\theta_n)}$$

$$\leq \varepsilon.$$

Therefore, $w^{(t_n)}$ is a Cauchy sequence, and then it converges. This means that for any increasing sequence $t_n \to \infty, w^{(t_n)}$ converges. Hence, the limits $\lim_{n \to \infty} w^{(t_n)}$ must be the same for any sequence $t_n \to \infty$. Otherwise, we can take two sequences $t_{n_k}, t_{m_k} (k = 1, 2, \ldots)$ with different limits $\lim_{k \to \infty} w^{(t_{n_k})} \neq \lim_{k \to \infty} w^{(t_{m_k})}$, then there exists an increasing sequence $u_k$ that contains two subsequences of $n_k, m_k$, such that these two subsequences have different limits (since $\lim_{k \to \infty} w^{(t_{n_k})} \neq \lim_{k \to \infty} w^{(t_{m_k})}$), which contradicts with the convergence of $w^{(t_{u_k})}$. This yields that for any increasing discrete sequence $t_n \to \infty, w^{(t_n)}$ converges to the same limit. Finally, we conclude that the continuous dynamics $w^{(t)}$ converges to some limit $w^{(\infty)}$, which is a stationary point with $\mathcal{L}_n(w^{(\infty)}) = \mathcal{L}_\infty$. Then we can replace $\mathcal{L}_\infty$ to $\mathcal{L}_n(w^{(\infty)})$ in all the above equations. Finally, note that the length is always an upper bound for the distance, which completes the proof.

$\square$

## C  Discrete time analysis

In this section, we provide optimization results for the gradient descent algorithm, i.e., we consider

$$w^{(t+1)} = w^{(t)} - \eta \nabla \mathcal{L}_n(w^{(t)}), \quad t = 0, 1, 2, \ldots$$

where $\eta$ is a fixed learning rate. We assume that the loss function is $\beta_\mathcal{L}$-smooth in its whole domain $\mathcal{W}$, i.e.,

$$|\mathcal{L}_n(w) - \mathcal{L}_n(v) - \langle \nabla \mathcal{L}_n(v), w - v \rangle| \leq \frac{\beta_\mathcal{L}}{2} \|w - v\|^2, \quad \forall w, v \in \mathcal{W}.$$

Note that a continuously differentiable function is $\beta$-smooth if its gradient is $\beta$-Lipschitz. In our analysis, we only focus on the bounded region of the gradient descent path, and the gradient $\nabla \mathcal{L}_n(w)$ is usually locally Lipschitz inside this region. Hence, this is a reasonable assumption given that $\mathcal{W}$ is a compact set. Now we present our main theorem over the gradient descent algorithm.

**Theorem C.1.** *For a fixed initialization $w^{(0)}$, suppose that there exist $\theta_n \in [1/2, 1)$ and $c_n > 0$ such that the loss function $\mathcal{L}_n(w)$ satisfies the Uniform-LGI on $\{w^{(t)} : t = 0, 1, \ldots\}$ with $\theta_n, c_n$. Let the learning rate $\eta \in (0, 2/\beta_\mathcal{L})$, then $w^{(t)}$ converges to a stationary point $w^{(\infty)}$ with convergence rate given by*

$$\theta_n = 1/2 : \quad \mathcal{L}_n(w^{(t)}) - \mathcal{L}_n(w^{(\infty)}) \leq e^{-c_n^2(\eta - \eta^2 \beta_\mathcal{L}/2)t} \left( \mathcal{L}_n(w^{(0)}) - \mathcal{L}_n(w^{(\infty)}) \right);$$

$$\theta_n \in (1/2, 1) : \quad \mathcal{L}_n(w^{(t)}) - \mathcal{L}_n(w^{(\infty)}) \leq (1 + Mt)^{-1/(2\theta_n - 1)} \left( \mathcal{L}_n(w^{(0)}) - \mathcal{L}(w^{(\infty)}) \right),$$

*where $M = c_n^2(2\theta_n - 1)\left(\eta - \frac{\eta^2 \beta_\mathcal{L}}{2}\right) \left( \mathcal{L}_n(w^{(0)}) - \mathcal{L}_n(w^{(\infty)}) \right)^{2\theta_n - 1}$. The distance between the initialization $w^{(0)}$ and the parameter $w^{(t)}$ at step $t$ is bounded by*

$$\|w^{(0)} - w^{(t)}\| \leq \frac{1}{c_n(1 - \theta_n)(1 - \eta \beta_\mathcal{L}/2)} \left[ \left( \mathcal{L}_n(w^{(0)}) - \mathcal{L}_n(w^{(\infty)}) \right)^{1-\theta_n} - \left( \mathcal{L}_n(w^{(t)}) - \mathcal{L}_n(w^{(\infty)}) \right)^{1-\theta_n} \right].$$

*Proof.* First, by the $\beta_{\mathcal{L}}$-smoothness, we have

$$
\begin{aligned}
\mathcal{L}_n(w^{(t+1)}) &\leq \mathcal{L}_n(w^{(t)}) + \left\langle \nabla \mathcal{L}_n(w^{(t)}), w^{(t+1)} - w^{(t)} \right\rangle + \frac{\beta_{\mathcal{L}}}{2} \left\| w^{(t+1)} - w^{(t)} \right\|^2 \\
&= \mathcal{L}_n(w^{(t)}) - \left( \frac{1}{\eta} - \frac{\beta_{\mathcal{L}}}{2} \right) \left\| w^{(t+1)} - w^{(t)} \right\|^2 \\
&< \mathcal{L}_n(w^{(t)}).
\end{aligned}
$$

Therefore, $\mathcal{L}_n(w^{(t+1)})$ is strictly decreasing for $t$. By the above argument, we can also get

$$
\left\| \nabla \mathcal{L}_n(w^{(t)}) \right\|^2 \leq \frac{\mathcal{L}_n(w^{(t)}) - \mathcal{L}_n(w^{(t+1)})}{\eta - \eta^2 \beta_{\mathcal{L}}/2}. \tag{14}
$$

Since $\mathcal{L}_n(w^{(t)})$ is non-increasing, then $\mathcal{L}_n(w^{(t)})$ converges to some limit $\mathcal{L}_\infty$, which is also the optimal loss value along the gradient descent path.

For the convergence rate, by the Uniform-LGI condition and equation (14),

$$
\begin{aligned}
\mathcal{L}_n(w^{(t+1)}) - \mathcal{L}_n(w^{(t)}) &\leq -\left( \eta - \frac{\eta^2 \beta_{\mathcal{L}}}{2} \right) \left\| \nabla \mathcal{L}_n(w^{(t)}) \right\|^2 \\
&\leq -c_n^2 \left( \eta - \frac{\eta^2 \beta_{\mathcal{L}}}{2} \right) \left( \mathcal{L}_n(w^{(t)}) - \mathcal{L}_\infty \right)^{2\theta_n}.
\end{aligned}
$$

Notice that

$$
\begin{aligned}
c_n^2 \left( \eta - \frac{\eta^2 \beta_{\mathcal{L}}}{2} \right) &\leq \frac{\mathcal{L}_n(w^{(t)}) - \mathcal{L}_n(w^{(t+1)})}{\left( \mathcal{L}_n(w^{(t)}) - \mathcal{L}_\infty \right)^{2\theta_n}} \\
&= \int_{\mathcal{L}_n(w^{(t+1)})}^{\mathcal{L}_n(w^{(t)})} \frac{1}{\left( \mathcal{L}_n(w^{(t)}) - \mathcal{L}_\infty \right)^{2\theta_n}} d\mathcal{L}_n(w) \\
&\leq \int_{\mathcal{L}_n(w^{(t+1)})}^{\mathcal{L}_n(w^{(t)})} \frac{1}{\left( \mathcal{L}_n(w) - \mathcal{L}_\infty \right)^{2\theta_n}} d\mathcal{L}_n(w).
\end{aligned}
$$

when $\theta_n = \frac{1}{2}$, we have

$$
\int_{\mathcal{L}_n(w^{(t+1)})}^{\mathcal{L}_n(w^{(t)})} \frac{1}{\left( \mathcal{L}_n(w) - \mathcal{L}_\infty \right)^{2\theta_n}} d\mathcal{L}_n(w) = \log \frac{\mathcal{L}_n(w^{(t)}) - \mathcal{L}_\infty}{\mathcal{L}_n(w^{(t+1)}) - \mathcal{L}_\infty},
$$

which yields that

$$
\frac{\mathcal{L}_n(w^{(t+1)}) - \mathcal{L}_\infty}{\mathcal{L}_n(w^{(t)}) - \mathcal{L}_\infty} \leq e^{-c_n^2 (\eta - \frac{\eta^2 \beta_{\mathcal{L}}}{2})}.
$$

Hence,

$$
\mathcal{L}_n(w^{(t)}) - \mathcal{L}_\infty \leq e^{-c_n^2 (\eta - \frac{\eta^2 \beta_{\mathcal{L}}}{2}) t} \left( \mathcal{L}_n(w^{(0)}) - \mathcal{L}_\infty \right).
$$

When $\frac{1}{2} < \theta_n < 1$, we have

$$
\int_{\mathcal{L}_n(w^{(t+1)})}^{\mathcal{L}_n(w^{(t)})} \frac{1}{\left( \mathcal{L}_n(w) - \mathcal{L}_\infty \right)^{2\theta_n}} d\mathcal{L}_n(w) = \frac{\left( \mathcal{L}_n(w^{(t)}) - \mathcal{L}_\infty \right)^{1-2\theta_n} - \left( \mathcal{L}_n(w^{(t+1)}) - \mathcal{L}_\infty \right)^{1-2\theta_n}}{1 - 2\theta_n},
$$

which yields that

$$
\frac{1}{\left( \mathcal{L}_n(w^{(t+1)}) - \mathcal{L}_\infty \right)^{2\theta_n - 1}} - \frac{1}{\left( \mathcal{L}_n(w^{(t)}) - \mathcal{L}_\infty \right)^{2\theta_n - 1}} \geq c_n^2 (2\theta_n - 1) \left( \eta - \frac{\eta^2 \beta_{\mathcal{L}}}{2} \right).
$$

Hence,

$$\mathcal{L}_n(w^{(t)}) - \mathcal{L}_\infty \leq (1 + Mt)^{-1/(2\theta_n - 1)} \left( \mathcal{L}_n(w^{(0)}) - \mathcal{L}_\infty \right),$$

where $M = c_n^2 (2\theta_n - 1) \left( \eta - \frac{\eta^2 \beta_\mathcal{L}}{2} \right) \left( \mathcal{L}_n(w^{(0)}) - \mathcal{L}_\infty \right)^{2\theta_n - 1}$.

For the distance bound, by the $\beta_\mathcal{L}$-smoothness and the Uniform-LGI condition, we have

$$\left\| w^{(t+1)} - w^{(t)} \right\| \leq \frac{\mathcal{L}_n(w^{(t)}) - \mathcal{L}(w^{(t+1)})}{\left( 1 - \frac{\eta \beta_\mathcal{L}}{2} \right) \left\| \nabla \mathcal{L}_n(w^{(t)}) \right\|}$$

$$\leq \frac{\mathcal{L}_n(w^{(t)}) - \mathcal{L}(w^{(t+1)})}{\left( 1 - \frac{\eta \beta_\mathcal{L}}{2} \right) c_n (\mathcal{L}_n(w^{(t)}) - \mathcal{L}_\infty)^{\theta_n}}$$

$$= \frac{1}{c_n (1 - \frac{\eta \beta_\mathcal{L}}{2})} \int_{\mathcal{L}_n(w^{(t+1)})}^{\mathcal{L}_n(w^{(t)})} \frac{1}{\left( \mathcal{L}_n(w^{(t)}) - \mathcal{L}_\infty \right)^{\theta_n}} d\mathcal{L}_n(w)$$

$$\leq \frac{1}{c_n (1 - \frac{\eta \beta_\mathcal{L}}{2})} \int_{\mathcal{L}_n(w^{(t+1)})}^{\mathcal{L}_n(w^{(t)})} \frac{1}{\left( \mathcal{L}_n(w) - \mathcal{L}_\infty \right)^{\theta_n}} d\mathcal{L}_n(w)$$

$$= \frac{1}{c_n (1 - \theta_n)(1 - \frac{\eta \beta_\mathcal{L}}{2})} \left[ \left( \mathcal{L}_n(w^{(t)}) - \mathcal{L}_\infty \right)^{1 - \theta_n} - \left( \mathcal{L}_n(w^{(t+1)}) - \mathcal{L}_\infty \right)^{1 - \theta_n} \right].$$

Now we look at the sequence $w^{(0)}, w^{(1)} \ldots, w^{(t)}, \ldots$. Notice that $\mathcal{L}_n(w^{(t)})$ converges to $\mathcal{L}_\infty$, hence $\frac{\left( \mathcal{L}_n(w^{(t)}) - \mathcal{L}_\infty \right)^{1-\theta_n}}{c_n (1 - \theta_n)(1 - \eta \beta_\mathcal{L}/2)}$ converges to zero as $t$ goes to infinity. Therefore, the sequence $\frac{\left( \mathcal{L}_n(w^{(t)}) - \mathcal{L}_\infty \right)^{1-\theta_n}}{c_n (1 - \theta_n)(1 - \eta \beta_\mathcal{L}/2)}$ is a Cauchy sequence. Then for any $\varepsilon > 0$, there is $M > 0$ such that

$$\frac{\left( \mathcal{L}_n(w^{(t)}) - \mathcal{L}_\infty \right)^{1-\theta_n}}{c_n (1 - \theta_n)(1 - \eta \beta_\mathcal{L}/2)} - \frac{\left( \mathcal{L}_n(w^{(s)}) - \mathcal{L}_\infty \right)^{1-\theta_n}}{c_n (1 - \theta_n)(1 - \eta \beta_\mathcal{L}/2)} < \varepsilon, \quad \forall s > t > M.$$

Hence $\forall s > t > M$,

$$\left\| w^{(s)} - w^{(t)} \right\| \leq \sum_{i=t}^{s-1} \left\| w^{(i+1)} - w^{(i)} \right\| \leq \frac{\left( \mathcal{L}_n(w^{(t)}) - \mathcal{L}_\infty \right)^{1-\theta_n}}{c_n (1 - \theta_n)(1 - \eta \beta_\mathcal{L}/2)} - \frac{\left( \mathcal{L}_n(w^{(s)}) - \mathcal{L}_\infty \right)^{1-\theta_n}}{c_n (1 - \theta_n)(1 - \eta \beta_\mathcal{L}/2)} < \varepsilon.$$

Therefore, $w^{(0)}, w^{(1)} \ldots, w^{(t)}, \ldots$ is a Cauchy sequence in the Euclidean space with $\ell_2$ metric. Thus, $w^{(t)}$ converges to some limit $w^{(\infty)}$, which is a stationary point with $\mathcal{L}_n(w^{(\infty)}) = \mathcal{L}_\infty$. Then we can replace $\mathcal{L}_\infty$ to $\mathcal{L}_n(w^{(\infty)})$ in all the above equations. Finally, we have

$$\| w^{(0)} - w^{(t)} \| \leq \frac{1}{c_n (1 - \theta_n)(1 - \frac{\eta \beta_\mathcal{L}}{2})} \left[ \left( \mathcal{L}_n(w^{(0)}) - \mathcal{L}_n(w^{(\infty)}) \right)^{1-\theta_n} - \left( \mathcal{L}_n(w^{(t)}) - \mathcal{L}_n(w^{(\infty)}) \right)^{1-\theta_n} \right].$$

$\square$

# D   Proof of Theorem 2.3

In this section, we will prove Theorem 2.3. This proof is based on the Rademacher complexity theory and the covering number of $\ell_2$ balls. The key idea is to use the Uniform-LGI condition to bound the gradient flow path length $\int_0^T \left\| \frac{dw^{(t)}}{dt} \right\| dt$. Then since the path length is always an upper bound for the distance, the parameter $w^{(T)}$ lies in a norm-constrained parameter space $\left\{ w : \left\| w - w^{(0)} \right\| \leq \int_0^T \left\| \frac{dw^{(t)}}{dt} \right\| dt \right\}$. This allows us to use Rademacher complexity theory to obtain the generalization results.

Now we introduce some known technical lemmas that are used to build our proof. In the first lemma, we derive an explicit estimate for the gradient flow path length over the random choice of the training sample $S$.

**Lemma D.1.** *Under the same conditions and notations in Theorem 2.3, let $T$ be the time when the empirical loss $\mathcal{L}_n(w^{(T)}) = \varepsilon \mathcal{L}_n(w^{(0)})$ for some $\varepsilon \in [0,1]$, then with probability at least $1 - \delta/2$ over $S$ we have*

$$\int_0^T \left\| \frac{dw^{(t)}}{dt} \right\| dt \leq \frac{\left( M_\delta - \bar{M}_\delta \right)^{1-\theta_n} - \left( \varepsilon - \bar{M}_\delta \right)^{1-\theta_n}}{c_n(1-\theta_n)}.$$

*Proof.* By the Uniform-LGI condition, we have

$$\frac{d\mathcal{L}_n(w^{(t)})}{dt} = \left\langle \nabla \mathcal{L}_n(w^{(t)}), \frac{dw^{(t)}}{dt} \right\rangle$$

$$= - \left\| \nabla \mathcal{L}_n(w^{(t)}) \right\| \left\| \frac{dw^{(t)}}{dt} \right\|$$

$$\leq -c_n \left( \mathcal{L}_n(w^{(t)}) - \mathcal{L}_n(w^{(\infty)}) \right)^{\theta_n} \left\| \frac{dw^{(t)}}{dt} \right\|.$$

Hence with probability at least $1 - \delta/2$ over $S$,

$$\int_0^T \left\| \frac{dw^{(t)}}{dt} \right\| dt \leq \int_0^T -\frac{1}{c_n} \left( \mathcal{L}_n(w^{(t)}) - \mathcal{L}_n(w^{(\infty)}) \right)^{-\theta_n} d\mathcal{L}_n(w^{(t)})$$

$$= \frac{1}{c_n(1-\theta_n)} \left[ \left( \mathcal{L}_n(w^{(0)}) - \mathcal{L}_n(w^{(\infty)}) \right)^{1-\theta_n} - \left( \mathcal{L}_n(w^{(T)}) - \mathcal{L}_n(w^{(\infty)}) \right)^{1-\theta_n} \right]$$

$$= \frac{1}{c_n(1-\theta_n)} \left[ \left( \mathcal{L}_n(w^{(0)}) - \mathcal{L}_n(w^{(\infty)}) \right)^{1-\theta_n} - \left( \varepsilon \mathcal{L}_n(w^{(0)}) - \mathcal{L}_n(w^{(\infty)}) \right)^{1-\theta_n} \right]$$

$$\leq \frac{1}{c_n(1-\theta_n)} \left[ \left( M_\delta - \mathcal{L}_n(w^{(\infty)}) \right)^{1-\theta_n} - \left( \varepsilon M_\delta - \mathcal{L}_n(w^{(\infty)}) \right)^{1-\theta_n} \right]$$

$$\leq \frac{1}{c_n(1-\theta_n)} \left[ \left( M_\delta - \bar{M}_\delta \right)^{1-\theta_n} - \left( \varepsilon M_\delta - \bar{M}_\delta \right)^{1-\theta_n} \right].$$

The last inequality is by the fact that the function $f(x) = (t-x)^{1-\theta} - (s-x)^{1-\theta}$ is non-decreasing on $[0,s]$ given that $t \geq s$.

$\square$

The second lemma gives a generalization bound of a function class based on the Rademacher complexity, which is proved in Mohri et al. (2018).

**Lemma D.2.** *Consider a family of functions $\mathcal{F}$ mapping from $\mathcal{Z}$ to $[a,b]$. Let $\mathcal{D}$ denote the distribution according to which samples are drawn. Then for any $\delta > 0$, with probability at least $1 - \delta$ over the draw of an i.i.d. sample $S = \{z_1, \ldots, z_n\}$ of size $n$, the following holds for all $f \in \mathcal{F}$:*

$$\mathbb{E}_{z \sim \mathcal{D}} [f(z)] - \frac{1}{n} \sum_{i=1}^n f(z_i) \leq 2\mathcal{R}_S(\mathcal{F}) + 3(b-a)\sqrt{\frac{\log(2/\delta)}{2n}},$$

*where $\mathcal{R}_S(\mathcal{F})$ is the empirical Rademacher complexity with respect to the sample $S$, defined as:*

$$\mathcal{R}_S(\mathcal{F}) = \mathbb{E}_\sigma \left[ \sup_{f \in \mathcal{F}} \frac{1}{n} \sum_{i=1}^n \sigma_i f(z_i) \right].$$

*Here $\{\sigma_i\}_{i=1}^n$ are i.i.d. random variables drawn from $U\{-1,1\}$.*

In the next lemma, we prove a shifted version of the Ledoux-Talagrand contraction inequality (Ledoux & Talagrand, 2013), which is useful to bound the length-based Rademacher complexity.

**Lemma D.3** (Shifted contraction inequality). *Let $g : \mathbb{R} \to \mathbb{R}$ be a convex and increasing function. Let $\phi_i : \mathbb{R} \to \mathbb{R}$ be $L$-Lipschitz functions, then for any bounded set $T \subset \mathbb{R}$ and any $t^{(0)} \in \mathbb{R}$, we have*

$$\mathbb{E}_\sigma \left[ g \left( \sup_{\left(t - t^{(0)}\right) \in T} \sum_{i=1}^n \sigma_i \left( \phi_i(t_i) - \phi_i \left( t_i^{(0)} \right) \right) \right) \right] \leq \mathbb{E}_\sigma \left[ g \left( L \sup_{\left(t - t^{(0)}\right) \in T} \sum_{i=1}^n \sigma_i \left( t_i - t_i^{(0)} \right) \right) \right],$$

*and*

$$\mathbb{E}_\sigma \left[ \sup_{\left(t - t^{(0)}\right) \in T} \left| \sum_{i=1}^n \sigma_i \left( \phi_i(t_i) - \phi_i \left( t_i^{(0)} \right) \right) \right| \right] \leq 2L \mathbb{E}_\sigma \left[ \sup_{\left(t - t^{(0)}\right) \in T} \left| \sum_{i=1}^n \sigma_i \left( t_i - t_i^{(0)} \right) \right| \right].$$

The special case for $t^{(0)} = 0$ corresponds to the original Ledoux-Talagrand contraction inequality. Here we prove a shifted version.

*Proof.* First notice that

$$\mathbb{E}_\sigma \left[ g \left( \sup_{\left(t - t^{(0)}\right) \in T} \sum_{i=1}^n \sigma_i \left( \phi_i(t_i) - \phi_i \left( t_i^{(0)} \right) \right) \right) \right] = \mathbb{E}_{\sigma_1, \dots, \sigma_{n-1}} \left[ \mathbb{E}_{\sigma_n} \left[ g \left( \sup_{\left(t - t^{(0)}\right) \in T} \sum_{i=1}^n \sigma_i \left( \phi_i(t_i) - \phi_i \left( t_i^{(0)} \right) \right) \right) \right] \right].$$

Let $u_{n-1}(t) = \sum_{i=1}^{n-1} \sigma_i \left( \phi_i(t_i) - \phi_i \left( t_i^{(0)} \right) \right)$, then

$$\mathbb{E}_{\sigma_n} \left[ g \left( \sup_{\left(t - t^{(0)}\right) \in T} \sum_{i=1}^n \sigma_i \left( \phi_i(t_i) - \phi_i \left( t_i^{(0)} \right) \right) \right) \right]$$

$$= \frac{1}{2} g \left( \sup_{\left(t - t^{(0)}\right) \in T} u_{n-1}(t) + \left( \phi_n(t_n) - \phi_n \left( t_n^{(0)} \right) \right) \right) + \frac{1}{2} g \left( \sup_{\left(t - t^{(0)}\right) \in T} u_{n-1}(t) - \left( \phi_n(t_n) - \phi_n \left( t_n^{(0)} \right) \right) \right).$$

Suppose that the above two suprema can be reached at $t'$ and $\tilde{t}$ respectively, i.e.,

$$\sup_{\left(t - t^{(0)}\right) \in T} u_{n-1}(t) + \left( \phi_n(t_n) - \phi_n \left( t_n^{(0)} \right) \right) = u_{n-1} \left( t' \right) + \left( \phi_n \left( t'_n \right) - \phi_n \left( t_n^{(0)} \right) \right);$$

$$\sup_{\left(t - t^{(0)}\right) \in T} u_{n-1}(t) - \left( \phi_n(t_n) - \phi_n \left( t_n^{(0)} \right) \right) = u_{n-1} \left( \tilde{t} \right) - \left( \phi_n \left( \tilde{t}_n \right) - \phi_n \left( t_n^{(0)} \right) \right).$$

Otherwise we add an arbitrary positive number $\varepsilon$ in the above equations. Therefore,

$$\mathbb{E}_{\sigma_n} \left[ g \left( \sup_{\left(t - t^{(0)}\right) \in T} \sum_{i=1}^n \sigma_i \left( \phi_i(t_i) - \phi_i \left( t_i^{(0)} \right) \right) \right) \right]$$

$$= \frac{1}{2} \left[ g \left( u_{n-1} \left( t' \right) + \left( \phi_n \left( t'_n \right) - \phi_n \left( t_n^{(0)} \right) \right) \right) \right] + \frac{1}{2} \left[ g \left( u_{n-1} \left( \tilde{t} \right) - \left( \phi_n \left( \tilde{t}_n \right) - \phi_n \left( t_n^{(0)} \right) \right) \right) \right].$$

Without loss of generality, we assume

$$u_{n-1} \left( t' \right) + \left( \phi_n \left( t'_n \right) - \phi_n \left( t_n^{(0)} \right) \right) \geq u_{n-1} \left( \tilde{t} \right) + \left( \phi_n \left( \tilde{t}_n \right) - \phi_n \left( t_n^{(0)} \right) \right);$$

$$u_{n-1} \left( \tilde{t} \right) - \left( \phi_n \left( \tilde{t}_n \right) - \phi_n \left( t_n^{(0)} \right) \right) \geq u_{n-1} \left( t' \right) - \left( \phi_n \left( t'_n \right) - \phi_n \left( t_n^{(0)} \right) \right).$$

$$(15)$$

For the other cases, the method remains the same. We set

$$
\begin{aligned}
a &= u_{n-1}\left(\tilde{t}\right) - \left(\phi_n\left(\tilde{t}_n\right) - \phi_n\left(t_n^{(0)}\right)\right), \\
b &= u_{n-1}\left(\tilde{t}\right) - L\left(\tilde{t}_n - t_n^{(0)}\right), \\
a' &= u_{n-1}\left(t'\right) + L\left(t'_n - t_n^{(0)}\right), \\
b' &= u_{n-1}\left(t'\right) + \left(\phi_n\left(t'_n\right) - \phi_n\left(t_n^{(0)}\right)\right).
\end{aligned}
$$

Now our goal is to prove:

$$
g(a) - g(b) \le g\left(a'\right) - g\left(b'\right). \tag{16}
$$

Considering the following four cases:

1. $t'_n \ge t_n^{(0)}$ and $\tilde{t}_n \ge t_n^{(0)}$. By the Lipschitzness of $\phi_n$ and equation (15) we know $a \ge b, b' \ge b$, and

$$
(a - b) - (a' - b') = \phi_n\left(t'_n\right) - \phi_n\left(\tilde{t}_n\right) - L\left(t'_n - \tilde{t}_n\right).
$$

If $t'_n \ge \tilde{t}_n$, we can get $a - b \le a' - b'$. By the fact that $g$ is convex and increasing, we have $g(y + x) - g(x)$ is increasing in $y$ for every $x \ge 0$. Hence for $x = a - b$,

$$
g(a) - g(b) = g(b + x) - g(b) \le g\left(b' + x\right) - g(b') \le g\left(a'\right) - g\left(b'\right).
$$

If $t'_n < \tilde{t}_n$, we change $\phi_n$ into $-\phi_n$ and switch $t'$ and $\tilde{t}$, and the proof is similar.

2. $t'_n \le t_n^{(0)}$ and $\tilde{t}_n \le t_n^{(0)}$. Similarly, by changing the signs we can get the same result.

3. $t'_n \ge t_n^{(0)}$ and $\tilde{t}_n \le t_n^{(0)}$. For this case we have $a \le b$ and $b' \le a'$, so $g(a) + g\left(b'\right) \le g(b) + g\left(a'\right)$.

4. $t'_n \le t_n^{(0)}$ and $\tilde{t}_n \ge t_n^{(0)}$. For this case we can change $\phi_n$ to $-\phi_n$, then we have $a \ge b$ and $a' \le b'$, and finally we get $g(a) + g\left(b'\right) \le g(b) + g\left(a'\right)$.

Thus equation (16) yields that

$$
\begin{aligned}
&\mathbb{E}_{\sigma_n}\left[g\left(\sup_{(t - t^{(0)}) \in T} \sum_{i=1}^{n} \sigma_i\left(\phi_i(t_i) - \phi_i\left(t_i^{(0)}\right)\right)\right)\right] \\
&= \frac{1}{2}\left[g\left(u_{n-1}\left(t'\right) + \left(\phi_n\left(t'_n\right) - \phi_n\left(t_n^{(0)}\right)\right)\right)\right] + \frac{1}{2}\left[g\left(u_{n-1}\left(\tilde{t}\right) - \left(\phi_n\left(\tilde{t}_n\right) - \phi_n\left(t_n^{(0)}\right)\right)\right)\right] \\
&\le \frac{1}{2}\left[g\left(u_{n-1}\left(t'\right) + L\left(t'_n - t_n^{(0)}\right)\right)\right] + \frac{1}{2}\left[g\left(u_{n-1}\left(\tilde{t}\right) - L\left(\tilde{t}_n - t_n^{(0)}\right)\right)\right] \\
&\le \frac{1}{2}\left[g\left(\sup_{(t - t^{(0)}) \in T} u_{n-1}(t) + L\left(t_n - t_n^{(0)}\right)\right)\right] + \frac{1}{2}\left[g\left(\sup_{(t - t^{(0)}) \in T} u_{n-1}(t) - L\left(t_n - t_n^{(0)}\right)\right)\right] \\
&= \mathbb{E}_{\sigma_n}\left[g\left(\sup_{(t - t^{(0)}) \in T} u_{n-1}(t) + \sigma_n L\left(t_n - t_n^{(0)}\right)\right)\right]
\end{aligned}
$$

Applying the same method to $\sigma_{n-1}, \ldots, \sigma_1$ successively, we obtain the first inequality

$$
\mathbb{E}_{\sigma}\left[g\left(\sup_{(t - t^{(0)}) \in T} \sum_{i=1}^{n} \sigma_i\left(\phi_i(t_i) - \phi_i\left(t_i^{(0)}\right)\right)\right)\right] \le \mathbb{E}_{\sigma}\left[g\left(L \sup_{(t - t^{(0)}) \in T} \sum_{i=1}^{n} \sigma_i\left(t_i - t_i^{(0)}\right)\right)\right].
$$

For the second inequality, since $|x| = [x]_+ + [x]_-$ with $[x]_+ = \max(0, x)$ and $[x]_- = \max(0, -x)$,

$$\mathbb{E}_\sigma \left[ \sup_{(t-t^{(0)}) \in T} \left| \sum_{i=1}^n \sigma_i \left( \phi_i(t_i) - \phi_i\left(t_i^{(0)}\right) \right) \right| \right]$$

$$\leq \mathbb{E}_\sigma \left[ \sup_{(t-t^{(0)}) \in T} \left[ \sum_{i=1}^n \sigma_i \left( \phi_i(t_i) - \phi_i\left(t_i^{(0)}\right) \right) \right]_+ \right] + \mathbb{E}_\sigma \left[ \sup_{(t-t^{(0)}) \in T} \left[ \sum_{i=1}^n \sigma_i \left( \phi_i(t_i) - \phi_i\left(t_i^{(0)}\right) \right) \right]_- \right]$$

$$= 2\mathbb{E}_\sigma \left[ \sup_{(t-t^{(0)}) \in T} \left[ \sum_{i=1}^n \sigma_i \left( \phi_i(t_i) - \phi_i\left(t_i^{(0)}\right) \right) \right]_+ \right],$$

where the last equality is by $[-x]_- = [x]_+$ and $\sigma$ has the same distribution with $-\sigma$.

A simple fact is that

$$\sup_{(t-t^{(0)}) \in T} \left[ \sum_{i=1}^n \sigma_i \left( \phi_i(t_i) - \phi_i\left(t_i^{(0)}\right) \right) \right]_+ = \left[ \sup_{(t-t^{(0)}) \in T} \sum_{i=1}^n \sigma_i \left( \phi_i(t_i) - \phi_i\left(t_i^{(0)}\right) \right) \right]_+.$$

Since $\max(0, x)$ is convex and increasing, then by the first inequality we have

$$\mathbb{E}_\sigma \left[ \sup_{(t-t^{(0)}) \in T} \left[ \sum_{i=1}^n \sigma_i \left( \phi_i(t_i) - \phi_i\left(t_i^{(0)}\right) \right) \right]_+ \right] = \mathbb{E}_\sigma \left[ \left[ \sup_{(t-t^{(0)}) \in T} \sum_{i=1}^n \sigma_i \left( \phi_i(t_i) - \phi_i\left(t_i^{(0)}\right) \right) \right]_+ \right]$$

$$\leq \mathbb{E}_\sigma \left[ \left[ L \sup_{(t-t^{(0)}) \in T} \sum_{i=1}^n \sigma_i \left( t_i - t_i^{(0)} \right) \right]_+ \right]$$

$$\leq L \mathbb{E}_\sigma \left[ \sup_{(t-t^{(0)}) \in T} \left| \sum_{i=1}^n \sigma_i \left( t_i - t_i^{(0)} \right) \right| \right]$$

This completes the proof. $\qquad\qquad\qquad\square$

Now we apply Lemma D.3 to bound the empirical Rademacher complexity of an element-wise distance constrained function class. In the following lemma, all the notations are consistent with Theorem 2.3 unless stated otherwise.

**Lemma D.4.** *Given a function class $\mathcal{F}_{a,b} := \{x \mapsto f(w, x) : w \in \mathcal{S}_{a,b}\}$ and sample $S = \{x_1, \ldots, x_n\}$ with $\|x_i\| = 1$ for all $i \in [n]$, then we have*

$$\mathcal{R}_S(\mathcal{F}_{a,b}) \leq \sqrt{\frac{\|a\|^2 + \|b\|^2}{n}} \|L_\Psi(\mathcal{S}_{a,b})\|.$$

*Proof.* By definition,

$$n\mathcal{R}_S(\mathcal{F}_{a,b}) = \mathbb{E}_\sigma \left[ \sup_{w \in \mathcal{S}_{a,b}} \sum_{i=1}^n \sigma_i f(w, x_i) \right]$$

$$= \mathbb{E}_\sigma \left[ \sup_{w \in \mathcal{S}_{a,b}} \sum_{i=1}^n \sigma_i f(w, x_i) \right] - \mathbb{E}_\sigma \left[ \sum_{i=1}^n \sigma_i f(0, x_i) \right]$$

$$= \mathbb{E}_\sigma \left[ \sup_{w \in \mathcal{S}_{a,b}} \sum_{i=1}^n \sigma_i \left( f(w, x_i) - f(0, x_i) \right) \right].$$

Now we decompose the term $f(w, x_i) - f(0, x_i)$ as:

$$
\begin{aligned}
&f(w, x_i) - f(0, x_i)\\
=&\Psi\left(x_i^\top \alpha_1, \ldots, x_i^\top \alpha_p, \beta_1, \ldots, \beta_q\right) - \Psi\left(0, \ldots, 0, 0, \ldots, 0\right)\\
=&\left(\Psi\left(x_i^\top \alpha_1,\ldots,x_i^\top \alpha_p,\beta_1,\ldots,\beta_q\right)-\Psi\left(0,\ldots,x_i^\top \alpha_p,\beta_1,\ldots,\beta_q\right)\right)+\left(\Psi\left(0,\ldots,x_i^\top \alpha_p,\beta_1,\ldots,\beta_q\right)-\Psi\left(0,0,\ldots,x_i^\top \alpha_p,\beta_1,\ldots,\beta_q\right)\right)\\
&+\cdots+\left(\Psi\left(0,\ldots,0,0,\ldots,0,\beta_q\right)-\Psi\left(0,\ldots,0,0,\ldots,0\right)\right).
\end{aligned}
$$

Then by the above decomposition and Lemma D.3, we have

$$
\mathbb{E}_\sigma\left[\sup_{w\in\mathcal{S}_{a,b}}\sum_{i=1}^n \sigma_i\left(f(w, x_i) - f(0, x_i)\right)\right]
$$

$$
\leq L_\Psi^{(1)}(\mathcal{S}_{a,b})\mathbb{E}_\sigma\left[\sup_{w\in\mathcal{S}_{a,b}}\sum_{i=1}^n \sigma_i x_i^\top \alpha_1\right] + \cdots + L_\Psi^{(p+q)}(\mathcal{S}_{a,b})\mathbb{E}_\sigma\left[\sup_{w\in\mathcal{S}_{a,b}}\sum_{i=1}^n \sigma_i \beta_q\right].
$$

Notice that

$$
\begin{aligned}
\mathbb{E}_\sigma\left[\sup_{w\in\mathcal{S}_{a,b}}\sum_{i=1}^n \sigma_i x_i^\top \alpha_1\right] &= \mathbb{E}_\sigma\left[\sup_{\|\alpha_1\|\leq a_1}\sum_{i=1}^n \sigma_i x_i^\top \alpha_1\right]\\
&\leq a_1\mathbb{E}_\sigma\left[\left\|\sum_{i=1}^n \sigma_i x_i\right\|\right]\\
&\leq a_1\sqrt{\mathbb{E}_\sigma\left[\left\|\sum_{i=1}^n \sigma_i x_i\right\|^2\right]}\\
&= a_1\sqrt{n}.
\end{aligned}
$$

And

$$
\begin{aligned}
\mathbb{E}_\sigma\left[\sup_{w\in\mathcal{S}_{a,b}}\sum_{i=1}^n \sigma_i \beta_q\right] &= \mathbb{E}_\sigma\left[\sup_{|\beta_q|\leq b_q}\sum_{i=1}^n \sigma_i \beta_q\right]\\
&\leq b_q\mathbb{E}_\sigma\left[\left|\sum_{i=1}^n \sigma_i\right|\right]\\
&\leq b_q\sqrt{\mathbb{E}_\sigma\left[\left|\sum_{i=1}^n \sigma_i\right|^2\right]}\\
&= b_q\sqrt{n}.
\end{aligned}
$$

Therefore, by the Cauchy-Schwarz inequality, we can get

$$
\mathbb{E}_\sigma\left[\sup_{w\in\mathcal{S}_{a,b}}\sum_{i=1}^n \sigma_i\left(f(w, x_i) - f(0, x_i)\right)\right] \leq L_\Psi^{(1)}(\mathcal{S}_{a,b})a_1\sqrt{n} + \cdots + L_\Psi^{(p+q)}(\mathcal{S}_{a,b})b_q\sqrt{n}
$$

$$
\leq \sqrt{n\left(\|a\|^2 + \|b\|^2\right)}\,\|L_\Psi(\mathcal{S}_{a,b})\|.
$$

Finally, we have

$$
\mathcal{R}_S(\mathcal{F}_{a,b}) \leq \sqrt{\frac{\|a\|^2 + \|b\|^2}{n}}\,\|L_\Psi(\mathcal{S}_{a,b})\|.
$$

$\square$

Lemma D.4 gives an upper bound of the Rademacher complexity based on the element-wise distance. To obtain a norm-based generalization bound, we consider to use $\mathcal{S}_{a,b}$ to cover the norm-constrained space $\{w : \|w\| \leq R\}$, and then taking a union bound. For the $\ell_2$ ball covering number, we use the following result from (Neyshabur et al., 2019, Lemma 11).

**Lemma D.5.** *Given any $\epsilon, D, \beta > 0$, consider the set $S_\beta^D = \{x \in \mathbb{R}^D : \|x\| \leq \beta\}$. Then there exist $N$ sets $\{T_i\}_{i=1}^N$ of the form $T_i = \{x \in \mathbb{R}^D : |x_j| \leq \alpha_j^i, \forall j \in [D]\}$ such that $S_\beta^D \subseteq \bigcup_{i=1}^N T_i$ and $\|\alpha^i\| \leq \beta(1+\epsilon), \forall i \in [N]$ where $N = \binom{K+D-1}{D-1}$ and*

$$K = \left\lceil \frac{D}{(1+\epsilon)^2 - 1} \right\rceil.$$

**Lemma D.6.** *For any two positive integers $n, k$ with $n \geq k$, we have*

$$\binom{n}{k} \leq \left(\frac{en}{k}\right)^k.$$

*Proof.* Note that

$$\binom{n}{k} = \frac{n!}{k!(n-k)!} \leq \frac{n^k}{k!} \leq e^k \left(\frac{n}{k}\right)^k.$$

The last step is by

$$e^k = \sum_{i=0}^\infty \frac{k^i}{i!} \geq \frac{k^k}{k!}.$$

$\square$

Now combining Lemma D.1, D.2, D.4, D.5 and D.6, we are ready to prove Theorem 2.3.

*Proof of Theorem 2.3.* Let $r_{n,\delta,\varepsilon} = \frac{(M_\delta - \bar{M}_\delta)^{1-\theta_n} - (\varepsilon - \bar{M}_\delta)^{1-\theta_n}}{c_n(1-\theta_n)}$, then by Lemma D.1, we may apply Lemma D.5 with $\epsilon = \sqrt{2} - 1$, $D = p + q$, and $\beta = r_{n,\delta,\varepsilon}$, then there exist $N$ sets $\mathcal{S}_{a^k,b^k}$ such that $S_\beta^D \subseteq \bigcup_{k=1}^N \mathcal{S}_{a^k,b^k}$ and $\sqrt{\|a^k\|^2 + \|b^k\|^2} \leq \sqrt{2}\beta$, with $N = \binom{2D-1}{D-1}$.

Therefore, for each parameter space $\mathcal{S}_{a^k,b^k}$, by Lemma D.4 we have

$$\mathcal{R}_S(\mathcal{F}_{a^k,b^k}) \leq \sqrt{\frac{2}{n}}\beta \left\|L_\Psi(\mathcal{S}_{a^k,b^k})\right\|.$$

Notice that the local Lipschitz constant of $\ell$ in $\mathcal{S}_{a^k,b^k}$ is $L_\ell(\mathcal{S}_{a^k,b^k})$. Hence, by Lemma D.2 and the Ledoux-Talagrand contraction inequality, for any $\delta \in (0,1)$, with probability at least $1 - \delta/2N$ over the training sample, the following holds for all $w \in \mathcal{S}_{a^k,b^k}$:

$$\mathcal{L}_\mathcal{D}(w_\varepsilon) \leq \varepsilon + \frac{2\sqrt{2}\beta L_\ell(\mathcal{S}_{a^k,b^k}) \left\|L_\Psi(\mathcal{S}_{a^k,b^k})\right\|}{\sqrt{n}} + 3M_\beta \sqrt{\frac{\log(4N/\delta)}{2n}},$$

where $M_\beta = \sup_{\|a^k\|^2 + \|b^k\|^2 \leq 2\beta^2} \sup_{w \in \mathcal{S}_{a^k,b^k}, \|x\| \leq 1, |y| \leq 1} \ell(f(w,x), y)$.

Since $w_\varepsilon - w^{(0)} \in S_\beta^D \subseteq \bigcup_{k=1}^N \mathcal{S}_{a^k,b^k}$ with probability at least $1 - \delta/2$ over $S$, by taking the union bound over all sets $\mathcal{S}_{a^k,b^k}$, we can get with probability at least $1 - \delta$ over the training samples,

$$\mathcal{L}_\mathcal{D}(w_\varepsilon) \leq \varepsilon + \sup_{\|a\|^2 + \|b\|^2 \leq 2\beta^2} \frac{2\sqrt{2}\beta L_\ell(\mathcal{S}_{a,b}) \left\|L_\Psi(\mathcal{S}_{a,b})\right\|}{\sqrt{n}} + 3M_{a,b} \sqrt{\frac{\log(4N/\delta)}{2n}}.$$

Thus it remains to bound the term $\log N$. For $D = 1$, $N = 1$. For $D \geq 2$, by Lemma D.6,

$$\log N \leq (D-1)\log\left(\frac{e(2D-1)}{D-1}\right) < 2.1(D-1) < 3D = 3(p+q).$$

This completes the proof of Theorem 2.3.

$\square$

# E   Proofs for Section 3

In this section, our goal is to prove all the theorems in Section 3. A crucial part of the proofs is the spectral analysis of the random matrix $\mathcal{X}$. Therefore, we start with introducing the non-asymptotic results of the smallest and largest eigenvalues of subguassian matrices.

The first result is from (Rudelson & Vershynin, 2010, Proposition 2.4), characterizing the non-asymptotic behavior of the largest singular value of subgaussian matrices.

**Lemma E.1.** *Let $A$ be an $N \times n$ random matrix whose entries are independent mean zero subgaussian random variables whose subgaussian moments are bounded by 1. Then for every $t \geq 0$, with probability at least $1 - 2e^{-ct^2}$ over the randomness of the entries,*

$$\sqrt{\lambda_{\max}(AA^\top)} \leq C(\sqrt{N} + \sqrt{n}) + t,$$

*where $c$ and $C$ are two positive constants that depend only on the subgaussian moment of the entries.*

The second result is from (Rudelson & Vershynin, 2009, Theorem 1.1), characterizing the non-asymptotic behavior of the smallest singular value of subgaussian matrices.

**Lemma E.2.** *Let $A$ be an $N \times n$ random matrix whose entries are independent and identically distributed subgaussian random variables with zero mean and unit variance. If $N > n$, then for every $\varepsilon > 0$, with probability at least $1 - (C_1\varepsilon)^{N-n+1} - c_1^N$ over the randomness of the entries,*

$$\sqrt{\lambda_{\min}(A^\top A)} \geq \varepsilon(\sqrt{N} - \sqrt{n-1}),$$

*where $C_1 > 0$ and $c_1 \in (0,1)$ depend only on the subgaussian moment of the entries.*

In the next we introduce a useful lemma to bound $\lambda_{\max}(\mathcal{X}\mathcal{X}^\top)$ and $\lambda_{\min}(\mathcal{X}\mathcal{X}^\top)$.

**Lemma E.3.** *Let $\mathcal{X} \in \mathbb{R}^{n \times d}$ be a matrix with full row rank, $\Sigma_d \in \mathbb{R}^{d \times d}$ be a non-singular matrix, then*

$$\lambda_{\max}(\mathcal{X}\mathcal{X}^\top) \leq \frac{\lambda_{\max}(\mathcal{X}\Sigma_d\Sigma_d^\top\mathcal{X}^\top)}{\lambda_{\min}(\Sigma_d\Sigma_d^\top)}$$

$$\lambda_{\min}(\mathcal{X}\mathcal{X}^\top) \geq \frac{\lambda_{\min}(\mathcal{X}\Sigma_d\Sigma_d^\top\mathcal{X}^\top)}{\lambda_{\max}(\Sigma_d\Sigma_d^\top)}.$$

*Proof.* Since $\mathcal{X}$ has full row rank and $\Sigma_d$ is non-singular, then $\mathcal{X}\Sigma_d$ also has full row rank. Notice that

$$\lambda_{\max}(\mathcal{X}\mathcal{X}^\top) = \sup_{\|v\|=1} \left\|v^\top \mathcal{X}\Sigma_d\Sigma_d^{-1}\right\|^2 \leq \sup_{\|v\|=1} \left\|v^\top \mathcal{X}\Sigma_d\right\|^2 \left\|\Sigma_d^{-1}\right\|^2.$$

Note that

$$\|\Sigma_d^{-1}\| = \sup_{v \neq 0} \frac{\|\Sigma_d^{-1}v\|}{\|v\|} = \sup_{\Sigma_d^{-1}v \neq 0} \frac{\|\Sigma_d^{-1}v\|}{\|v\|} = \sup_{v \neq 0} \frac{\|v\|}{\|\Sigma_d v\|} = \frac{1}{\sqrt{\lambda_{\min}(\Sigma_d\Sigma_d^\top)}}.$$

Thus,

$$\lambda_{\max}(\mathcal{X}\mathcal{X}^\top) \leq \frac{\sup_{\|v\|=1} \left\|v^\top \mathcal{X}\Sigma_d\right\|^2}{\lambda_{\min}(\Sigma_d\Sigma_d^\top)} = \frac{\lambda_{\max}(\mathcal{X}\Sigma_d\Sigma_d^\top\mathcal{X}^\top)}{\lambda_{\min}(\Sigma_d\Sigma_d^\top)}.$$

For the smallest eigenvalue, note that

$$\lambda_{\min}(\mathcal{X}\mathcal{X}^\top) = \inf_{\|v\|=1} \left\| v^\top \mathcal{X}\Sigma_d\Sigma_d^{-1} \right\|^2$$

$$= \inf_{\|v\|=1} \frac{\left\| v^\top \mathcal{X}\Sigma_d\Sigma_d^{-1} \right\|^2}{\left\| v^\top \mathcal{X}\Sigma_d \right\|^2} \left\| v^\top \mathcal{X}\Sigma_d \right\|^2$$

$$\geq \inf_{\|v\|=1} \frac{\left\| v^\top \mathcal{X}\Sigma_d\Sigma_d^{-1} \right\|^2}{\left\| v^\top \mathcal{X}\Sigma_d \right\|^2} \inf_{\|v\|=1} \left\| v^\top \mathcal{X}\Sigma_d \right\|^2$$

$$\geq \inf_{v \neq 0} \frac{\| v^\top \Sigma_d^{-1} \|^2}{\|v\|^2} \lambda_{\min}(\mathcal{X}\Sigma_d\Sigma_d^\top \mathcal{X}^\top)$$

$$= \inf_{v \neq 0} \frac{\|v\|^2}{\|\Sigma_d^\top v\|^2} \lambda_{\min}(\mathcal{X}\Sigma_d\Sigma_d^\top \mathcal{X}^\top)$$

$$= \frac{\lambda_{\min}(\mathcal{X}\Sigma_d\Sigma_d^\top \mathcal{X}^\top)}{\lambda_{\max}(\Sigma_d\Sigma_d^\top)}.$$

$\square$

## E.1 Proof of Theorem 3.3

In this section, we will prove Theorem 3.3. All the notations are consistent with Theorem 3.3 unless stated otherwise.

*Proof of Theorem 3.3.* For the $\ell_2$ linear regression loss function $\mathcal{L}_n(w)$, notice that

$$\nabla \mathcal{L}_n(w) = \frac{1}{n} \sum_{i=1}^n \left( w^\top x_i - y_i \right) x_i = \frac{1}{n} \mathcal{X}^\top \left( \mathcal{X}w - \mathcal{Y} \right).$$

Then since $\mathcal{X}$ has full row rank, we have $\forall w \in \mathbb{R}^d$,

$$\|\nabla \mathcal{L}_n(w)\| = \frac{1}{n} \left\| \mathcal{X}^\top \left( \mathcal{X}w - \mathcal{Y} \right) \right\|$$

$$\geq \frac{\sqrt{\lambda_{\min}(\mathcal{X}\mathcal{X}^\top)}}{n} \|\mathcal{X}w - \mathcal{Y}\|$$

$$= \sqrt{\frac{2\lambda_{\min}(\mathcal{X}\mathcal{X}^\top)}{n}} \mathcal{L}_n(w)^{1/2}.$$

Since the optimal loss value is zero for underdetermined linear regression, we have

$$c_n = \sqrt{\frac{2\lambda_{\min}(\mathcal{X}\mathcal{X}^\top)}{n}}, \quad \theta_n = 1/2.$$

The convergence rate can be proved by directly plugging $c_n$ and $\theta_n$ into Theorem 2.2.

Next, we prove the generalization bound for $w_\varepsilon$. Notice that for the linear hypothesis function $f$ and the Lipschitz function $\tilde{\ell}$, we have $L_\ell(\mathcal{S}_{a,b}) = M_{a,b} = 1$, $\|L_\Psi(\mathcal{S}_{a,b})\| = \sqrt{2}$, $p = 1, q = 0, \bar{M}_\delta = 0$.

By the property of the target function, we have for any $w^{(0)}$ that satisfies $\left\| w^{(0)} \right\|_2 \leq c_0$,

$$\mathcal{L}_n(w^{(0)}) = \frac{1}{2n} \left\| \mathcal{X}w^{(0)} - \mathcal{Y} \right\|_2^2$$

$$\leq \frac{1}{n} \left( \left\| \mathcal{X}w^{(0)} \right\|_2^2 + \|\mathcal{Y}\|_2^2 \right)$$

$$\leq \frac{c_0^2 + (c^*)^2}{n} \lambda_{\max}(\mathcal{X}\mathcal{X}^\top).$$

Since in the following, we only need to bound the condition number of the data matrix $\mathcal{X}$, with loss of generality, we may assume that the entries of $\mathcal{X}$ have variance 1. Now we apply Lemma E.1 with $A = \mathcal{X}\Sigma_d$ and $t = \sqrt{\frac{\log(4/\delta)}{c}}$, then we have with probability at least $1 - \delta/2$ over the samples,

$$\sqrt{\lambda_{\max}(\mathcal{X}\Sigma_d\Sigma_d^\top \mathcal{X}^\top)} \leq C(\sqrt{n} + \sqrt{d}) + \sqrt{\frac{\log(4/\delta)}{c}}, \tag{17}$$

where $c$ and $C$ are two positive constants that depend only on the subgaussian moment of the entries.

By Lemma E.3, we have with the same probability,

$$\sqrt{\lambda_{\max}(\mathcal{X}\mathcal{X}^\top)} \leq \frac{C(\sqrt{n} + \sqrt{d}) + \sqrt{\frac{\log(4/\delta)}{c}}}{\sqrt{\lambda_0}}. \tag{18}$$

Thus,

$$
\begin{aligned}
M_\delta &= \frac{c_0^2 + (c^*)^2}{\lambda_0} \left( C\left(1 + \sqrt{\frac{d}{n}}\right) + \sqrt{\frac{\log(4/\delta)}{cn}} \right)^2 \\
&\leq \frac{c_0^2 + (c^*)^2}{\lambda_0} \left( C\left(1 + \frac{1}{\sqrt{\gamma_0}}\right) + \sqrt{\frac{\log(4/\delta)}{cn}} \right)^2.
\end{aligned}
$$

Similarly, let $\tau = c_1 \in (0,1), \varepsilon = \tau/C_1 > 0$, then Lemma E.2 implies that with probability at least $1 - \tau^{d-n+1} - \tau^d$ over the samples,

$$\sqrt{\lambda_{\min}(\mathcal{X}\Sigma_d\Sigma_d^\top \mathcal{X}^\top)} \geq \frac{\tau}{C_1}(\sqrt{d} - \sqrt{n-1}), \tag{19}$$

where $C_1 > 0$ and $\tau \in (0,1)$ depend only on the subgaussian moment of the entries.

By Lemma E.3, we have with the same probability,

$$\sqrt{\lambda_{\min}(\mathcal{X}\mathcal{X}^\top)} \geq \frac{\tau}{C_1\sqrt{\lambda_1}}(\sqrt{d} - \sqrt{n-1}). \tag{20}$$

Taking the union bound, we have with probability at least $1 - \delta/2 - \tau^{d-n+1} - \tau^d$ over the training samples,

$$
\begin{aligned}
r_{n,\delta,\varepsilon} &= \sqrt{2n}\frac{\sqrt{M_\delta} - \sqrt{\varepsilon M_\delta}}{\sqrt{\lambda_{\min}(\mathcal{X}\mathcal{X}^\top)}} \\
&\leq \sqrt{2(1-\varepsilon)n}\frac{\sqrt{M_\delta}}{\frac{\tau}{C_1\sqrt{\lambda_1}}(\sqrt{d} - \sqrt{n-1})} \\
&\leq \sqrt{2(1-\varepsilon)\lambda_1}\frac{\frac{c_0+c^*}{\sqrt{\lambda_0}}\left( C\left(1 + \frac{1}{\sqrt{\gamma_0}}\right) + \sqrt{\frac{\log(4/\delta)}{cn}} \right)}{\frac{\tau}{C_1}\left( \sqrt{\frac{d}{n}} - \sqrt{1 - \frac{1}{n}} \right)} \\
&\leq \sqrt{2(1-\varepsilon)\lambda_1}\frac{\frac{c_0+c^*}{\sqrt{\lambda_0}}\left( C\left(1 + \frac{1}{\sqrt{\gamma_0}}\right) + \sqrt{\frac{\log(4/\delta)}{cn}} \right)}{\frac{\tau}{C_1}\left( \frac{1}{\sqrt{\gamma_1}} - 1 \right)}.
\end{aligned}
\tag{21}
$$

Notice that $\ell$ is $\sqrt{l_0}$-Lipschitz on the first argument, then by Cauchy-Schwarz inequality, we have

$$
\begin{aligned}
\frac{1}{n}\sum_{i=1}^{n}\tilde{\ell}(f(w_\varepsilon, x_i), y_i) &\leq \frac{\sqrt{l_0}}{n}\sum_{i=1}^{n}|f(w_\varepsilon, x_i) - y_i| \\
&\leq \sqrt{\frac{l_0}{n}\sum_{i=1}^{n}|f(w_\varepsilon, x_i) - y_i|^2} \\
&\leq \sqrt{2l_0\mathcal{L}_n(w_\varepsilon)} \\
&= \sqrt{2l_0\varepsilon\mathcal{L}_n(w^{(0)})}.
\end{aligned}
\tag{22}
$$

Combining Theorem 2.3 and the inequality (21), we have with probability at least $1 - \delta - \tau^{d-n+1} - \tau^d$ over the training samples,

$$
\begin{aligned}
\mathbb{E}_{(x,y)\sim\mathcal{D}}\left[\tilde{\ell}\left(f(w_\varepsilon, x), y\right)\right] &\leq \sqrt{2l_0\varepsilon\mathcal{L}_n(w^{(0)})} + \frac{4r_{n,\delta,\varepsilon}}{\sqrt{n}} + 3l_0\sqrt{\frac{3 + \log(4/\delta)}{2n}} \\
&\leq \sqrt{2l_0\varepsilon\mathcal{L}_n(w^{(0)})} + \frac{4(c_0 + c^*)\sqrt{\frac{2(1-\varepsilon)\lambda_1}{\lambda_0}}\left(C\left(1 + \frac{1}{\sqrt{\gamma_0}}\right) + \sqrt{\frac{\log(4/\delta)}{cn}}\right)}{\frac{\tau}{C_1}\left(\frac{1}{\sqrt{\gamma_1}} - 1\right)\sqrt{n}} + 3l_0\sqrt{\frac{3 + \log(4/\delta)}{2n}},
\end{aligned}
$$

where $c, C, C_1 > 0$ and $\tau \in (0, 1)$ depend only on the subgaussian moment of the entries. This completes the proof of Theorem 3.3.

$\square$

## E.2 Proof of Theorem 3.5

In this section, we will prove Theorem 3.5. First, we present some useful lemmas for proving our results, and then we give the proofs of Theorem 3.5 for the RBF kernel and the inner product kernel separately.

For the RBF kernel $k(x, y) = \varrho(\|y - x\|)$, the following two lemmas give non-asymptotic bounds for $\lambda_{\max}(k(\mathcal{X}, \mathcal{X}))$ and $\lambda_{\min}(k(\mathcal{X}, \mathcal{X}))$ based on the separation distance $\mathsf{SD}$ of $\mathcal{X}$.

The first lemma is from (Diederichs & Iske, 2019, Lemma 3.1), providing an upper bound for $\lambda_{\max}(k(\mathcal{X}, \mathcal{X}))$.

**Lemma E.4.** *For the RBF kernel, if $\varrho : \mathbb{R}_{\geq 0} \to \mathbb{R}_{\geq 0}$ is a decreasing function, then*

$$
\lambda_{\max}(k(\mathcal{X}, \mathcal{X})) \leq \varrho(0) + 3d\sum_{t=1}^{\infty}(t + 2)^{d-1}\varrho(t \cdot \mathsf{SD}),
\tag{23}
$$

*and the sum of the infinite series in equation (23) is finite if and only if $\varrho(\|x\|) \in L^1(\mathbb{R}^d)$.*

The next lemma is from (Wendland, 2004, Theorem 12.3), giving a lower bound for $\lambda_{\min}(k(\mathcal{X}, \mathcal{X}))$.

**Lemma E.5.** *Suppose that $k$ is a positive-definite RBF kernel. If $\varrho(\|x\|) \in L^1(\mathbb{R}^d)$, one can define the Fourier transform of $\varrho$ as $\hat{\varrho}(\omega) := (2\pi)^{-d/2}\int_{\mathbb{R}^d}\varrho(\omega)e^{-ix^\top\omega}d\omega$. With a decreasing function $\varrho_0(M)$ and two constants $M_d, C_d$ defined as*

$$
\varrho_0(M) := \inf_{\|x\|\leq 2M}\hat{\varrho}(x), \quad M_d = 6.38d, \quad C_d = \frac{1}{2\Gamma(d/2 + 1)}\left(\frac{M_d}{2^{3/2}}\right)^d,
$$

*where $\Gamma$ is the gamma function. Then a lower bound on $\lambda_{\min}(k(\mathcal{X}, \mathcal{X}))$ is given by*

$$
\lambda_{\min}(k(\mathcal{X}, \mathcal{X})) \geq C_d \cdot \varrho_0(M_d/\mathsf{SD}) \cdot \mathsf{SD}^{-d}.
$$

For the inner product kernel, it is shown in El Karoui et al. (2010) that the kernel matrix can be approximated by the linear combination of all-ones matrix $11^\top$, sample covariance matrix and identity matrix. To obtain non-asymptotic results on the spectra of the kernel matrix, we borrow the technique from (Liang & Rakhlin, 2020, Proposition A.2), and show the result for subgaussian entries in the next lemma.

**Lemma E.6.** *For the inner product kernel, under Assumption 3.2, we have with probability at least $1-\delta-d^{-2}$ over the entries,*

$$\left\| k(\mathcal{X}, \mathcal{X}) - k^{\mathrm{lin}}(\mathcal{X}, \mathcal{X}) \right\| \leq d^{-1/2}\left( \delta^{-1/2} + \log^{0.51} d \right),$$

*where $k^{\mathrm{lin}}(\mathcal{X}, \mathcal{X})$ is defined as*

$$k^{\mathrm{lin}}(\mathcal{X}, \mathcal{X}) := \left( \varrho(0) + \frac{\varrho''(0)}{d} \right) 11^\top + \varrho'(0)\frac{\mathcal{X}\Sigma_d^2\mathcal{X}^\top}{d} + \left( \varrho(1) - \varrho(0) - \varrho'(0) \right) \mathbb{I}_{n\times n}.$$

*Proof.* Note that the sample covariance of $\mathcal{X}\Sigma_d$ is $\mathbb{I}_{d\times d}$, then by applying (Liang & Rakhlin, 2020, Proposition A.2) with subgaussian random entries we can prove this lemma.

$\square$

**Lemma E.7.** *Suppose that $A, B \in \mathbb{R}^{n\times n}$ are two symmetric matrices, then we have*

$$\lambda_{\min}(A + B) \geq \lambda_{\min}(A) + \lambda_{\min}(B).$$

*Proof.* Note that for any $x \in \mathbb{R}^n$ with $\|x\| = 1$,

$$x^\top(A + B)x = x^\top Ax + x^\top Bx \geq \lambda_{\min}(A) + \lambda_{\min}(B).$$

By definition, we have

$$\lambda_{\min}(A + B) = \inf_{\|x\|=1} x^\top(A + B)x \geq \lambda_{\min}(A) + \lambda_{\min}(B),$$

which completes the proof.

$\square$

Now we are ready to prove Theorem 3.5.

*Proof of Theorem 3.5.* First, notice that $\forall w \in \mathbb{R}^s$,

$$\begin{aligned}
\|\nabla\mathcal{L}_n(w)\| &= \frac{1}{n}\left\| \varphi(\mathcal{X})^\top\left(\varphi(\mathcal{X})w - \mathcal{Y}\right) \right\| \\
&\geq \frac{\sqrt{\lambda_{\min}(\varphi(\mathcal{X})\varphi(\mathcal{X})^\top)}}{n}\|\varphi(\mathcal{X})w - \mathcal{Y}\| \\
&= \sqrt{2\frac{\lambda_{\min}(k(\mathcal{X}, \mathcal{X}))}{n}}\mathcal{L}_n(w)^{1/2}.
\end{aligned}$$

Since the optimal loss value is zero for kernel regression, we have

$$c_n = \sqrt{\frac{2\lambda_{\min}(k(\mathcal{X}, \mathcal{X}))}{n}}, \quad \theta_n = 1/2.$$

Since $k$ is a positive-definite kernel, the convergence rate can be proved by directly plugging $c_n$ and $\theta_n$ into Theorem 2.2.

The proof of the generalization bound is two-sided. First, since $\forall x \in \mathcal{X}, \|\varphi(x)\| = \sqrt{k(x, x)} \leq 1$, then the kernel regression model (8) can be viewed as $\ell_2$ linear regression on inputs $\varphi(\mathcal{X})$. Hence, $\Psi$ is an identity

function with $p = 1, q = 0$, and $L_\ell(\mathcal{S}_{a,b}) = M_{a,b} = 1$, $\|L_\Psi(\mathcal{S}_{a,b})\| = \sqrt{2}$ for any $a, b$. The optimal empirical loss value $\bar{M}_\delta = 0$. This means that we only need to bound the terms $M_\delta$ and $r_{n,\delta,\varepsilon}$.

By the property of the target function, for any $w^{(0)}$ that satisfies $\|w^{(0)}\|_2 \le c_0$, we have

$$
\begin{aligned}
\mathcal{L}_n(w^{(0)}) &= \frac{1}{2n} \left\| \varphi(\mathcal{X}) w^{(0)} - \mathcal{Y} \right\|_2^2 \\
&\le \frac{1}{n} \left( \left\| \varphi(\mathcal{X}) w^{(0)} \right\|_2^2 + \|\mathcal{Y}\|_2^2 \right) \\
&\le \frac{c_0^2 + (c^*)^2}{n} \lambda_{\max}(k(\mathcal{X}, \mathcal{X})).
\end{aligned}
\tag{24}
$$

For $r_{n,\delta,\varepsilon}$, notice that

$$
\begin{aligned}
r_{n,\delta,\varepsilon} &= \sqrt{2n} \frac{\sqrt{M_\delta} - \sqrt{\varepsilon M_\delta}}{\sqrt{\lambda_{\min}(k(\mathcal{X}, \mathcal{X}))}} \\
&\le \sqrt{2(1-\varepsilon)} (c_0 + c^*) \sqrt{\frac{\lambda_{\max}(k(\mathcal{X}, \mathcal{X}))}{\lambda_{\min}(k(\mathcal{X}, \mathcal{X}))}}.
\end{aligned}
$$

Then for the RBF kernel with fixed input dimension $d$, Lemma E.4 and Lemma E.5 indicate that $\lambda_{\max}(k(\mathcal{X}, \mathcal{X}))$ and $\lambda_{\min}(k(\mathcal{X}, \mathcal{X}))$ are uniformly bounded with bounds depend only on $\varrho, d, q_{\min}, q_{\max}$. Hence, there exists a positive constant $C(\varrho, d, q_{\min}, q_{\max})$ that only depends on $\varrho, d, q_{\min}, q_{\max}$, such that

$$
r_{n,\delta,\varepsilon} \le \sqrt{2(1-\varepsilon)} (c_0 + c^*) C(\varrho, d, q_{\min}, q_{\max}).
$$

Therefore, by equation (22), Theorem 2.3 and Lemma D.2, with probability at least $1 - \delta$ over the training samples,

$$
\begin{aligned}
\mathbb{E}_{(x,y)\sim\mathcal{D}} \left[ \tilde{\ell}\left(f(w_\varepsilon, x), y\right) \right] &\le \sqrt{2\varepsilon \mathcal{L}_n(w^{(0)})} + \frac{4 r_{n,\delta,\varepsilon}}{\sqrt{n}} + 3\sqrt{\frac{3 + \log(4/\delta)}{2n}} \\
&\le \sqrt{2l_0 \varepsilon \mathcal{L}_n(w^{(0)})} + \frac{4\sqrt{2(1-\varepsilon)} (c_0 + c^*) C(\varrho, d, q_{\min}, q_{\max})}{\sqrt{n}} + 3l_0 \sqrt{\frac{3 + \log(4/\delta)}{2n}},
\end{aligned}
$$

which completes the proof for the RBF kernel.

For the inner product kernel, first notice that

$$
\begin{aligned}
\lambda_{\max}(k(\mathcal{X}, \mathcal{X})) &= \|k(\mathcal{X}, \mathcal{X})\| \\
&\le \|k^{\mathrm{lin}}(\mathcal{X}, \mathcal{X})\| + \|k(\mathcal{X}, \mathcal{X}) - k^{\mathrm{lin}}(\mathcal{X}, \mathcal{X})\|.
\end{aligned}
\tag{25}
$$

By Lemma E.7, we can get

$$
\begin{aligned}
\lambda_{\min}(k(\mathcal{X}, \mathcal{X})) &\ge \lambda_{\min}(k^{\mathrm{lin}}(\mathcal{X}, \mathcal{X})) + \lambda_{\min}(k(\mathcal{X}, \mathcal{X}) - k^{\mathrm{lin}}(\mathcal{X}, \mathcal{X})) \\
&\ge \lambda_{\min}(k^{\mathrm{lin}}(\mathcal{X}, \mathcal{X})) - \|k(\mathcal{X}, \mathcal{X}) - k^{\mathrm{lin}}(\mathcal{X}, \mathcal{X})\|.
\end{aligned}
\tag{26}
$$

Under Assumption 3.2 & 3.4, Lemma E.6 implies that

$$
\begin{aligned}
\|k^{\mathrm{lin}}(\mathcal{X}, \mathcal{X})\| &\le \frac{\varrho''(0)}{d} \|11^\top\| + \frac{\varrho'(0)}{d} \|\mathcal{X} \Sigma_d^2 \mathcal{X}^\top\| + (\varrho(1) - \varrho'(0)) \\
&\le \frac{n \varrho''(0)}{d} + \varrho'(0) \frac{\lambda_{\max}(\mathcal{X} \Sigma_d^2 \mathcal{X}^\top)}{d} + (\varrho(1) - \varrho'(0)) \\
&\le \gamma_1 \varrho''(0) + \varrho'(0) \frac{\lambda_{\max}(\mathcal{X} \Sigma_d^2 \mathcal{X}^\top)}{d} + (\varrho(1) - \varrho'(0)),
\end{aligned}
$$

and
$$\lambda_{\min}(k^{\lin}(\mathcal{X}, \mathcal{X})) \geq \varrho(1) - \varrho'(0) > 0.$$

Thus, by equation (26) we have
$$\lambda_{\min}(k(\mathcal{X}, \mathcal{X})) \geq (\varrho(1) - \varrho'(0)) - \left\| k(\mathcal{X}, \mathcal{X}) - k^{\lin}(\mathcal{X}, \mathcal{X}) \right\|. \tag{27}$$

Under Assumption 3.2, by equation (17), we have with probability at least $1 - \delta/3$ over the samples,

$$\begin{aligned}
\frac{\lambda_{\max}(\mathcal{X}\Sigma_d^2\mathcal{X}^\top)}{d} &\leq \left( C\left( \sqrt{\frac{n}{d}} + 1 \right) + \sqrt{\frac{\log(6/\delta)}{cd}} \right)^2 \\
&\leq \left( C\left( \sqrt{\gamma_1} + 1 \right) + \sqrt{\frac{\gamma_1 \log(6/\delta)}{cn}} \right)^2.
\end{aligned}$$

Therefore, by equation (25), with probability at least $1 - \delta/3$ over the samples,

$$\lambda_{\max}(k(\mathcal{X}, \mathcal{X})) \leq \gamma_1 \varrho''(0) + \varrho'(0)\left( C\left( \sqrt{\gamma_1} + 1 \right) + \sqrt{\frac{\gamma_1 \log(6/\delta)}{cn}} \right)^2 + \varrho(1) - \varrho'(0) + \left\| k(\mathcal{X}, \mathcal{X}) - k^{\lin}(\mathcal{X}, \mathcal{X}) \right\|. \tag{28}$$

By Lemma E.6, for large $d$ and small $\delta$ such that $d^{-1/2}\left( \sqrt{3}\delta^{-1/2} + \log^{0.51} d \right) \leq 0.5(\varrho(1) - \varrho'(0))$, we have with probability at least $1 - \delta/3 - d^{-2}$ over the entries,

$$\left\| k(\mathcal{X}, \mathcal{X}) - k^{\lin}(\mathcal{X}, \mathcal{X}) \right\| \leq 0.5(\varrho(1) - \varrho'(0)).$$

Then equation (27) and (28) yields that with probability at least $1 - 2\delta/3 - d^{-2}$ over the samples,

$$\lambda_{\min}(k(\mathcal{X}, \mathcal{X})) \geq 0.5(\varrho(1) - \varrho'(0)),$$
$$\lambda_{\max}(k(\mathcal{X}, \mathcal{X})) \leq \gamma_1 \varrho''(0) + \varrho'(0)\left( C\left( \sqrt{\gamma_1} + 1 \right) + \sqrt{\frac{\gamma_1 \log(4/\delta)}{cn}} \right)^2 + 1.5(\varrho(1) - \varrho'(0)).$$

Hence, we have with probability at least $1 - 2\delta/3 - d^{-2}$ over the samples,

$$\begin{aligned}
r_{n,\delta,\varepsilon} &= \sqrt{2n}\frac{\sqrt{M_\delta} - \sqrt{\varepsilon M_\delta}}{\sqrt{\lambda_{\min}(k(\mathcal{X}, \mathcal{X}))}} \\
&\leq \sqrt{2(1-\varepsilon)}\,(c_0 + c^*)\sqrt{\frac{\lambda_{\max}(k(\mathcal{X}, \mathcal{X}))}{\lambda_{\min}(k(\mathcal{X}, \mathcal{X}))}} \\
&\leq \sqrt{2(1-\varepsilon)}\,(c_0 + c^*)\sqrt{\frac{\gamma_1 \varrho''(0) + \varrho'(0)\left( C\left( \sqrt{\gamma_1} + 1 \right) + \sqrt{\frac{\gamma_1 \log(4/\delta)}{cn}} \right)^2 + 1.5(\varrho(1) - \varrho'(0))}{0.5(\varrho(1) - \varrho'(0))}}.
\end{aligned}$$

Therefore, there exists a constant $C$ that depends on $c_0, c^*, \gamma_1, \varrho''(0), \varrho'(0), \varrho(1)$ and the subgaussian moment of the entries such that with probability at least $1 - 2\delta/3 - d^{-2}$ over the samples,

$$r_{n,\delta,\varepsilon} \leq C\left( 1 + \sqrt{\frac{\log(1/\delta)}{n}} \right)\sqrt{1-\varepsilon}.$$

Combining equation (22) and Theorem 2.3, we get with probability at least $1 - \delta - d^{-2}$ over the samples,

$$\mathbb{E}_{(x,y)\sim\mathcal{D}}\left[\tilde{\ell}\left(f(w_\varepsilon, x), y\right)\right] \leq \sqrt{2l_0\varepsilon\mathcal{L}_n(w^{(0)})} + \frac{4r_{n,\delta,\varepsilon}}{\sqrt{n}} + 3l_0\sqrt{\frac{3 + \log(4/\delta)}{2n}}$$

$$\leq \sqrt{2l_0\varepsilon\mathcal{L}_n(w^{(0)})} + \frac{C\left(1 + \sqrt{\log(1/\delta)/n}\right)\sqrt{1-\varepsilon}}{\sqrt{n}} + 3l_0\sqrt{\frac{3 + \log(4/\delta)}{2n}},$$

where $C$ depends on $c_0, c^*, \gamma_1, \varrho''(0), \varrho'(0), \varrho(1)$ and the subgaussian moment of the entries, which completes the proof.

$\square$

### E.3 Proof of Proposition 3.6

In this section, we will prove Proposition 3.6.

*Proof.* We consider two phases: $t \in [0, T]$ and $t \in (T, \infty)$. First, notice that the loss function $\mathcal{L}_n(w)$ is a subanalytic function, thus it satisfies the classic LGI (Bolte et al., 2007). The classic LGI yields that there exist $c_n^*(S_n) > 0$, $\theta_n^*(S_n) \in [1/2, 1)$ such that the Uniform-LGI holds on some neighbor $U^*$ of the stationary point $w^{(\infty)}$. Thus, there exists $T > 0$ such that $\mathcal{L}_n(w)$ satisfies the Uniform-LGI on $\{w^{(t)} : t \in (T, \infty)\}$ with $c_n^*(S_n), \theta_n^*(S_n)$. Now we consider the gradient flow curve that is outside $U^*$, i.e., $t \in [0, T]$, where the gradient norm has a positive infimum. Then there exist $\hat{c}_n(S_n), \hat{\theta}_n(S_n)$ such that the Uniform-LGI holds for $t \in [0, T]$. Let $c_n(S_n) = \min(c_n^*(S_n), \hat{c}_n(S_n)) > 0$, $\theta_n(S_n) = \max(\theta_n^*(S_n), \hat{\theta}_n(S_n)) \in [1/2, 1)$, then $\mathcal{L}_n(w)$ satisfies Uniform-LGI along the whole gradient flow curve with $c_n(S_n)$ and $\theta_n(S_n)$.

For the population loss function $\mathcal{L}_{\mathcal{D}}(w) = \int \ell(f(w, x), y)d\mu(x, y)$ with the probability measure $d\mu(x, y)$ over the data distribution $\mathcal{D}$, if it is subanalytic, then it satisfies the classic LGI (Bolte et al., 2007) around the stationary point $w_{\mathcal{D}}^{(\infty)}$. By the same argument, there exist $c_{\mathcal{D}} > 0, \theta_{\mathcal{D}} \in [1/2, 1)$ such that $\mathcal{L}_{\mathcal{D}}(w)$ satisfies the Uniform-LGI along $\{w_{\mathcal{D}}^{(t)} : t \geq 0\}$ with $c_{\mathcal{D}}, \theta_{\mathcal{D}}$. $\square$

### E.4 Proof of Theorem 3.8

In this section, we will prove Theorem 3.8 for two-layer neural networks.

*Proof of Theorem 3.8.* The convergence rate can be directly obtained by Theorem 2.2. For the generalization result, we first give an upper bound for $\mathcal{L}_n(w^{(0)})$. Notice that

$$\mathcal{L}_n(w^{(0)}) = \frac{1}{2n}\sum_{i=1}^n \left((v^{(0)})^\top \phi(U^{(0)}x_i) - y_i\right)^2$$

$$\leq \frac{\sum_{i=1}^n \left((v^{(0)})^\top \phi(U^{(0)}x_i)\right)^2 + y_i^2}{n}$$

$$\leq \frac{1}{n}\sum_{i=1}^n \left((v^{(0)})^\top \phi(U^{(0)}x_i)\right)^2 + \frac{(c^*)^2}{n}\lambda_{\max}(\mathcal{X}\mathcal{X}^\top)$$

$$\leq \|v^{(0)}\|^2 \frac{\sum_{i=1}^n \|U^{(0)}x_i\|^2}{n} + \frac{(c^*)^2}{n}\lambda_{\max}(\mathcal{X}\mathcal{X}^\top)$$

$$= \frac{\|v^{(0)}\|^2 \|U^{(0)}\mathcal{X}^\top\|_F^2}{n} + \frac{(c^*)^2}{n}\lambda_{\max}(\mathcal{X}\mathcal{X}^\top)$$

$$\leq \frac{\|v^{(0)}\|^2 \|U^{(0)}\|_F^2 \|\mathcal{X}\|^2}{n} + \frac{(c^*)^2}{n}\lambda_{\max}(\mathcal{X}\mathcal{X}^\top)$$

$$\leq \left(\frac{1}{2}\|w^{(0)}\|^2 + (c^*)^2\right)\frac{\lambda_{\max}(\mathcal{X}\mathcal{X}^\top)}{n}.$$

Then by equation (18), we have with probability at least $1 - \delta/2$ over the samples,

$$
\begin{aligned}
\mathcal{L}_n(w^{(0)}) &\leq \left(\frac{1}{2}\|w^{(0)}\|^2 + (c^*)^2\right) \left(\frac{C(1 + \sqrt{d/n}) + \sqrt{\frac{\log(4/\delta)}{cn}}}{\sqrt{\lambda_0}}\right)^2 \\
&\leq \left(\frac{1}{2}\|w^{(0)}\|^2 + (c^*)^2\right) \left(\frac{C(1 + 1/\sqrt{\gamma_0}) + \sqrt{\frac{\log(4/\delta)}{cn}}}{\sqrt{\lambda_0}}\right)^2 \\
&\leq \frac{(\|w^{(0)}\|^2 + 2(c^*)^2)\left(C^2(1 + 1/\sqrt{\gamma_0})^2 + \frac{\log(4/\delta)}{cn}\right)}{\lambda_0}.
\end{aligned}
$$

Therefore, we can set $M_\delta$ to be

$$
M_\delta = \tilde{C}\left(1 + \frac{\log(1/\delta)}{n}\right),
$$

where $\tilde{C}$ is a constant that depends only on depends on $c^*, \gamma_0, \lambda_0, \|w^{(0)}\|$ and the subgaussian moment of the entries.

Next, we bound the term $\sup_{\|a\|^2 + \|b\|^2 \leq 2r^2_{n,\delta,\varepsilon}} \|L_\Psi(\mathcal{S}_{a,b})\|$. Note that

$$
\Psi(s_1, \ldots, s_m, t_1, \ldots, t_m) = \sum_{i=1}^m s_i \phi(t_i).
$$

Hence

$$
\begin{aligned}
\sup_{\|a\|^2 + \|b\|^2 \leq 2r^2_{n,\delta,\varepsilon}} \|L_\Psi(\mathcal{S}_{a,b})\| &= \sup_{\|a\|^2 + \|b\|^2 \leq 2r^2_{n,\delta,\varepsilon}} \sqrt{\sum_{i=1}^m s_i^2 + \phi(t_i)^2} \\
&\leq \sup_{\|a\|^2 + \|b\|^2 \leq 2r^2_{n,\delta,\varepsilon}} \sqrt{\sum_{i=1}^m s_i^2 + t_i^2} \\
&= \sqrt{2} r_{n,\delta,\varepsilon}.
\end{aligned}
\tag{29}
$$

Now it remains to bound $r_{n,\delta,\varepsilon}$. Notice that with probability at least $1 - \delta/2$ over the training samples, we have

$$
r_{n,\delta,\varepsilon} \leq \frac{M_\delta^{1-\theta_{n,\delta}} - (\varepsilon M_\delta - \bar{M}_\delta)^{1-\theta_{n,\delta}}}{c_{n,\delta}(1 - \theta_{n,\delta})}.
$$

Lastly, for the global Lipschitz evaluation loss function $\tilde{\ell}$ and the two-layer neural network, we have $L_\ell(\mathcal{S}_{a,b}) = M_{a,b} = 1$, $p = q = m$. By equation (22) and Theorem 2.3, we can get with probability at least $1 - \delta$ over the training samples,

$$
\mathbb{E}_{(x,y)\sim\mathcal{D}}\left[\tilde{\ell}(f(w_\varepsilon, x), y)\right] \leq \sqrt{2l_0 \varepsilon \mathcal{L}_n(w^{(0)})} + \frac{4}{\sqrt{n}}\left(\frac{(\tilde{C}(1 + \log(1/\delta)/n))^{1-\theta_{n,\delta}} - (\varepsilon M_\delta - \bar{M}_\delta)^{1-\theta_{n,\delta}}}{c_{n,\delta}(1 - \theta_{n,\delta})}\right)^2 + 3l_0\sqrt{\frac{6m + \log(12/\delta)}{2n}},
$$

where $\tilde{C}$ is a constant that depends only on $c^*, \gamma_0, \lambda_0, \|w^{(0)}\|$ and the subgaussian moment of the entries.

$\square$

## E.5 Results for overparameterized two-layer neural networks

In this section, we derive the generalization result for the overparameterized two-layer neural networks, where our generalization bound (Theorem 2.3) becomes non-vacuous. The generalization bound is based on the

Rademacher complexity theory in Arora et al. (2019) in the NTK regime. To simplify the analysis, we only consider the case when $\varepsilon = 0$, i.e., the final convergence model.

Following the setting in Du et al. (2019), we only train the hidden layer $U$ and leave the output layer $v$ as random initialization to simplify the analysis.

**NTK matrix.** The NTK matrix $\Theta(t)$ is defined as: $\Theta_{ij}(t) = \langle \nabla_U f(w^{(t)}, x_i), \nabla_U f(w^{(t)}, x_j) \rangle$, and denote $\widehat{\Theta}$ by the limiting matrix[4]: $\widehat{\Theta}_{ij} = x_i^\top x_j \mathbb{E}_{w \sim \mathcal{N}(0, \frac{1}{d} \mathbb{I}_d)} \left[ \phi'(w^\top x_i) \phi'(w^\top x_j) \right], \forall i, j \in [n]$.

**Standard random initialization.** To get a clean non-asymptotic bound for the initial loss value, we consider the following random initialization scheme. $U^{(0)}$ is drawn from Gaussian $\mathcal{N}(0, \frac{1}{d} \mathbb{I}_{m \times d})$ and $v_i^{(0)}$ are drawn i.i.d. from uniform distribution $U\{-1/\sqrt{m}, 1/\sqrt{m}\}, \forall i \in [m]$. These random initialization schemes are also known as Xavier initialization (Glorot & Bengio, 2010) and Kaiming initialization (He et al., 2015).

**Theorem E.8.** *Consider an overparameterized two-layer ReLU neural network (10). For any $\delta \in (0, 1)$, if $m \geq \mathrm{poly}\left(n, \lambda_{\min}^{-1}(\widehat{\Theta}), \delta^{-1}\right)$, then we have the followings:*

- *With probability at least $1 - \delta$ over training samples and random initialization, $\mathcal{L}_n(w^{(t)})$ satisfies the Uniform-LGI on $\{w^{(t)} : t \geq 0\}$ with*

$$c_n = \sqrt{\lambda_{\min}(\widehat{\Theta})/n}, \quad \theta_n = 1/2.$$

- *$\mathcal{L}_n(w^{(t)})$ converges to zero linearly:*

$$\mathcal{L}_n(w^{(t)}) \leq \exp\left(-\lambda_{\min}(\widehat{\Theta})t/n\right) \mathcal{L}_n(w^{(0)}).$$

- *Under Assumption 3.2, for any target function that satisfies (7), if $\gamma_1 \in (0, 1)$, then with probability at least $1 - \delta - \tau^{d-n+1} - \tau^d$ over the samples and random initialization,*

$$\mathbb{E}_{(x,y) \sim \mathcal{D}} \left[ \tilde{\ell}\left( f(w^{(\infty)}, x), y \right) \right] \leq \mathcal{O}\left( \sqrt{\frac{\log(n/\delta)}{n}} \right),$$

*where $\tau \in (0, 1)$ depends only on the subgaussian moment of the entries.*

To prove Theorem E.8, we first introduce some important lemmas for proving our final results. The first two lemmas are used to get an estimate for the initial loss value under the random initialization.

**Lemma E.9** ((Montgomery-Smith, 1990)). *If $\{\sigma_i\}_{i=1}^n$ are i.i.d. drawn from $U\{-1, 1\}$, then for any $x = (x_1, \ldots, x_n)^\top \in \mathbb{R}^n$, with probability at least $1 - \delta$ over $\sigma$,*

$$\left| \sum_{i=1}^n \sigma_i x_i \right| \leq \sqrt{2 \log(2/\delta)} \|x\|.$$

The following lemma gives a sharp bound for a Chi-square variable, which is from (Laurent & Massart, 2000, Lemma 1).

**Lemma E.10.** *Let $(Y_1, \ldots, Y_D)$ be i.i.d. Gaussian variables, with mean 0 and variance 1. Then with probability at least $1 - \delta$ over $Y$,*

$$\sum_{i=1}^D Y_i^2 \leq D + 2\sqrt{D \log\left(\frac{1}{\delta}\right)} + 2 \log\left(\frac{1}{\delta}\right).$$

---

[4]Here $\lambda_{\min}(\widehat{\Theta})$ changes with $n$.

Lemma E.11 shows that the smallest eigenvalue of the NTK matrix $\Theta(t)$ has a lower bounded given the overparameterization, by which we can prove the optimization result. In Lemma E.12, we show that the eigenvalues of the NTK matrix are related to the data covariance matrix. Then by combining Lemma E.13 and Lemma E.9 we can prove the generalization result.

**Lemma E.11.** *For any $\delta \in (0,1)$, if $m \geq \mathrm{poly}\left(n, \lambda_{\min}^{-1}(\widehat{\Theta}), \delta^{-1}\right)$, then with probability at least $1 - \delta$ over the random initialization,*

$$\lambda_{\min}(\Theta(t)) \geq \frac{1}{2}\lambda_{\min}(\widehat{\Theta}), \quad \forall t \geq 0.$$

*Proof.* The proof is the same as the proof of (Du et al., 2019, Lemma 3.4).

$\square$

**Lemma E.12.**
$$\lambda_{\min}(\widehat{\Theta}) \geq \lambda_{\min}\left(\mathcal{X}\mathcal{X}^\top\right)/4.$$

*Proof.* Notice that for ReLU activation $\phi$, a simple fact is that $\phi'(ax) = \phi'(x)$ holds for any $x \in \mathbb{R}$ given that $a > 0$. Therefore,

$$\begin{aligned}
\widehat{\Theta}_{ij} &= x_i^\top x_j \mathbb{E}_{w \sim \mathcal{N}(0, \frac{1}{d}\mathbb{I}_d)}\left[\phi'(w^\top x_i)\phi'(w^\top x_j)\right] \\
&= x_i^\top x_j \mathbb{E}_{w \sim \mathcal{N}(0, \mathbb{I}_d)}\left[\phi'(w^\top x_i)\phi'(w^\top x_j)\right] \\
&= \frac{x_i^\top x_j(\pi - \arccos(x_i^\top x_j))}{2\pi} \\
&= \frac{x_i^\top x_j}{4} + \frac{x_i^\top x_j}{2\pi}\arcsin(x_i^\top x_j) \\
&= \frac{x_i^\top x_j}{4} + \frac{1}{2\pi}\sum_{k=0}^{\infty}\frac{(2k)!}{4^k(k!)^2(2k+1)}(x_i^\top x_j)^{2k+2}.
\end{aligned}$$

Then

$$\begin{aligned}
\widehat{\Theta} &= \frac{\mathcal{X}\mathcal{X}^\top}{4} + \frac{1}{2\pi}\sum_{k=0}^{\infty}\frac{(2k)!}{4^k(k!)^2(2k+1)}\left(\mathcal{X}\mathcal{X}^\top\right)^{\circ(2k+2)} \\
&= \frac{\mathcal{X}\mathcal{X}^\top}{4} + \frac{1}{2\pi}\sum_{k=0}^{\infty}\frac{(2k)!}{4^k(k!)^2(2k+1)}\left((\mathcal{X}^\top)^{\odot(2k+2)}\right)^\top(\mathcal{X}^\top)^{\odot(2k+2)},
\end{aligned}$$

where $\circ$ is the element-wise product, and $\odot$ is the Khatri-Rao product[5].

Since $\left((\mathcal{X}^\top)^{\odot(2k+2)}\right)^\top(\mathcal{X}^\top)^{\odot(2k+2)}$ is positive semidefinite, we have

$$\lambda_{\min}(\widehat{\Theta}) \geq \lambda_{\min}\left(\mathcal{X}\mathcal{X}^\top\right)/4,$$

which completes the proof.

$\square$

The next lemma is quoted from (Arora et al., 2019, Lemma 5.4), giving an upper bound for the empirical Rademacher complexity if one has an accurate estimate for the distance with respect to each hidden unit.

---

[5]For $A = (a_1, \ldots, a_n) \in \mathbb{R}^{m \times n}, B = (b_1, \ldots, b_n) \in \mathbb{R}^{p \times n}$, then $A \odot B = [a_1 \otimes b_1, \ldots, a_n \otimes b_n]$, where $\otimes$ is the Kronecker product.

**Lemma E.13.** *Given $R > 0$, consider the following function class*

$$\mathcal{F} = \left\{ x \mapsto f(w, x) : \left\| u_r - u_r^{(0)} \right\| \leq R(\forall r \in [m]), \left\| U - U^{(0)} \right\|_F \leq B \right\}$$

*Then for an i.i.d. sample $S = \{x_1, \ldots, x_n\}$ and every $B > 0$, with probability at least $1 - \delta$ over the random initialization, the empirical Rademacher complexity is bounded as:*

$$\mathcal{R}_S(\mathcal{F}) \leq \frac{B}{\sqrt{2n}} \left( 1 + \left( \frac{2 \log(2/\delta)}{m} \right)^{1/4} \right) + 2R^2 \sqrt{md} + R\sqrt{2 \log(2/\delta)}.$$

Now we are ready to prove Theorem E.8.

*Proof of Theorem E.8.* By Lemma E.11, if $m \geq \text{poly}\left( n, \lambda_{\min}^{-1}(\widehat{\Theta}), \delta^{-1} \right)$, then with probability at least $1 - \delta$ over the random initialization,

$$
\begin{aligned}
\left\| \nabla \mathcal{L}_n(w^{(t)}) \right\| &= \frac{1}{n} \left\| \sum_{i=1}^n (f(w, x_i) - y_i) \nabla f(w^{(t)}, x_i) \right\| \\
&= \frac{1}{n} \left\| \nabla f(w^{(t)}, \mathcal{X})^\top \left( f(w^{(t)}, \mathcal{X}) - \mathcal{Y} \right) \right\| \\
&= \frac{1}{n} \sqrt{\left( f(w^{(t)}, \mathcal{X}) - \mathcal{Y} \right)^\top \nabla f(w^{(t)}, \mathcal{X}) \nabla f(w^{(t)}, \mathcal{X})^\top \left( f(w^{(t)}, \mathcal{X}) - \mathcal{Y} \right)} \\
&\geq \sqrt{\frac{2\lambda_{\min}(\Theta(t))}{n}} \sqrt{\mathcal{L}_n(w^{(t)})} \\
&\geq \sqrt{\frac{\lambda_{\min}(\widehat{\Theta})}{n}} \sqrt{\mathcal{L}_n(w^{(t)})}
\end{aligned}
$$

holds for any $t \geq 0$, which means that $\mathcal{L}_n(w^{(t)})$ satisfies the Uniform-LGI for any $t \geq 0$ with

$$c_n = \sqrt{\lambda_{\min}(\widehat{\Theta})/n}, \quad \theta_n = 1/2.$$

For the convergence rate, by equation (11), we can directly get $\mathcal{L}_n(w^{(t)})$ converges to zero with a linear convergence rate:

$$\mathcal{L}_n(w^{(t)}) \leq \exp\left( -\lambda_{\min}(\widehat{\Theta}) t/n \right) \mathcal{L}_n(w^{(0)}).$$

For the generalization bound, we first bound the initial loss value under the random initialization. By the property of the target function, we have

$$
\begin{aligned}
\mathcal{L}_n(w^{(0)}) &= \frac{1}{2n} \sum_{i=1}^n \left( (v^{(0)})^\top \phi(U^{(0)} x_i) - y_i \right)^2 \\
&\leq \frac{\sum_{i=1}^n \left( (v^{(0)})^\top \phi(U^{(0)} x_i) \right)^2 + y_i^2}{n} \\
&\leq \frac{1}{n} \sum_{i=1}^n \left( (v^{(0)})^\top \phi(U^{(0)} x_i) \right)^2 + \frac{(c^*)^2}{n} \lambda_{\max}(\mathcal{X}\mathcal{X}^\top).
\end{aligned}
$$

Since the entries of $v^{(0)}$ are drawn i.i.d. from $U\{-1/\sqrt{m}, 1/\sqrt{m}\}$, then by Lemma E.9, for each $i \in [n]$, with probability at least $1 - \delta/12n$ over $v^{(0)}$,

$$\frac{1}{n} \left( (v^{(0)})^\top \phi(U^{(0)} x_i) \right)^2 \leq \frac{2}{mn} \log\left( \frac{24n}{\delta} \right) \left\| \phi(U^{(0)} x_i) \right\|^2.$$

Taking the union bound over all $i = 1, 2, \ldots, n$, we have with probability at least $1 - \delta/12$ over the random initialization,

$$
\begin{aligned}
\frac{1}{n} \sum_{i=1}^{n} \left( (v^{(0)})^{\top} \phi(U^{(0)} x_i) \right)^2 &\leq \frac{2 \log(24n/\delta)}{mn} \sum_{i=1}^{n} \left\| \phi(U^{(0)} x_i) \right\|^2 \\
&= \frac{2 \log(24n/\delta)}{mn} \left\| \phi \left( U^{(0)} \mathcal{X}^{\top} \right) \right\|_F^2 \\
&\leq \frac{2 \log(24n/\delta)}{mn} \left\| U^{(0)} \mathcal{X}^{\top} \right\|_F^2 \\
&\leq \frac{2 \log(24n/\delta)}{mn} \left\| U^{(0)} \right\|_F^2 \left\| \mathcal{X} \right\|^2 \\
&= \frac{2 \log(24n/\delta)}{mn} \left\| U^{(0)} \right\|_F^2 \lambda_{\max}(\mathcal{X} \mathcal{X}^{\top}).
\end{aligned}
\tag{30}
$$

For the Gaussian random matrix $U^{(0)} \sim \mathcal{N}(0, \frac{1}{d} \mathbb{I}_{m \times d})$, by Lemma E.10, we have with probability at least $1 - \delta/24$ over the random initialization,

$$
\frac{\left\| U^{(0)} \right\|_F^2}{m} \leq 1 + 2\sqrt{\frac{\log(24/\delta)}{md}} + \frac{2 \log(24/\delta)}{md}.
\tag{31}
$$

Taking the union bound of equation (18), (30) and (31), we have with probability at least $1 - \delta/6$ over the samples and the random initialization,

$$
\begin{aligned}
\mathcal{L}_n(w^{(0)}) &\leq \left( \frac{2 \log(24n/\delta)}{mn} \left\| U^{(0)} \right\|_F^2 + \frac{(c^*)^2}{n} \right) \lambda_{\max}(\mathcal{X} \mathcal{X}^{\top}) \\
&\leq \left( 2 \log(24n/\delta) \left( 1 + 2\sqrt{\frac{\log(24/\delta)}{md}} + \frac{2 \log(24/\delta)}{md} \right) + (c^*)^2 \right) \left( C(1 + \sqrt{d/n}) + \sqrt{\frac{\log(48/\delta)}{cn}} \right)^2 / \lambda_0 \\
&\leq \left( 2 \log(24n/\delta) \left( 2 + \frac{3 \log(24/\delta)}{md} \right) + (c^*)^2 \right) \left( C(1 + 1/\sqrt{\gamma_0}) + \sqrt{\frac{\log(48/\delta)}{cn}} \right)^2 / \lambda_0 \\
&\leq \left( 2 \log(24n/\delta) \left( 2 + \frac{3\gamma_1 \log(24/\delta)}{n} \right) + (c^*)^2 \right) \left( C(1 + 1/\sqrt{\gamma_0}) + \sqrt{\frac{\log(48/\delta)}{cn}} \right)^2 / \lambda_0
\end{aligned}
$$

where $c$ and $C$ are two positive constants that depend only on the subgaussian moment of the entries.

Therefore, there exists a constant $\tilde{C}$ that depends only on $\gamma_0, \gamma_1, \lambda_0$ and the subgaussian moment of the entries, such that with probability at least $1 - \delta/6$ over the samples and the random initialization,

$$
\mathcal{L}_n(w^{(0)}) \leq \tilde{C} \log(n/\delta).
\tag{32}
$$

Thus, we can set $M_\delta = \tilde{C} \log(n/\delta)$.

By Lemma E.12, when $\varepsilon = 0$, $r_{n,\delta,\varepsilon}$ can be bounded as

$$
r_{n,\delta,\varepsilon} = 2\sqrt{\frac{nM_\delta}{\lambda_{\min}(\widehat{\Theta})}} \leq 4\sqrt{\frac{nM_\delta}{\lambda_{\min}(\mathcal{X} \mathcal{X}^{\top})}}.
$$

Taking the union bound of equation (20) and (32), if $m \geq \text{poly}\left(n, \lambda_{\min}^{-1}(\widehat{\Theta}), \delta^{-1}\right)$, then with probability at least $1 - \tau^{d-n+1} - \tau^d - \delta/6$ over the samples and random initialization,

$$
\begin{aligned}
r_{n,\delta,\varepsilon} &\leq 4\sqrt{\frac{nM_\delta}{\lambda_{\min}(\mathcal{X}\mathcal{X}^\top)}} \\
&\leq 4\sqrt{\frac{n\tilde{C}\log(n/\delta)}{\lambda_{\min}(\mathcal{X}\mathcal{X}^\top)}} \\
&\leq \frac{4\sqrt{n\tilde{C}\log(n/\delta)}}{\frac{\tau}{C_1\sqrt{\lambda_1}}(\sqrt{d} - \sqrt{n-1})} \\
&\leq \mathcal{O}(\sqrt{\log(n/\delta)}).
\end{aligned}
\tag{33}
$$

Now we begin to bound the distance $\left\| u_r^{(t)} - u_r^{(0)} \right\|$ for each $r \in [m]$.

Notice that

$$
\begin{aligned}
\left\| \frac{du_r^{(t)}}{dt} \right\| &= \left\| \nabla_{u_r^{(t)}} \mathcal{L}_n(w^{(t)}) \right\| \\
&= \left\| \frac{1}{n} \sum_{i=1}^n \left( f(w^{(t)}, x_i) - y_i \right) v_r \phi'(u_r^{(t)} x_i) x_i \right\| \\
&\leq \frac{1}{n\sqrt{m}} \sum_{i=1}^n \left| f(w^{(t)}, x_i) - y_i \right| \\
&\leq \frac{1}{\sqrt{nm}} \left\| f(w^{(t)}, \mathcal{X}) - \mathcal{Y} \right\| \\
&\leq \sqrt{\frac{2\mathcal{L}_n(w^{(0)})}{m}} \exp\left( -\lambda_{\min}(\widehat{\Theta}) t / 2n \right).
\end{aligned}
$$

Hence,

$$
\left\| u_r^{(t)} - u_r^{(0)} \right\| \leq \int_0^t \left\| \frac{du_r^{(s)}}{ds} \right\| ds \leq \frac{2n}{\lambda_{\min}(\widehat{\Theta})} \sqrt{\frac{2\mathcal{L}_n(w^{(0)})}{m}} \leq \sqrt{\frac{2n}{m\lambda_{\min}(\widehat{\Theta})}} r_{n,\delta,\varepsilon}.
$$

Now we apply Lemma E.13 with $B = r_{n,\delta,\varepsilon}, R = \sqrt{\frac{2n}{m\lambda_{\min}(\widehat{\Theta})}} r_{n,\delta,\varepsilon}$, we get with probability at least $1 - \delta/12$ over the random initalization,

$$
\mathcal{R}_S(\mathcal{F}) \leq \frac{B}{\sqrt{2n}} \left( 1 + \left( \frac{2\log(24/\delta)}{m} \right)^{1/4} \right) + 2R^2\sqrt{md} + R\sqrt{2\log(24/\delta)}.
$$

Then by equation (33), if $m \geq \text{poly}\left(n, \lambda_{\min}^{-1}(\widehat{\Theta}), \delta^{-1}\right)$, we have with probability at least $1 - \tau^{d-n+1} - \tau^d - \delta/4$ over the samples and random initialization,

$$
\mathcal{R}_S(\mathcal{F}) \leq \mathcal{O}\left( \sqrt{\frac{\log(n/\delta)}{n}} \right).
$$

Finally, by Lemma D.2, we have with probability at least $1 - \tau^{d-n+1} - \tau^d - \delta$ over the samples and random initialization,

$$
\mathbb{E}_{(x,y)\sim\mathcal{D}} \left[ \tilde{\ell}\left( f(w^{(\infty)}, x), y \right) \right] \leq \mathcal{O}\left( \sqrt{\frac{\log(n/\delta)}{n}} \right),
$$

for some constant $\tau \in (0, 1)$ that depend only on the subgaussian moment of the entries.

This completes the proof.

$\square$

**Comparison.** This result is related to Arora et al. (2019), which gave an NTK-based generalization bound for overparameterized two-layer ReLU neural networks. This result matches with theirs in the sense that we discover some underlying functions that are provably learnable. Examples of learnable target functions in Arora et al. (2019) include polynomials $y = (\beta^\top x)^p$, non-linear activations $y = \cos(\beta^\top x) - 1$, $y = \tilde{\phi}(\beta^\top x)$ with $\tilde{\phi}(z) = z \cdot \arctan(z/2)$, $\|\beta\| \leq 1$. Our result, furthermore, expands the target function class that is provably learnable since we only require $\tilde{\phi}$ to be Lipshcitz. In addition, they set the standard deviation of the initialization to be at most $\mathcal{O}(1/n)$, whereas we use a different initialization with order $\mathcal{O}(1/\sqrt{d})$ that is more often applied in practice. This result is also related to Liu et al. (2020), which proved that overparameterized deep neural networks satisfy the PL condition. Further, we extend this work by analyzing the generalization.

