# OpenReview forum: "From Optimization Dynamics to Generalization Bounds via Łojasiewicz Gradient Inequality"
_TMLR — Accepted by TMLR_

### Review · Reviewer_47yc · 2022-07-27

**Summary Of Contributions:**

This paper proposes a general framework to study the optimization and generalization of machine learning problems that satisfy the so-called Uniform-LGI. Accordingly, the authors develop the optimization and generalization guarantees for linear regression, kernel regression, and two-layer neural network models. Numerical simulations support the theoretical presumptions (i.e., Uniform-LGI).

**Broader Impact Concerns:**

No broader impact concerns.

**Requested Changes:**

See the weaknesses section.

**Strengths And Weaknesses:**

Strengths:

* This paper is clearly written and the theoretical results are well presented and evaluated by experiments.
* The developed finite sample test for Uniform-LGI can be applied to measure other useful metrics.
* This paper develops rigorous optimization and generalization guarantees for the problems under the uniform-LGI framework, which is new and seems to complete the prior works on some particular problems (e.g., linear regression with square loss, NTK with square loss).

Weakness:

* One weakness is that this paper cannot handle the case of $\theta=1$, which corresponds to the classification problem with cross-entropy loss or hinge loss. Under this setting, the developed generalization bound will explode as there is a term $1-\theta_n$ in the denominator.

* Besides, In theorem 3.3 (and the discussion afterward), the parameter $\\|L_{\psi}(S_{a,b})\\|$ can be super large when the model is complicated. The authors only discuss the case of the linear model to mention that $\\|L_{\psi}(S_{a,b})\\|$ is a constant, it would be necessary to consider a more complicated model (i.e., deep linear model).

* The authors only consider gradient flow, which is indeed a simpler version of the standard gradient descent algorithm. A detailed discussion on the extension to gradient descent should be added.

* After Theorem 4.2, the authors mention that this paper also uncovers some scenarios for benign overfitting in linear regression. However, I cannot see this from Theorem 4.2 since the authors do not give any discussion about when benign overfitting (small training error and small test error for over-parameterized linear model) will occur.

* Note that the constant $c_{n,\delta}$ and $\theta_{n,\delta}$ depend on the sample size $n$, then the developed generalization error bound can still explode if these two parameters have some bad dependency on $n$. The authors may need to discuss this point and perform experiments to reveal the relationship between the parameters and sample size.

* Lastly, in practice, the regularization will be involved during the training, can the developed framework cover the training on the regularized empirical risk?

---

> ### Author Response · Authors · 2022-08-14
> **Authors' response**
>
> 1. ''This paper cannot handle the case of $\theta = 1$, which corresponds to the classification problem with cross-entropy loss or hinge loss. Under this setting, the generalization bound will explode.''
>    * We believe there is a misunderstanding on this point. The classic LGI states that for any subanalytic function [1] including cross-entropy loss and hinge loss, *there exists* a $\theta$ that is strictly less than $1$ such that the Uniform-LGI holds on a neighborhood of a stationary point. Thus, there is no ''nice'' function that can achieve $\theta = 1$. For example, for the losses of the type $w^{2k}$,  $k$ has to go to infinity for $\theta = 1$. Second, our numerical results (Figure 2 and Table 1) also suggest that $\theta<1$ for classification problems with cross-entropy loss.
>        [1] Bolte, Jérôme, Aris Daniilidis, and Adrian Lewis. "The Łojasiewicz inequality for nonsmooth subanalytic functions with applications to subgradient dynamical systems." *SIAM Journal on Optimization* 17.4 (2007): 1205-1223.
>
>       **We have added the above discussion after the Uniform-LGI definition 3.1.**
>
> 2. ''In theorem 3.3 (and the discussion afterward), the parameter $\|L_\Psi(S_{a, b})\|$ can be super large when the model is complicated. The authors only discuss the case of the linear model to mention that $\|L_\Psi(S_{a, b})\|$ is a constant, it would be necessary to consider a more complicated model (i.e., deep linear model).''
>       * In fact, the parameter $\sup_{\|a\|^2 + \|b\|^2 \leq 2 r^2} \|L_{\Psi} (S_{a, b})\|$ not only depends on the model architecture, but also depends on $r$. For practical models, this constant can be small if $r$ is not large. For example, for two-layer neural networks, we proved that $\sup_{\left\|a\right\|^2 + \left\|b\right\|^2 \leq 2 r^2} \left\|L_\Psi (\mathcal{S}_{a, b})\right\| \leq \sqrt{2} r$.
>
>         **In light of your comments, we have added the case of the two-layer neural network model at the discussion of Theorem 3.3.**
>
> 3. ''The authors only consider gradient flow, which is indeed a simpler version of the standard gradient descent algorithm. A detailed discussion on the extension to gradient descent should be added.''
>       * We would like to point out that only the optimization result (Theorem 3.2) relies on the gradient flow setup. The gradient flow is only used to bound the path distance, which then can be used to obtain the generalization bound (Theorem 3.3). We can also do the same with gradient descent.
>
>         **We have added the discrete time analysis on the optimization in the Appendix C of the revised paper. The generalization result can be directly obtained by substituting the gradient descent distance bound for $r_{n, \delta, \varepsilon}$ .**
>
> 4. ''After Theorem 4.2, the authors mention that this paper also uncovers some scenarios for benign overfitting in linear regression. However, I cannot see this from Theorem 4.2 since the authors do not give any discussion about when benign overfitting (small training error and small test error for over-parameterized linear model) will occur.''
>     * In Theorem 4.2, if we set $\varepsilon = 0$ (zero training error), then we can see that the benign overfitting occurs when $\lambda_0$ and $\lambda_1$ have the same order,  where $\lambda_0$ and $\lambda_1$ stand for the smallest and largest eigenvalue of $\Sigma^{-1}$ in the case of i.i.d. Gaussian data $\mathcal{N}(0, \Sigma)$. The generalization bound becomes $\mathcal{O} \left( \sqrt{\log(1/\delta)/n}\right)$.
>
>       **We have added a paragraph detailing this after Theorem 4.2.**
>
> 5. ''Note that the constant $c_{n, \delta}$ and $\theta_{n, \delta}$ depend on the sample size $n$, then the developed generalization error bound can still explode if these two parameters have some bad dependency on $n$. The authors may need to discuss this point and perform experiments to reveal the relationship between the parameters and sample size.''
>      * **We have provided new supporting experiments to reveal the relationship between the Uniform-LGI constants, the generalization error bound and the sample size after Theorem 4.7.**
>
>         Specifically, we perform experiments on the CIFAR10 dataset (first two classes).
>         As shown in Figure 3(a), the Uniform-LGI constants $c_{n, \delta}$ and $\theta_{n, \delta}$ have good dependence on $n$.
>         In Figure 3(b), we show that the generalization bound (up to some constant) $c_{n, \delta}^{-2} (1 - \theta_{n, \delta})^{-2} n^{-1/2} + \sqrt{m / n}$ decays as $n$ increases, indicating that our bound is non-vacuous and can effectively capture the test error.
>
>     *  Meanwhile, even though that  *theoretically* estimating the Uniform-LGI  constants are difficult in general, there are some special cases where this can be done, e.g. the overparameterized case. In our paper (Appendix E.5), we have theoretically showed that for overparameterized two-layer neural networks, $\theta = 1/2$ and $c \sim \mathcal{O}(1)$.

---

> > ### Author Response · Authors · 2022-08-14
> > **Authors' response**
> >
> > 6. ''Lastly, in practice, the regularization will be involved during the training, can the developed framework cover the training on the regularized empirical risk?''
> >
> >       * In principle, it is possible to apply the framework to the regularized empirical risk as long as the Uniform-LGI constants are known.
> >         However, this is not the core scope of the current work.
> >         The current work only studies implicit regularization based on early stopping.

---

> > ### Comment · Reviewer_47yc · 2022-08-28
> > **To the authors' response.**
> >
> > 1. It is true that if the loss function is of the form $w^{2k}$, then $\theta=1$ only when $k\rightarrow \infty$. However, the cross-entropy loss, or exponential loss, is of the form $e^{-w}$ when $w$ is large, then it is clear that $\theta=1$ in this case (consider one single data and use exponential loss).
> >
> > 2. I am a bit curious about the benign overfitting result. From Theorem 3.3 it seems that the generalization error can be small if $\lambda_0$ and $\lambda_1$ are in the same order. However, if the condition number of the data covariance has such a flat eigenspectrum, then why one can achieve a good generalization error in the over-parameterized setting ($d>n$) as one cannot learn all dimensions using only $n$ training data.

---

> > > ### Author Response · Authors · 2022-09-01
> > > **Author's response**
> > >
> > > 1. ''It is true that if the loss function is of the form $w^{2k}$, then $\theta=1$ only when $k \xrightarrow{} \infty$. However, the cross-entropy loss, or exponential loss, is of the form $e^{-w}$ when $w$ is large, then it is clear that $\theta=1$ in this case (consider one single data and use exponential loss).''
> > >
> > >       * We thank the reviewer for clarifying their previous statement.
> > >         We agree that for this example, if we consider the whole optimization path which diverges to the infinity (as the global minimum is $\infty$), then $\theta = 1$, and the generalization bound will explode. Such cases (where stationary point is not finite) are not covered in our framework. Besides, in the cases where this occurs we may add a smooth weight regularizer, which will result in a finite minimum and thus a theta < 1. Handling the cases without a finite minimum is out of the scope of the current paper.
> > >         For the common case when there exists a stationary point in a *compact set*, the classic LGI indicates that $\theta$ can be strictly less than $1$.
> > >
> > >
> > > 2. ''I am a bit curious about the benign overfitting result. From Theorem 3.3 it seems that the generalization error can be small if $\lambda_0$ and $\lambda_1$ are in the same order. However, if the condition number of the data covariance has such a flat eigenspectrum, then why one can achieve a good generalization error in the over-parameterized setting as one cannot learn all dimensions using only $n$ training data.''
> > >
> > >      * This is an interesting question. First, we summarize the different settings and assumptions in our paper and [1] through an example: the training data $\{(x_i, y_i)\}_{i=1}^n \subset \mathbb{R}^d \times \mathbb{R}$ are i.i.d. drawn from $\mathcal{D}$.
> > >
> > >        In our case, for any $(x, y) \in \mathcal{D}$, $ y = x^\top w^*$ for some $w^* \in \mathbb{R}^d$ with uniformly bounded norm in $d$, and entries of $x$ are i.i.d. (thus $\lambda_0 = \lambda_1 = 1$) with mean $0$, variance $\Sigma = \mathbb{E}[x x^\top]$ (diagonal). Thus $\Sigma$ has a flat spectrum with condition number $1$. We further assume that $\|x\| \leq 1$ for all $d$. Let $\mathcal{X} = (x_1, \ldots, x_n)^\top \in \mathbb{R}^{n \times d}, \mathcal{Y} = (y_1, \ldots, y_n) \in \mathbb{R}^n$, then the minimum norm solution is given by $\hat{w}=\mathcal{X}^{\dagger} \mathcal{Y}$ ($\dagger$ is the pseudo inverse). Let the excess risk $E = \mathbb{E}_x[(x^\top \hat{w} - x^\top w^*)^2]$, then our result (Theorem 3.3) states that $E = \mathcal{O}(1/\sqrt{n})$ given that $d$ and $n$ diverge but their ratio remains finite.
> > >
> > >        Now we consider a new data set $(\tilde{x}_i, \tilde{y}_i) \in \mathbb{R}^d \times \mathbb{R}$ that are i.i.d. drawn from $\tilde{\mathcal{D}}$. In [1], the setting is that for any $(\tilde{x}, \tilde{y}) \in \tilde{\mathcal{D}}$, $\tilde{y} = \tilde{x}^\top \tilde{w}^*$ for some $\tilde{w}^* \in \mathbb{R}^d$ with uniformly bounded norm in $d$, and entries of $\tilde{x}$ are independent with mean $0$, variance $\tilde{\Sigma} = \mathbb{E}[\tilde{x} \tilde{x}^\top]$. If the entries of $\tilde{x}$ are independent, then $\tilde{\Sigma}$ is diagonal. Let the minimum norm solution w.r.t. the new data set $\tilde{w} = \tilde{\mathcal{X}}^{\dagger} \tilde{\mathcal{Y}}$, then it is shown in [1] that when $\tilde{\Sigma}$ has a suitable decaying eigenspectrum, the excess risk
> > >       $\tilde{E} = \mathbb{E}_{\tilde{x}} [(\tilde{x}^\top \tilde{w} - \tilde{x}^\top \tilde{w}^*)^2] \xrightarrow{} 0$ when $n \xrightarrow{} \infty$.
> > >
> > >       Therefore, we can see that there exist two cases for benign overfitting: the data covariance matrix has a decaying eigenspectrum ([1]); the data covariance matrix has a flat eigenspectrum but each data vector has a bounded norm (our paper). Due to the rescaling of the data vector, even though $\Sigma$ has a flat eigenspectrum, the benign overfitting provably happens when $\hat{w}$ has a bounded norm.
> > >
> > >      * To have a better understanding of the benign overfitting phenomenon when the data covariance matrix has a flat eigenspectrum, we consider an example in [1] that we can deduce from our result. Under the same notation as above, we consider the following data transformation for any $(x, y) \in \mathcal{D}$: $\tilde{x} = \tilde{\Sigma}^{1/2} x \sqrt{d}$, $\tilde{w}^* = \frac{\tilde{\Sigma}^{-1/2} w^* / \sqrt{d}}{\|\tilde{\Sigma}^{-1/2} w^* / \sqrt{d}\|}$, $\tilde{y} = \frac{y}{\|\tilde{\Sigma}^{-1/2} w^* / \sqrt{d}\|}$. Then the induced new data distribution $\tilde{\mathcal{D}}$ satisfies the assumptions and settings in [1]. Note that $\tilde{\mathcal{X}} = \mathcal{X} \tilde{\Sigma}^{1/2} \sqrt{d}, \tilde{\mathcal{Y}} = \frac{\mathcal{Y}}{\|\tilde{\Sigma}^{-1/2} w^* / \sqrt{d}\|}$, thus the minimum norm solution $\tilde{w} = \tilde{\mathcal{X}}^{\dagger} \tilde{\mathcal{Y}} = \frac{\tilde{\Sigma}^{-1/2} \hat{w}}{\|\tilde{\Sigma}^{-1/2} w^*\|}$.

---

> > > > ### Author Response · Authors · 2022-09-01
> > > > **Author's response**
> > > >
> > > > Continuation of the answer to Question 2.
> > > >
> > > >    * Therefore, the excess risk $\tilde{E} = \mathbb{E}_{\tilde{x}} [(\tilde{x}^\top \tilde{w} - \tilde{x}^\top \tilde{w}^*)^2] = \frac{E}{\|\tilde{\Sigma}^{-1/2} w^* / \sqrt{d}\|^2}$.
> > > >
> > > >      We can see that there exists an explicit relation between $\tilde{E}$ and $E$.
> > > >             In the high dimension setting ($d$ and $n$ diverge but have a finite ratio), our result shows that $E = \mathcal{O}(1/\sqrt{d})$.
> > > >             We consider a benign overfitting example in [1, Part 1 of Theorem 6]: the eigenvalues of $\tilde{\Sigma}$ decay with a rate given by $\sigma_j = j^{-1} \log^{-\beta}(j+1)$ with $\beta>1$.
> > > >             Now we show that $\tilde{E} \xrightarrow{} 0$ when entries of $w^*$ are order $1/\sqrt{d}$.
> > > >
> > > >      Notice that $\frac{E}{\|\tilde{\Sigma}^{-1/2} w^* / \sqrt{d}\|^2} \sim \frac{d \sqrt{d}}{\sigma_1^{-1} + \ldots + \sigma_d^{-1}} = \frac{d \sqrt{d}}{\sum_{j=1}^d j \log^{\beta}(j+1)} < \frac{d \sqrt{d}}{\sum_{j=2}^d j} \xrightarrow{} 0$.
> > > >
> > > >      In summary, we can transform our assumptions and settings to analyze the case of sharp eigenspectrum for benign overfitting, and get a result that is consistent with [1].
> > > >
> > > > **In light of your comments, we have added the above discussion after Theorem 3.3.**
> > > >
> > > > ---------------------------------------------------------------------------------------------------
> > > >
> > > > [1] Bartlett, Peter L., et al. "Benign overfitting in linear regression." Proceedings of the National Academy of Sciences 117.48 (2020): 30063-30070.

---

### Review · Reviewer_2YTV · 2022-08-04

**Summary Of Contributions:**

This paper studies the uniform Łojasiewicz Gradient Inequality (LIG) on the loss function of several machine learning models, as well as the implication of the LIG property. In the first part, the LIG is defined in global and local sense. An algorithm is proposed to check the LIG property on the trajectory of gradient descent, using points on the trajectory. The algorithm depends on a linear regression to obtain the \theta parameter in the LIG. The algorithm is applied on several synthetic and realistic problems, showing its ability to get an estimate of the \theta and c in the LIG. In the second part, based on the assumption that the uniform LIG holds,  the convergence and generalization performance of machine learning models are studied. The convergence results are derived from the LIG, similar to that from the PL-condition. Together with the convergence, the displacement of the parameters from the initialization is also bounded, which helps derive a generalization bounds by controlling the complexity of the Hypothesis space in a ball in the parameter space. The generalization bounds are studied in more detail for several specific models, including linear models and two-layer neural networks.

**Broader Impact Concerns:**

The reviewer does not have broader impact concerns.

**Requested Changes:**

Most problems are listed in the "weakness" part above. The reviewer hope the authors could address the questions there. Some other questions are listed in the following:

1. In page 7, below Theorem 3.3, the authors mentioned "there are 6 key terms in (5)", but later discussion only mentioned 4.

2. What is the \tilde{M}_\delta in Theorem 3.3 and later theorems? It seems its a bound for the empirical risk when the time tends to infinity. However, for gradient flow the empirical risk is always decreasing. Why we need this \tilde{M}_\delta?

3. In Theorem 3.3, the first term on the right hand side is \epsilon*L. However, in Theorem 4.2, 4.4, and 4.7, they are all square root of \epsilon*L. What is the reason of this mismatch in the theory.

4. In the introduction, the authors say "Therefore, the connection between optimization and generalization still remains largely misunderstood in general scenarios". The reviewer does not see any misunderstanding in the connection between optimization and generalization. There is just not enough understanding.

**Strengths And Weaknesses:**

Strengths:

1. This paper takes a rather general approach to study the generalization of machine learning models. The results are not derived using very specific structures of the model, but for a general class of models that satisfies a certain property (the LIG). Discussions in section 4 shows that the analysis framework can be applied to different models.

2. Optimization is taken into consideration during the study of generalization. This is important especially for over-parameterized models, for which the implicit regularization is crucial for the model to generalize.


Weakness:

1. The reviewer does not see the different of the formulation of the LIG with the well-known PL-condition. The way to bound the convergence rate and parameter displacement is also the same as what people did with the PL condition. In this sense, this work does not derive new theoretical tools for the analysis of machine learning models. The reviewer hopes the authors can provide a detailed explanation if the above understanding is wrong.

2. The Algorithm 1 used to estimate the parameters in the LIG does not have theoretical guarantee. Hence, the numerical experiments do not prove that the test loss functions satisfy the LIG. Also, the estimated \theta and c might not be the real \theta and c. They might not even be close given the lack of theoretical results. If theoretical analysis is hard, the authors should at least provide some discussion on the intuition that why this algorithm may work and find the correct \theta and c.

3. Still in Algorithm 1. The reviewer does not understand why a linear regression is used to compute the \theta. In my understanding, the \theta here should work in the worst case, i.e. the same LIG should hold for all parameters along the GD trajectory. Hence, the most straightforward method should be directly checking the \theta and c satisfied by each point on the GD trajectory, and then taking a maximum. The linear regression obviously does not catch the worst case, and the resulting \theta is large enough for some points, while not large enough for the other points. Therefore, it cannot be used as the \theta in the uniform LIG.

4. The results in section 4.3 for two-layer neural networks are not convincing. The proposition 4.5 is only an existence result. This is why its proof is quite simple and does not use any special structure of the neural networks. But the problem is we do not have any control on the c_n and \theta_n. They may depend on n, giving vacuous bounds. For the generalization bound here to make sense, the authors need to give an estimate of these values and study their dependency with n.

---

> ### Author Response · Authors · 2022-08-14
> **Authors' response**
>
> 1. ''The reviewer does not see the different of the formulation of the LIG with the well-known PL-condition. The way to bound the convergence rate and parameter displacement is also the same as what people did with the PL condition. In this sense, this work does not derive new theoretical tools for the analysis of machine learning models.''
>      * First, we would like to point out that the Uniform-LGI is a more general condition than the well-known PL-condition in the sense that (1) The $\mu$-PL condition is a special case for the Uniform-LGI when $c = \sqrt{2\mu}, \theta = 1/2$. The Uniform-LGI contains a wider class of functions than the PL-condition. For instance, in the paper, we show in Figure 1(a), 1(b), Figure 2(a), 2(b), 2(c) that many nontrivial setups do not have $\theta = 1/2$. (2) The well-known PL-condition is defined in a neighborhood of a *global* minimum, indicating that ever stationary point is a global minimum, while the Uniform-LGI is more general, and is defined in an arbitrary set, where there may exist no global minimum. Therefore, the Uniform-LGI does not imply that any stationary point is a global minimum. Hence, the Uniform-LGI covers more scenarios than the PL-condition when analyzing optimization properties since global minimum is usually hard to reach by an optimization algorithm in practice.
>
>         **To make the comparison more clear, we have added the above discussion after the Definition 3.1.**
>
>      * Second, we highlight the new theoretical tools as follows: (1) We introduce the Uniform-LGI condition as above, that can handle more general problems where the PL condition fails. For example, with the Uniform-LGI, we can analyze the optimization and  generalization of models at any training time while with the PL condition, this cannot be achieved because the condition focuses on a local region around a global minimum. (2) Our framework based on the Uniform-LGI is a new tool that yield new generalization results by combining optimization and generalization. For example, for linear/kernel regression and two-layer neural networks, our results match or extend previous results by exhibiting bias-variance tradeoff pattern.
>
> 2. ''The Algorithm 1 used to estimate the parameters in the LIG does not have theoretical guarantee. Hence, the numerical experiments do not prove that the test loss functions satisfy the LIG. Also, the estimated $\theta$ and $c$ might not be the real $\theta$ and $c$. They might not even be close given the lack of theoretical results. If theoretical analysis is hard, the authors should at least provide some discussion on the intuition that why this algorithm may work and find the correct $\theta$ and $c$.''
>     * First, we point out that the Uniform-LGI is an inequality, thus the pair of the Uniform-LGI $(c, \theta)$ such that this inequality holds is not unique. While numerical experiments can never *prove* an inequality, it is useful in estimating the constants by looking at the limiting values as the number of points increases. This is called finite-size scaling analysis [1] and is frequently used in fields such as statistical physics to investigate numerically the behaviour of infinite-size systems (e.g. phase transitions). Algorithm 1 is designed to approximately obtain *one* possible pair of $(c, \theta)$ such that the Uniform-LGI holds during the training process (usually infinitely many data points).
> To get *one* possible $(c, \theta)$ by using only a *finite* number of data points, we use Algorithm 1 to get $\theta$ first by fitting the first $k$ collected data points with linear regression, then choose $c$ as the *maximum* possible constant such that the Uniform-LGI holds for these first $k$ collected data points, i.e., $c_k := \min_{i \in [0 : k-1]} \frac{\left\|\nabla \mathcal{L} (w^{(i)})\right\|}{\left(\mathcal{L} (w^{(i)}) - \mathcal{L} (w^*) \right)^{\theta_k}}$. Then, $(c_k, \theta_k)$ is one possible pair of constants such that the Uniform-LGI holds on the first $k$ data points. Ideally, as $k$ increasing to infinity, if $c_k, \theta_k$ converge to $c^*, \theta^*$, then we can safely say that the function $\mathcal{L}(w)$ satisfies the Uniform-LGI on the set $\{w^{(0)}, w^{(1)}, \ldots, w^{(\infty)}\}$ with $(c^*, \theta^*)$.
>     * Second, the numerical verification of the Uniform-LGI on the optimization path induced by the test loss function *cannot* be done unless we know the distribution of the training samples. But we have theoretically proved the Uniform-LGI in Proposition 4.5 given that the test loss function is subanalytic, which is a general condition and holds for various loss functions, e.g., squared loss and cross-entropy loss.
>
>      [1] Privman, Vladimir, ed. Finite size scaling and numerical simulation of statistical systems. *World Scientific*, 1990.

---

> > ### Author Response · Authors · 2022-08-14
> > **Authors' response**
> >
> > 2. Continuation of the answer to Question 2.
> >       * Third, we have provided numerical evidence that the finite size scaling method works as expected for toy examples where we know the Uniform-LGI constants (and the method can also detect cases where Uniform-LGI conditions do not hold). See the synthetic models analysis and Figure 1 in page 5. For the non-analytic model, by the classic LGI property we know that there exists no $\theta$ and $c$ such that the Uniform-LGI holds, which is consistent with Figure 1(a). For the loss function $\mathcal{L}(w) = \frac{1}{4}w^{\frac{4}{3}}+\frac{1}{2} w^2$, the global minimum is at $0$, and $\frac{|\nabla \mathcal{L}(w)|}{\mathcal{L}(w)^\theta} \sim \mathcal{O}(w^{\frac{1-4\theta}{3}})$. Therefore, the Uniform-LGI constant $\theta$ should be at least $1/4$, which is consistent with the result in Figure 1(b) that the estimated $\theta = 0.308$. For the linear regression model, we know that the squared loss is always strongly convex, thus it satisfies the PL-condition ($\theta = 0.5$), which is consistent with Figure 1(c) that the estimated $\theta \approx 0.5$. These synthetic models are evidence that Algorithm 1 can capture the Uniform-LGI constants accurately.
> >
> >          **To make this point more clear, we have added a discussion before and after Algorithm 1.**
> >
> > 3. ''Still in Algorithm 1. The reviewer does not understand why a linear regression is used to compute the $\theta$. In my understanding, the $\theta$ here should work in the worst case, i.e. the same LIG should hold for all parameters along the GD trajectory. Hence, the most straightforward method should be directly checking the $\theta$ and $c$ satisfied by each point on the GD trajectory, and then taking a maximum. The linear regression obviously does not catch the worst case, and the resulting $\theta$ is large enough for some points, while not large enough for the other points. Therefore, it cannot be used as the $\theta$ in the uniform LIG.''
> >     * As we mentioned in the answer of the second question, the pair of  Uniform-LGI constants is not unique in general, and we do not deal with the case of finding numerically an *optimal* pair of constants. To find one possible $(c, \theta)$ such that $\log \|\nabla \mathcal{L} (w)\| \geq \log c + \theta \log \(\mathcal{L} (w) - \min_{v\in \mathcal{S}} \mathcal{L} (v) \)$ holds on $\{w^{(i)}\}_{i=1}^K$, one natural way is to first set $\theta$ and then adjust $c$ such that the inequality holds for the first $K$ points. Therefore, we consider to use linear regression to find $\theta$. The process of adjusting $c$ is actually catching the worst case. As you said, one can also use the maximum $\theta$ as an exponent, but this is *not* a good approach as we can definitely find a smaller $\theta$ if Algorithm 1 outputs a convergent sequence of the estimated Uniform-LGI constants. Smaller $\theta$ is better than a larger one as we have shown in the paper (Theorem 3.2 \& 3.3) that smaller $\theta$ yields better optimization and generalization. Moreover, Algorithm 1 is validated by our synthetic models for which we can *provably* estimate the Uniform-LGI constants (see the third points of the answer for the second question), showing that it can be used to obtain the Uniform-LGI constants.
> >
> >       **To make this point more clear, we have added this explanation before and after Algorithm 1.**
> >
> > 4. ''The results in section 4.3 for two-layer neural networks are not convincing. The proposition 4.5 is only an existence result. This is why its proof is quite simple and does not use any special structure of the neural networks. But the problem is we do not have any control on the $c_n$ and $\theta_n$. They may depend on $n$, giving vacuous bounds. For the generalization bound here to make sense, the authors need to give an estimate of these values and study their dependency with $n$.''
> >      * **In light of your comments,
> >         we have provided new supporting experiments to reveal the relationship between the Uniform-LGI constants, the generalization error bound and the sample size after Theorem 4.7.**
> >
> >          Specifically, we perform experiments on the CIFAR10 dataset (first two classes).
> >         As shown in Figure 3(a), the Uniform-LGI constants $c_{n, \delta}$ and $\theta_{n, \delta}$ do not explode with $n$.
> >         In Figure 3(b), we show that the generalization bound (up to some constant) $c_{n, \delta}^{-2} (1 - \theta_{n, \delta})^{-2} n^{-1/2} + \sqrt{m / n}$ decays as $n$ increases, indicating that our bound is non-vacuous and can effectively capture the test error.
> >
> >       * Meanwhile, even though that  \textit{theoretically} estimating the Uniform-LGI  constants are difficult in general, there are some special cases where this can be done, e.g. the overparameterized case.
> >         In our paper (Appendix E.5), we have theoretically showed that for overparameterized (large width) two-layer neural networks, $\theta = 1/2$ and $c \sim \mathcal{O}(1)$.

---

> > > ### Author Response · Authors · 2022-08-14
> > > **Authors' response**
> > >
> > > 5. ''In page 7, below Theorem 3.3, the authors mentioned "there are 6 key terms in (5)", but later discussion only mentioned 4.''
> > >
> > >    * Actually we mentioned $6$ terms: (1) $M_\delta$ (2) $\theta_n, c_n$ (3) $L_{\ell}(S_{a, b}), M_{a, b}$ (4) $\|L_\Psi (\mathcal{S}_{a, b})\|$. There are totally $6$ terms.
> > >
> > >      **To avoid misunderstanding, we change the statement to ``several terms''.**
> > >
> > >
> > > 6. ''What is the $\tilde{M}_\delta$ in Theorem 3.3 and later theorems? It seems its a bound for the empirical risk when the time tends to infinity. However, for gradient flow the empirical risk is always decreasing. Why we need this $\tilde{M}_\delta$?''
> > >
> > >
> > >     * You can think of $\bar{M}_\delta$ as the empirical risk when the time tends to infinity.
> > > We need this to bound the path length $\int_0^{T} \|\frac{d w^{(t)}}{d t}\|d t$, which is equal to $\frac{ \left(L_n(w^{(0)}) -  L_n (w^{(\infty)})\right)^{1 - \theta_{n}} - \left(L_n(w^{(T)}) -  L_n (w^{(\infty)})\right)^{1 - \theta_{n}}}{c_{n}(1 - \theta_{n})}$. We use $\bar{M}_\delta$ to get a path length estimate over the training sample.
> > > We would like to highlight that the results deal with the case where $\bar{M}_\delta$ is not $0$, which is always the case for non-overparameterized setting where the optimization algorithm can only find a local minimum in training.
> > >
> > >
> > > 7. ''In Theorem 3.3, the first term on the right hand side is $\varepsilon L$. However, in Theorem 4.2, 4.4, and 4.7, they are all square root of $\varepsilon L$. What is the reason of this mismatch in the theory.''
> > >
> > >
> > >      * The reason is that in the application part, we consider the population risk with respect to a new loss $\tilde{\ell}$ that is $1$-Lipschitz.
> > >         In that case, the loss value $\frac{1}{n} \sum_{i=1}^n \tilde{\ell}(f(w_{\varepsilon}, x_i), y_i)$ w.r.t. $\tilde{\ell}$ can be bounded above by $\sqrt{\varepsilon \mathcal{L}}$ (see the inequality (22) in Appendix E.1). Hence, there will be a square root in the generalization bound.
> > >         This setting is to simplify the notation. We have mentioned before Application 4.1 that $\tilde{\ell}$ can also be replaced by the original $\ell$ because most of the loss functions are locally Lipschitz and locally bounded.
> > >
> > >
> > > 8. ''In the introduction, the authors say "Therefore, the connection between optimization and generalization still remains largely misunderstood in general scenarios". The reviewer does not see any misunderstanding in the connection between optimization and generalization. There is just not enough understanding.''
> > >
> > >
> > >       * We agree with the reviewer that this is not enough understanding and this is what we want to express.
> > >         We have edited this sentence.

---

> ### Comment · Reviewer_2YTV · 2022-08-22
> **Reply to the authors' reponse**
>
> Thanks for the detailed responses provided by the authors! The following are my replies and follow-ups:
>
> 1. The reviewer agrees that the most traditional PL-condition only considers theta=1/2. However, more general PL-conditions have been discussed and studied a lot. For example, see [1]. Does the analysis in this paper cover new settings or provide new results?
>
> 2. Thanks for the response. Now I'm convinced that the algorithm is useful and gives trustable results.
>
> 3. Same as 2
>
> 4. I agree that in practice the c and theta might not increase with n, in which case the following analysis on the convergence/generalization works. However, that needs to be verified numerically case by case. The theorem is still not informative. But the reviewer acknowledge that a theoretical estimate for c and theta for practical neural networks can be very hard. I do not take this problem as a major point to criticize the contribution of this paper.
>
> [1] Frei, Spencer, and Quanquan Gu. "Proxy convexity: A unified framework for the analysis of neural networks trained by gradient descent." Advances in Neural Information Processing Systems 34 (2021): 7937-7949.

---

> > ### Author Response · Authors · 2022-08-28
> > **Authors' reponse**
> >
> > We thank the reviewer for pointing out the related work [1], which also studies optimization and generalization via general PL-conditions.
> >         Here we would like to emphasize the differences in the key assumptions, problem settings, and type of results.
> >
> > In [1], the authors analyze a loss function $f$ that satisfies the $(g, \xi, \alpha, \mu)$-proxy PL inequality, which is a more general definition than the LGI in this paper.
> >         For example, by setting $g = f, \xi = \min_w f(w)$, the proxy PL inequality corresponds to the LGI.
> >         However, a key assumption in [1] is that $f(w)$ satisfies a $(g, \xi, \alpha, \mu)$-proxy PL inequality *globally* for all $w \in \mathbb{R}^p$, while our assumption only requires that $f(w)$ satisfies the LGI *locally*, e.g., along the optimization path.
> >         For the problem settings, we cover different situations. [1]
> >         considers the online learning setting where samples are observed one-by-one. In contrast, we focus on the classic gradient flow/descent setting, where the training samples are given before training. Thus, these two works provide different types of results. Through the global proxy-PL inequality, [1] gives an upper bound for the \textit{best case} of the test error, i.e., $\min_{t < T} \mathcal{L}_\mathcal{D}(w^{(t)})$. In this work, we obtain bounds for the test error $\mathcal{L}_\mathcal{D}(w^{(t)})$ for any time $t$, which can be used to showcase bias-variance trade-off patterns. For instance, a simple application of our theory yields a new case of benign overfitting on linear/kernel regression in the high dimensional setting (Theorem 4.3 \& 4.5).
> >         Moreover, in this work, we propose a finite sample test algorithm that can be applied to verfiy the Uniform-LGI and estimate the Uniform-LGI constants for standard deep learning settings.
> >
> >
> > In addition to [1], we found two other existing works [2, 3] on the general PL conditions.
> >
> > In [2], the key assumption is that the \textit{population risk} $\mathcal{L}_\mathcal{D}(w)$ satisfies the LGI on some given parameter space $\mathcal{W}$. Then it is shown in [2] that the excess risk
> >
> > $L_\mathcal{D}(w) - \min_{w \in \mathcal{W}} L_\mathcal{D}(w)$
> > is bounded above by $\sup_{w \in \mathcal{W}} \|\nabla L_n(w) - \nabla L_\mathcal{D}(w)\|$.
> >         In practice, it is not easy to validate whether $\mathcal{L}_\mathcal{D}$ satisfies the LGI because it depends on the data distribution and requires a very large number of samples.
> >         In comparison, we assume that the \textit{empirical risk} $\mathcal{L}_n(w)$ satisfies the LGI along the training path, and the Uniform-LGI condition can be numerically checked by the finite sample test algorithm.
> >         Under these two different assumptions, both our results and those of [2] yield consistent generalization bounds $\mathcal{O}(1/\sqrt{n})$ for the linear regression models in the high-dimensional setting.
> >
> >  [3] considers a local PL condition (a particular case of our Uniform-LGI with $\theta = 1/2$) in the case of feed-forward neural networks, and show that gradient descent with proper initialization converges to a global minimum given that (1)
> >         the activation functions are smooth and strictly increasing;
> >         (2) the minimum value of the parameters is large enough;
> >         (3) the learning rate is small enough.
> >         In this work, we give both convergence guarantee (to a local minimum) and generalization analysis based on the Uniform-LGI condition.
> >
> >
> > **We have merged the above discussion with the related work section and put it before the conclusion section.**
> >
> > ------------------------------------------------------------------------------------------------------------------------------------
> >
> >
> >
> > [2] Foster, Dylan J., Ayush Sekhari, and Karthik Sridharan. "Uniform convergence of gradients for non-convex learning and optimization." Advances in Neural Information Processing Systems 31 (2018).
> >
> >
> >
> >
> > [3] Chatterjee, Sourav. "Convergence of gradient descent for deep neural networks." arXiv preprint arXiv:2203.16462 (2022).

---

### Review · Reviewer_85er · 2022-08-15

**Summary Of Contributions:**

This paper aims to analyze the interplay between optimization and generalization aspects of training machine learning models. To do this, the analysis is developed based on the Uniform-LGI condition lower-bounding the gradient norm with a power of the distance to the function's optimal value. First, the paper discusses the results of several numerical experiments suggesting that the Uniform-LGI condition indeed holds in standard deep learning experiments. Next, the paper presents Theorems 3.2 and 3.3 bounding the convergence rate of the gradient-flow optimization and the generalization error of the optimized weights. Finally, the paper applies the Uniform-LGI-based convergence and generalization results to standard linear and kernel regression settings as well as one-layer neural nets.

**Broader Impact Concerns:**

Not applicable to this paper

**Requested Changes:**

Based on the weaknesses mentioned above, I request the following changes:

1- The presentation of Theorem 3.3 needs to be improved by making the final sentence clearer and explaining in more detail how Theorems 3.2 and 3.3 can be interpreted together to reveal the connections between optimization and generalization aspects of training machine learning models.

2- Some explanation is needed to justify why $\sqrt{2\epsilon \mathcal{L}_n(\omega^{(0)})}$ appears in Theorems 4.3, 4.4, and 4.7's generalization bounds.

In a standard generalization analysis, one expects the error term to be $\mathcal{L}_{\mathcal{D}}(\omega_\epsilon)-\mathcal{L}_n(\omega_\epsilon)$ which is not the case in the mentioned theorems. The paper should explain why the theorems look different from standard generalization bounds in that sense.

**Strengths And Weaknesses:**

Strengths:

1- The paper is written clearly and the ideas have been presented in the proper order.

2- The paper's numerical results validate the Uniform-LGI assumption for several standard deep learning settings, which can be useful for other deep learning theory studies as well.

3- The Uniform-LGI convergence and generalization results provide a unified theoretical framework for analyzing various machine learning problems.

Weaknesses:

1- The paper's analysis focuses on the gradient flow (Equation 3) instead of the standard gradient descent steps. It is a little unclear whether the analysis can be extended to standard gradient descent algorithms.

2-  Theorem 3.3's presentation needs to be improved. First of all, I do not understand why the authors define $\mathcal{L}_n(\omega_\epsilon) = \epsilon \mathcal{L}_n(\omega^{(0)})$ and then replace $\mathcal{L}_n(\omega_\epsilon)$ with $\epsilon \mathcal{L}_n(\omega^{(0)})$ in the bound in Equation (5).

Why do not the authors just present Equation (5) as a bound on $\mathcal{L}_{\mathcal{D}}(\omega_\epsilon)-\mathcal{L}_n(\omega_\epsilon)$ which is the standard generalization error definition for $\omega_\epsilon$. As I said, the theorem's presentation seems unnecessarily complicated.

3- The last sentence in Theorem 3.3 is vague and needs further clarification. The sentence says "Then for any parameter $\omega_\epsilon$ with $\mathcal{L}_n(\omega_\epsilon) = \epsilon \mathcal{L}_n(\omega^{(0)})$ for some $\epsilon\in[0,1]$, we have with
probability at least $1-\delta$ over $S$, ...".

Does this sentence apply to every $\epsilon\in[0,1]$ or only a specific $\epsilon$ value? Also, does the bound hold uniformly with
probability $1-\delta$ for all $\omega_\epsilon$'s or does it hold for every individual $\omega_\epsilon$ with
probability $1-\delta$? The current presentation of this theorem is somewhat vague and the answer to these questions is unclear.

4- The model assumed in Equation (4) looks inefficient for multilayer neural nets with more than one hidden layer, because it requires including all the neural net weights after layer $1$ as $\beta$ parameters. Also, the number of $\beta$ parameters ($q$) appears in the generalization bound which will be unsatisfactorily large for multilayer neural nets. Is there a way to replace $q$ with the norm of weight matrices for a multilayer neural net?

5- It seems unintuitive to me that $\sqrt{2\epsilon \mathcal{L}_n(\omega^{(0)})}$ appears in the bounds in Theorems 4.3, 4.4, and 4.7.

Should not the point of generalization analysis be to bound  $\mathcal{L}_{\mathcal{D}}(\omega_\epsilon)-\mathcal{L}_n(\omega_\epsilon)$?

Then, why do the authors bound $\mathcal{L}_{\mathcal{D}}(\omega_\epsilon) - \sqrt{2\epsilon \mathcal{L}_n(\omega^{(0)})}$ in Theorems 4.3, 4.4, and 4.7?

---

> ### Author Response · Authors · 2022-08-22
> **Authors' response**
>
> We thank the reviewer for their feedback. It helped us improve the paper. We address hereafter the concerns.
>
> 1. ''The paper's analysis focuses on the gradient flow (Equation 3) instead of the standard gradient descent steps. It is a little unclear whether the analysis can be extended to standard gradient descent algorithms.''
>
>       * The answer is yes.
>         The reason that using gradient flow is to simplify the analysis.
>         We would like to point out that only the optimization result (Theorem 3.2) relies on the gradient flow setup.
>         The gradient flow is only used to bound the path distance, which then can be used to obtain the generalization bound (Theorem 3.3).
>         Similar results hold for gradient descent, which we have incorporated in the revised version.
>
>          **We have added the discrete time analysis on the optimization in the Appendix C of the revised paper.
>         The generalization result can be directly obtained by substituting the gradient descent distance bound for $r_{n, \delta, \varepsilon}$.**
>
>
> 2. 'First of all, I do not understand why the authors define $\mathcal{L}_n(w_\varepsilon) = \varepsilon \mathcal{L}_n(w^{(0)})$ and then replace $\mathcal{L}_n(w_\varepsilon)$ with $\varepsilon \mathcal{L}_n(w^{(0)})$ in the bound in Equation (5). Why do not the authors just present Equation (5) as a bound on $\mathcal{L}_\mathcal{D}(w_\varepsilon) - \mathcal{L}_n(w_\varepsilon)$
>     which is the standard generalization error definition for $w_\varepsilon$.'''
>
>      * The reason we define $\mathcal{L}_n(w_\varepsilon) = \varepsilon \mathcal{L}_n(w^{(0)})$ is because, in order to bound the generalization gap $\mathcal{L}_\mathcal{D}(w^{(t)}) - \mathcal{L}_n(w^{(t)})$, we need an explicit estimate for the distance $\|w^{(0)} - w^{(t)}\|$, which is given in Theorem 3.2: $\|w^{(0)} - w^{(t)}\| \leq \frac{\left(\mathcal{L}_n(w^{(0)}) - \mathcal{L}_n(w^{(\infty)})\right)^{1 - \theta_n} - \left(\mathcal{L}_n(w^{(t)}) - \mathcal{L}_n(w^{(\infty)})\right)^{1 - \theta_n}}{c_n (1 - \theta_n)}.$
>        If we consider the generalization of $w_\varepsilon$ satisfying $\mathcal{L}_n(w_\varepsilon) = \varepsilon \mathcal{L}_n(w^{(0)})$, then the distance $\|w^{(0)} - w_\varepsilon\|$ can be bounded above by $\frac{\left(\mathcal{L}_n(w^{(0)}) - \mathcal{L}_n(w^{(\infty)})\right)^{1 - \theta_n} - \left(\varepsilon \mathcal{L}_n(w^{(0)}) - \mathcal{L}_n(w^{(\infty)})\right)^{1 - \theta_n}}{c_n (1 - \theta_n)}.$  Then, we only need to estimate the initial loss value and the final (after convergence) loss value, which simplifies the notation.
>
>       * We agree with the reviewer that the bound can indeed be presented in the traditional. The main reason for not presenting the bound as $\mathcal{L}_\mathcal{D}(w_\varepsilon) - \mathcal{L}_n(w_\varepsilon)$ is that we want to show the bias-variance tradeoff pattern of the test error $\mathcal{L}_\mathcal{D}(w_\varepsilon)$ with respect to $\varepsilon$.
>
>           To see this, if $\varepsilon = 1$ (no training, high bias, low variance), then $\|w^{(0)} - w_\varepsilon\| = 0$, and the bound is dominated by the initial loss; if $\varepsilon = \bar{M}_\delta/M_\delta$
>
>            (convergence model, low bias, high variance), then the generalization bound is dominated by the model complexity $r_{n, \delta, \varepsilon}$.
>
>      **To make this point more clear, we have added the above discussion before Theorem 3.3.**
>
> 3. ''The last sentence in Theorem 3.3 is vague and needs further clarification. The sentence says "Then for any parameter $w_\varepsilon$ with $\mathcal{L}_n(w_\varepsilon) = \varepsilon \mathcal{L}_n(w^{(0)})$ for some $\varepsilon \in [0, 1]$, we have ...''
>     Does this sentence apply to every $\varepsilon \in [0, 1]$ or only a specific $\varepsilon$ value? Also, does the bound hold uniformly with probability $1 - \delta$ for all $w_\varepsilon$ or does it hold for every individual $w_\varepsilon$ with probability $1-\delta$? The current presentation of this theorem is somewhat vague and the answer to these questions is unclear.''
>
>       * Thanks for pointing out the unclear statement. Here we mean that ``for any given $ \varepsilon, \delta \in [0, 1]$, with probability at least $1-\delta$ over $S$, the generalization bound of any parameter $w_\varepsilon$ with $L_n(w_{\varepsilon}) = \varepsilon L_n(w^{(0)})$ is given by...''
>
>         **In light of your comments, we have modified the statement of Theorem 3.3.**

---

> > ### Author Response · Authors · 2022-08-22
> > **Author's response**
> >
> > 4. ''The model assumed in Equation (4) looks inefficient for multilayer neural nets with more than one hidden layer, because it requires including all the neural net weights after layer $1$ as $\beta$ parameters. Also, the number of $\beta$ parameters ($q$) appears in the generalization bound which will be unsatisfactorily large for multilayer neural nets. Is there a way to replace $q$ with the norm of weight matrices for a multilayer neural net?''
> >
> >     * Thanks for the suggestion. We agree that for multilayer neural nets, the number of $\beta$ parameters will be large.
> >         For example, for three layer neural networks with identical width $m$, $q = 2m^2$.
> >         The improvement for the model representation is an interesting future direction, and the suggested direction of considering norm of matrices (be it weight matrices of gradient matrices for example) could be interesting (especially if such matrices can be shown to be essentially low rank) and we plan to investigate this direction in future work.
> >
> >
> > 5. ''It seems unintuitive to me that $\sqrt{2 \varepsilon \mathcal{L}_n(w^{(0)})}$ appears in the bounds in Theorems 4.3, 4.4, and 4.7.
> >     Should not the point of generalization analysis be to bound $\mathcal{L}_\mathcal{D}(w_\varepsilon) - \mathcal{L}_n(w_\varepsilon)$?
> >     Then, why do the authors bound $\mathcal{L}_\mathcal{D}(w_\varepsilon) - \sqrt{2 \varepsilon \mathcal{L}_n(w^{(0)})}$ in Theorems 4.3, 4.4, and 4.7?''
> >
> >       * In the application part, we consider the population risk with respect to a new loss $\tilde{\ell}$ that is *globally* Lipschitz and globally bounded.
> >         The reason for evaluating the generalization in a new loss function is to simplify the notations. To see this,
> >         notice that for each application, we consider regression problems with squared loss $\ell(y, \hat{y}) = (y - \hat{y})^2/2$, which is not globally Lipschitz nor globally bounded. Thus, applying the Theorem 3.3 requires to bound $L_\ell(S_{a, b})$ and $M_{a, b}$ which depend on $a, b$ and the model architecture. In the revised paper, we consider to evaluate the generalization in a new loss $\tilde{\ell}$ that is globally Lipschitz and globally bounded.
> >         Specifically, let $\tilde{\ell} : \mathbb{R} \times \mathbb{R} \xrightarrow{} [0, l_0]$ that is $\sqrt{l_0}$-Lipschitz (on the first argument) for some $l_0>0$ and $\tilde{\ell} (y, y) = 0$. This can be achieved by truncating $\ell$ at $|y - \hat{y}| = \sqrt{l_0}$, and then using a constant extension past the truncated point to make it continuous. In that case,
> >         $L_\ell(S_{a, b}) = \sqrt{l_0}$, $M_{a, b} = l_0$ for any $a,b$.
> >         Finally, to use Theorem 3.3, it remains to estimate the new empirical loss value $\frac{1}{n} \sum_{i=1}^n \tilde{\ell}(f(w_\varepsilon, x_i), y_i)$ in terms of $\mathcal{L}_n(w_\varepsilon)$.
> >
> >         By the Lipschitz property and Cauchy–Schwarz inequality, one can show that $\frac{1}{n} \sum_{i=1}^n \tilde{\ell}(f(w_\varepsilon, x_i), y_i) \leq \sqrt{2l_0  \mathcal{L}_n(w_\varepsilon)}$ (see the inequality (22) in Appendix E.1). Combining the above derivations, applications of the framework are straightforward.
> >
> >
> >       * One can also get generalization bounds under the squared loss $\ell$. Note that $M_{a,b} \leq \sup_{w \in S_{a,b}, \|x\| \leq 1} f(w,x)^2+1$ and $L_\ell(S_{a, b}) \leq \sup_{w \in S_{a,b}, \|x\| \leq 1} |f(w,x)|+1$. Then we can get an upper bound for $\mathcal{L}_\mathcal{D}(w_\varepsilon)$ by Theorem 3.3.
> >
> >     **In light of your comments, we have added the above explanation before Application 4.1.**
> >
> >
> > 6. Requested Change 1.
> >
> >     * We have modified the statement of Theorem 3.3 and added more explanations on the connections between optimization and generalization before Theorem 3.3.
> >
> > 7. Requested Change 2.
> >
> >      * We have added the explanations on why the square root of the loss function appears in the bounds of the application parts before Application 4.1.
> >
> >
> > 8. Requested Change 3.
> >
> >      * We have added the explanations of bounding the test error rather than the generalization gap before Theorem 3.3.

---

### Author Response · Authors · 2022-08-14
**To all reviewers**

1. We would like to thank all the reviewers for their comments and suggestions to improve this paper.
2. We have answered all the concerns in the comments below. In addition, based on the feedback, we have revised our paper and highlighted major changes in blue. The corresponding changes are emphasised in boldface in the response.
3. Since the third review has yet to be submitted, we decided to submit the revised manuscript before the collection of all the reviews to address the existing reviews. We will address any concerns by the remaining reviewer once the third review comes out.

---

### Decision · Action_Editors · 2022-09-29

**Recommendation:** Accept as is

**Comment:**

The paper studies the optimization and generalization of machine learning problems which satisfy the Uniform-LGI condition. As the theoreical application, the authors derived interesting bounds for linear regression, kernel regression, and two-layer neural network models. The concerns raised by the reviewers are well addressed in the response. Two reviewers recommended "accept". One reviewer recommended borderline rejection since he still has concerns about the novelty of the paper.  Based on the overall recommendations of all the reviewers, I therefore recommend its acceptance.

---

> ### Author Response · Authors · 2022-10-05
> **Camera-ready version**
>
> We would like to thank the Action Editor and all the reviewers for their helpful suggestions and comments. We have uploaded the camera-ready version and added the code link as required.